# HiLoRA: Adaptive Hierarchical LoRA Routing for Training-Free Domain Generalization

## Abstract

Low-Rank Adaptation (LoRA) has emerged as a widely used technique for adapting large language models (LLMs) to new domains, due to its modular design and broad availability on platforms such as HuggingFace. This availability has motivated efforts to reuse existing LoRAs for domain generalization. However, existing methods often rely on explicit task labels or additional training, which are impractical for deployment. Moreover, they typically activate a fixed number of entire LoRA modules, leading to parameter redundancy or insufficiency that degrade performance. In this paper, we propose `HiLoRA`, a training-free framework that performs adaptive hierarchical routing over LoRA pools. Drawing on structural properties of LoRA, we define rank-one components (ROCs), in which each rank parameter is regarded as an independent unit. For a given input sequence, `HiLoRA` first adaptively selects a subset of LoRAs and determines their ROC allocation based on Gaussian likelihoods at the sequence level. At the token level, it further refines routing by activating only the most informative ROCs. We further provide theoretical guarantees that `HiLoRA` selects the most relevant LoRAs with high probability. Extensive experiments show that `HiLoRA` achieves substantial improvements in domain generalization, with accuracy gains of up to 55% over state-of-the-art baselines, while maintaining comparable inference throughput.

## 1 Introduction

Large Language Models (LLMs) have demonstrated remarkable capabilities across a wide variety of tasks (Zhou et al., 2024; Naveed et al., 2025). However, adapting LLMs to specialized domains or tasks requires computationally expensive full fine-tuning (Hu et al., 2022). To mitigate this cost, parameter-efficient fine-tuning (PEFT) techniques have been developed (Ding et al., 2023). Among them, Low-Rank Adaptation (LoRA) (Hu et al., 2022; Tian et al., 2024) has become one of the most effective and widely adopted methods. LoRA introduces lightweight low-rank matrices into selected layers of an LLM, thereby substantially reducing the number of trainable parameters while preserving strong downstream task performance. Building on this success, community platforms such as HuggingFace(HuggingFace, 2025) and ModelScope (ModelScope, 2025) now host thousands of task-specific LoRA modules trained across diverse domains. This rapidly expanding repository creates a unique opportunity: instead of training a new model for every task, one can directly exploit existing LoRAs to achieve scalable multi-domain adaptation.

However, realizing this potential is highly non-trivial, as effectively utilizing community-shared LoRAs introduces several challenges. *First*, explicit task labels of inputs are typically unavailable in practice. If such labels were known, inputs from seen tasks could be directly routed to their specialized LoRAs, while unseen tasks could be aligned with related LoRAs based on task similarity. Without labels, however, distinguishing between seen and unseen cases and assigning appropriate LoRAs becomes highly challenging. *Second*, For a given input, activating too many LoRAs or entire modules leads to parameter redundancy and interference, whereas activating too few may discard valuable knowledge, ultimately reducing accuracy (Cheng et al., 2025). *Third*, as repositories continue to expand with thousands of task-specific LoRAs, the routing mechanism must remain computationally efficient to ensure scalability (Ostapenko et al., 2024).

Recent work has attempted to address the above challenges by integrating Mixture-of-Experts (MoE) mechanisms with LoRAs (Ge et al., 2025), where gating functions are designed to route inputs to a

subset of LoRAs. However, these gating functions often rely on explicit task labels (Ma et al., 2024) or require gradient-based training of additional gating parameters (Muqeeth et al., 2024), which restricts their applicability in practical deployment. Moreover, most methods rely on top-$k$ gating scores (Ostapenko et al., 2024; Zhao et al., 2024), which lead to either excessive or insufficient activations and thus limit adaptability. In parallel, some studies focus on LoRA merging, which integrates multiple task-specific LoRAs into a single unified module to enhance cross-domain generalization by leveraging knowledge across tasks (Coleman et al., 2024; Zhao et al., 2025a). These approaches impose a uniform architecture across tasks, which limits flexibility and degrades performance in scenarios involving diverse tasks. A more detailed discussion of related work is provided in Appendix A. This motivates the following research question:

*Can we adaptively leverage a large collection of specialized LoRA modules to support both seen and unseen tasks without retraining or explicit task labels?*

In this paper, we highlight **three key observations** about the structure of LoRA, derived from empirical analysis and experimental evidence. (i) Each rank-one direction in a LoRA is formed by pairing a row vector from the down-projection matrix with a corresponding column vector from the up-projection matrix. Since these directions function independently, one can treat each pair as a *rank-one component (ROC)*, which serves as the basic unit of LoRA. (ii) Within a LoRA, the down-projection vectors across ROCs exhibit strong randomness and primarily serve as scaling factors that modulate the effect of the corresponding up-projection vectors. (iii) In contrast, the up-projection vectors show clear clustering patterns, often forming multiple groups within the same LoRA. These clusters capture distinct semantic aspects of the LoRA's adaptive capacity.

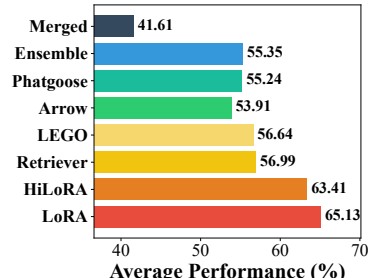

Figure 1: Average accuracy over ten NLI tasks, with five seen tasks and five unseen tasks. `HiLoRA` achieves the best performance and approaches the accuracy of task-specific LoRAs. Detailed results are shown in Tab. 1.

Building on these insights, we propose `HiLoRA`, a hierarchical LoRA routing framework designed to adaptively support robust domain generalization. To the best of our knowledge, `HiLoRA` is the first method to introduce hierarchical routing at the granularity of ROCs, while also providing theoretical guarantees for LoRA identification through error bounds. At the sequence level, `HiLoRA` narrows the candidate space and improves robustness by activating only a subset of LoRAs based on input-LoRA similarity. To enable comparison between inputs and LoRAs that reside in different parameter spaces, each LoRA is represented as a Gaussian distribution fitted to a small set of sampled embeddings, and similarity is measured using Gaussian likelihoods. This probabilistic formulation not only allows reliable distinction between seen and unseen tasks, but also provides confidence signals that guide the adaptive determination of both the number of activated LoRAs and their ROC allocation. At the token level, the down-projection vectors within ROCs are used to further select the most informative ROCs, refining routing without introducing additional parameters or requiring training. We summarize our contributions as follows.

- **New Insight.** We identify the ROC as the fundamental semantic unit of LoRA and show both the feasibility and necessity of performing routing at this fine-grained granularity.
- **Hierarchical LoRA Routing Framework.** `HiLoRA` constructs a dynamic LoRA pool, where each LoRA is represented as a Gaussian distribution fitted from samples of its training dataset. At the sequence level, the Gaussian likelihood scores between the input and LoRAs are calculated. The maximum score determines both the number of activated LoRAs and the overall ROC budget, while normalized scores guide probabilistic sampling for ROC allocation. At the token level, routing is further refined by selecting ROCs with stronger down-projection responses.
- **Theoretical Guarantee.** We derive error bounds for LoRA identification, providing the first formal guarantees that `HiLoRA` preserves the corresponding LoRAs for seen tasks and the closest LoRAs for unseen tasks with high probability, thereby ensuring robust routing across domains.
- **Experimental Performance.** As shown in Fig. 1 for a representative case, `HiLoRA` consistently outperforms state-of-the-art baselines in both within-cluster and cross-cluster evaluations, achieving accuracy gains of up to $55\%$ on LLaMA2-7B and $13\%$ on FLAN-T5-large, while maintaining practical inference throughput.

## 2 PRELIMINARIES

**Basic Formulation of LoRA.** LoRA (Hu et al., 2022) achieves performance comparable to full fine-tuning by freezing the pretrained weights $W_0$ and inserting trainable low-rank matrices $\Delta W$ into selected layers, yielding $W' = W_0 + \Delta W$. The update matrix is factorized as $\Delta W = BA$, where $A \in \mathbb{R}^{r \times d}$ is the down-projection matrix and $B \in \mathbb{R}^{d \times r}$ is the up-projection matrix, with rank $r \ll d$. This reduces the number of trainable parameters from $d^2$ to $2rd$ while retaining strong adaptability. Given an input $x \in \mathbb{R}^d$, the sub-module output $y \in \mathbb{R}^d$, originally computed as $y = W_0 x$, is reformulated under LoRA adaptation as:

$$y = W_0 x + \Delta W x = W_0 x + BA x. \tag{1}$$

**Dyadic Product Representation.** Let $\{a_i^\top\}_{i=1}^r$ denote the set of row vectors of $A$ and $\{b_i\}_{i=1}^r$ denote the set of column vectors of $B$, where $a_i, b_i \in \mathbb{R}^d$. Under this notation, the low-rank update can be written as $\Delta W = BA = \sum_{i=1}^r (b_i a_i^\top)$, which expresses $\Delta W$ as a sum of $r$ dyadic products, each formed by the outer product of two vectors $(a_i, b_i)$. Substituting this representation into the forward computation yields:

$$y = W_0 x + \sum_{i=1}^r (b_i a_i^\top) x. \tag{2}$$

In this decomposition, each row of the down-projection matrix $A$ is paired with the corresponding column of the up-projection matrix $B$. The pair $(a_i, b_i)$ acts as an indivisible unit, which we define as a rank-one component (ROC). A ROC corresponds to one rank in LoRA and serves as the fundamental element of its adaptive capacity. Consequently, the ROC constitutes the minimal routing unit of LoRA, and we next introduce an adaptive strategy to determine both the number and the selection of ROCs to activate for each input.

## 3 METHODOLOGY

### 3.1 HILORA FRAMEWORK

**Problem Formulation.** Consider a pre-trained LLM $L$ and a pool of $I$ task-specific LoRAs, denoted as $\Phi = \{\phi_1, \phi_2, \ldots, \phi_I\}$. It is implemented by inserting low-rank matrices into selected layers of $L$. For clarity, the low-rank parameters of $\phi_i$ at a given layer are denoted as $A_i$ and $B_i$, with rank $r_i$. Our objective is to design a *routing mechanism* that exploits the pool of LoRAs $\Phi$ without requiring additional training or explicit task labels. Such a mechanism should perform competitively on tasks with corresponding LoRAs available in the pool (*seen tasks*), while also generalizing to inputs from domains lacking specialized LoRAs (*unseen tasks*).

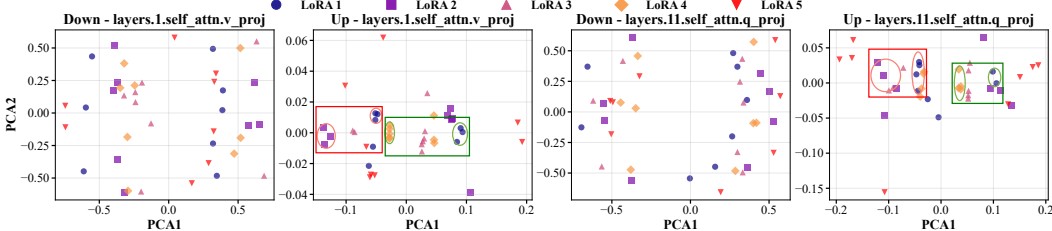

Figure 2: Scatter plots of the first two principal components derived from vectors in LoRA projection matrices specialized for five NLI tasks. The boxes highlight examples where optimal routing for an unseen task (pink) would involve selecting only the vectors aligned with relevant semantics.

**Motivating Observations.** The functional distinction between the down- and up-projection matrices follows directly from the structure of the LoRA update. Each rank-one component operates as $(b_i a_i^\top) x = (a_i^\top x) b_i$, indicating that the down-projection vector $a_i$ governs the activation strength of the component, whereas the semantic direction of the update is entirely determined by the up-projection vector $b_i$. This interpretation aligns with prior observations (Zhu et al., 2024; Tian et al., 2024). To further validate this distinction and examine additional properties of ROCs, we visualize LoRA parameters using Principal Component Analysis (PCA) (Abdi & Williams, 2010). In particular, vectors obtained by slicing the projection matrices along the rank dimension, *i.e.*, $\{a_i, b_i\}_{i=1}^r$, are projected into a two-dimensional space. We analyze five LoRAs fine-tuned on different NLI tasks, with the resulting scatter plots shown in Fig. 2, where vectors sharing the same color and shape are drawn from the same LoRA. To ensure that the reported observations are not limited to

these cases, additional visualizations are provided in the Appendix C.2. To further substantiate our claims, we also compute cosine similarities for both the down- and up-projection matrices at the rank-component level and aggregate the statistics, as presented in Appendix C.2 (Fig. 9-12).

Three key observations arise from these visualizations. (i) The down-projection vectors of ROCs exhibit a highly dispersed distribution and show little alignment with task semantics. This confirms that down-projection vector $\boldsymbol{a}$ primarily functions as a scaling factor, rather than encoding domain-specific information. (ii) In contrast, the up-projection vectors of ROCs within a given LoRA exhibit clear task-dependent patterns. These vectors often form multiple distinct clusters, with each cluster representing a different semantic fragment of the LoRA's adaptive capacity. (iii) For domain generalization, activating an entire LoRA introduces parameter redundancy and interference, since unrelated clusters are involved simultaneously. Taken together, these observations suggest that effective routing should selectively activate only those clusters or vectors aligned with relevant semantics. As illustrated in Fig. 2, when the pink LoRA corresponds to an unseen task, the optimal routing selectively activates only specific clusters (*e.g.*, the red box selects purple and blue clusters, while the green box selects orange and blue clusters). Similarly, in the fourth subfigure, the activated ROCs originate from the purple, blue, and orange clusters, although the precise cluster assignments differ.

**Workflow of `HiLoRA`.** Motivated by these observations, routing at the granularity of ROCs is highly desirable. However, directly selecting ROCs from the entire LoRA pool faces two main challenges. First, the candidate space is excessively large, which makes exhaustive selection computationally infeasible. Second, the space is noisy, as ROCs from different LoRAs vary in relevance and quality, making it difficult to evaluate them under a unified criterion. To address these issues, we introduce `HiLoRA`, an adaptive hierarchical routing framework over a pool of task-specific LoRAs designed to achieve training-free domain generalization. Given an input sequence $\boldsymbol{x}$, `HiLoRA` operates in two stages. (i) *Input-Aware ROC Allocation*:

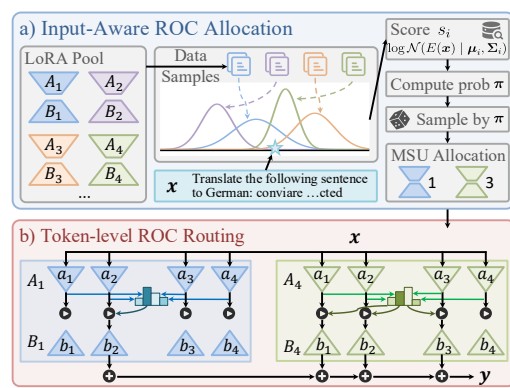

Figure 3: Overview of `HiLoRA` architecture.

At the sequence level, the framework measures the similarity between $\boldsymbol{x}$ and each LoRA $\phi_i$ using Gaussian likelihoods. Based on these probabilistic similarities, it selects a subset of LoRAs and assigns an appropriate number of ROCs to each. (ii) *Token-Level ROC Routing*: At the token level, the framework further refines adaptation by dynamically routing each token in $\boldsymbol{x}$ to the most relevant ROCs within the subset of LoRAs selected in stage (i). In both stages, comparisons are performed under a unified criterion, which ensures fair evaluation across LoRAs and their ROCs. The overview of our framework `HiLoRA` is illustrated in Fig. 3.

## 3.2 INPUT-AWARE ROC ALLOCATION

At the sequence level, the goal is to identify candidate LoRAs from the pool and allocate a suitable number of ROCs to each, according to their relevance to the input. A key challenge arises because the input representations and LoRA parameters reside in distinct spaces, which prevents direct comparison. To address this issue, inspired by retrieval-based methods, each LoRA can be represented by a small set of samples drawn from its training dataset (Zhao et al., 2024). Instead of embedding LoRAs and inputs into a shared space and computing cosine similarity, we approximate each LoRA with a Gaussian distribution fitted to the sampled embeddings. This yields a probabilistic representation that enables more robust matching (Cha et al., 2021; Li et al., 2023). This probabilistic representation provides an information-theoretic characterization: inputs from seen tasks attain high likelihood under their corresponding LoRA distributions, while inputs from unseen tasks can still be aligned by evaluating their likelihood across all source distributions. Moreover, the resulting probabilities guide stochastic allocation of ROCs, which not only reduces over-reliance on a single LoRA but also encourages exploration across multiple relevant candidates.

Formally, let $E$ denote a sentence embedding model and $\boldsymbol{c}$ denote an instruction. The instructed embedding of an input $\boldsymbol{x}$ is given by $\boldsymbol{z} = E(\boldsymbol{c} \oplus \boldsymbol{x})$, where $\oplus$ denotes concatenation. Following

Zhao et al. (2024), we set the instruction to "Represent the sentence for similar task retrieval" to encourage sequence-level similarity. For each LoRA module $\phi_i$, we randomly sample $m$ domain-specific examples, obtain their instructed embeddings $\{z_1^{(i)}, \ldots, z_m^{(i)}\}$, and fit a Gaussian distribution:

$$p_i(z) = \mathcal{N}(z \mid \boldsymbol{\mu}_i, \boldsymbol{\Sigma}_i), \text{ where } \boldsymbol{\mu}_i = \frac{1}{m}\sum_{j=1}^m z_j^{(i)}, \ \boldsymbol{\Sigma}_i = \frac{1}{m-1}\sum_{j=1}^m \left(z_j^{(i)} - \boldsymbol{\mu}_i\right)\left(z_j^{(i)} - \boldsymbol{\mu}_i\right)^\top + \varepsilon \boldsymbol{I}. \quad (3)$$

where $\boldsymbol{I}$ is the identity matrix and $\varepsilon$ is a small constant. The term $\varepsilon \boldsymbol{I}$ prevents degeneracy by ensuring that $\boldsymbol{\Sigma}_i$ remains full-rank and well-conditioned, which is essential when the number of samples is small or when the empirical covariance is nearly singular. For a given input $x$, we then compute its log-likelihood under each LoRA distribution as the similarity score: $s_i(x) = (1/\tilde{d}) \log p_i(z), \forall i \in \{1, \ldots, I\}$, where $\tilde{d}$ denotes the embedding dimension. Since inputs may come from either seen or unseen tasks, with seen tasks typically producing higher scores. Therefore, two cases are considered depending on whether a positive score is present:

$$\mathbb{C}(x) = \begin{cases} \{i \mid s_i(x) > 0\}, & \text{if } \max_i s_i(x) > 0, \\ \arg \text{top}_i^c s_i(x), & \text{if } \max_i s_i(x) \le 0, \end{cases} \quad (4)$$

where $c = \lceil |\max_i s_i(x)| \rceil$. A positive maximum score indicates that at least one LoRA is well aligned with the input, so we retain only LoRAs with a positive score. Otherwise, the Top-$c$ LoRAs are selected, thereby expanding the candidate set to the degree of misalignment. A more negative maximum score indicates that all LoRAs exhibit low compatibility with the input, and increasing the number of candidates in such cases improves the chance of capturing a relevant LoRA.

Because the set of activated LoRAs $\mathbb{C}(x)$ varies across inputs, the ROC budget also needs to adapt dynamically. Using a fixed number of ROCs can easily become suboptimal: when only a few LoRAs are selected, a static budget may introduce redundancy, whereas selecting many LoRAs may lead to insufficient capacity. Therefore, the total ROC budget is defined as $O(x) = \gamma \cdot \sum_{i \in \mathbb{C}(x)} r_i$, where $\gamma \in (0,1)$ is a scaling factor. A large value of $\gamma$ may introduce redundancy and interference, whereas a small value may exclude essential information. Thus, $\gamma$ is set to balance accuracy and efficiency by activating a compact yet sufficient set of ROCs. To allocate ROCs, we use Gaussian-likelihood scores as they offer a principled measure of input–LoRA alignment, enabling proportionally allocating more ROCs to better-matched LoRAs. Concretely, the scores of selected LoRAs are normalized into probabilities: $\pi_i(x) = \frac{\exp(s_i(x))}{\sum_{j \in \mathbb{C}(x)} \exp(s_j(x))}, \forall i \in \mathbb{C}(x)$. Using these probabilities, the ROC allocation $\{o_i\}_{i \in \mathbb{C}(x)}$ is sampled from a multinomial distribution with parameters $O(x)$ and $\boldsymbol{\pi}(x)$, subject to the per-LoRA capacity constraint $o_i \le r_i$.

### 3.3 TOKEN-LEVEL ROC ROUTING

**ROC Routing within Chosen LoRAs.** At the token level, routing is refined by operating on the granularity of ROCs. As discussed in Sec. 3.1, the down-projection vectors mainly act as scaling factors. Therefore, the projection value $a^\top x$ provides a natural criterion for ROC selection, with larger values indicating stronger relevance between the token and the corresponding ROC. This criterion helps reduce redundancy by prioritizing the most informative ROCs while filtering out those with limited contribution or potential interference. Formally, for each layer and each token, and for every LoRA $i \in \mathbb{C}(x)$ selected at the sequence level, we compute the projection values $\boldsymbol{A}_i x$. The most informative ROCs are then identified by selecting the indices of the top-$o_i$ components ranked by projection value: $\mathbb{J}_i = \arg \text{top}_j^{o_i}(a_{ij}^\top x)$. The LoRA output for this layer is then obtained by aggregating the contributions of all activated ROCs: $y' = \sum_{i \in \mathbb{C}(x)} \sum_{j \in \mathbb{J}_i} b_{ij}(a_{ij}^\top x)$.

It is important to emphasize that this routing introduces no additional parameters or retraining. Since projection values $a^\top x$ are required for all activated ROCs, the only extra computation arises from evaluating projections of ROCs that are ultimately not selected. This overhead is minimal compared to the overall forward pass, ensuring efficiency while preserving robust adaptation.

**Variance Normalization for Adaptive ROCs.** In `HiLoRA`, the number of activated ROCs is adaptive and may range from $1$ to $\sum_{i=1}^I r_i$, where $r_i$ is the rank of LoRA $\phi_i$. This variability can cause fluctuations in the scale of the aggregated LoRA output, which in turn may reduce the stability of model performance. Empirical findings in (Zhao et al., 2025a) show that LoRA outputs are approximately distributed as zero-mean Gaussians, with variance that grows with the number of activated ROCs. To mitigate this effect, we normalize the aggregated output by a scaling factor $\sqrt{\bar{r}(x)/O(x)}$, where $\bar{r}(x) = \frac{1}{|\mathbb{C}(x)|} \sum_{i \in \mathbb{C}(x)} r_i$ is the average rank of the

selected LoRAs (Vaswani et al., 2017). Therefore, the output of a given layer for input $x$ becomes: $y = W_0 x + \sqrt{\bar{r}/O(x)} \sum_{i \in \mathbb{C}(x)} \sum_{j \in \mathbb{J}_i} b_{ij}^\top (a_{ij}^\top) x$. This variance normalization property has been formally established in Theorem 3.1 of (Zhao et al., 2025a). For clarity and completeness, we restate it as a Lemma 2 in Appendix B.

## 3.4 THEORETICAL ANALYSIS

We present the error bounds of LoRA identification in `HiLoRA` under two scenarios: (i) *in-distribution (ID)* inputs from seen tasks, and (ii) *out-of-distribution (OOD)* inputs from unseen tasks.

**Error Bound for ID Inputs.** For inputs from seen tasks, we provide a Top-$k$ error bound that measures the probability of the corresponding LoRA being excluded from the selected set.

**Lemma 1** *For any two distributions $i, j$ with class-conditional Gaussians $\mathcal{N}(\boldsymbol{\mu}_i, \boldsymbol{\Sigma}_i)$ and $\mathcal{N}(\boldsymbol{\mu}_j, \boldsymbol{\Sigma}_j)$ and prior probabilities $\pi_i, \pi_j$, the Bayes error rate satisfies: $P_{\mathrm{err}}^{(2)}(i,j) \leq \sqrt{\pi_i \pi_j} \, \exp(-B_{ij})$, where*
$B_{ij} = \frac{1}{8}(\boldsymbol{\mu}_i - \boldsymbol{\mu}_j)^\top \left(\frac{\boldsymbol{\Sigma}_i + \boldsymbol{\Sigma}_j}{2}\right)^{-1} (\boldsymbol{\mu}_i - \boldsymbol{\mu}_j) + \frac{1}{2} \log \frac{|(\boldsymbol{\Sigma}_i + \boldsymbol{\Sigma}_j)/2|}{\sqrt{|\boldsymbol{\Sigma}_i||\boldsymbol{\Sigma}_j|}}$.

In this paper, priors are not incorporated in the score. The same derivation yields the simplified form $P_{\mathrm{err}}^{(2)}(i,j) \leq \exp(-B_{ij})$. Based on this Lemma, we have the following error bound.

**Theorem 1** *For an input $x$ with true label $t_i$, the prediction is determined by the top-$k$ scores $s_i(x)$. The probability that the LoRA corresponding to $t_i$ is not included in the Top-$k$ set $\mathbb{K}$ is bounded as:*

$$\Pr\left(i \notin \mathbb{K}\right) \leq \frac{1}{k} \sum_{j \neq i} \exp\left(-B_{ij}\right). \tag{5}$$

Theorem 1 shows that for ID inputs, the probability of excluding the correct LoRA decreases in two ways: (1) it drops exponentially as task distributions become more separable (larger $B_{ij}$); and (2) it decreases proportionally with the size of the Top-$k$ set.

**Error Bound for OOD Inputs.** For an input $x$ from unseen tasks, no exact task-specific LoRA exists in the pool, suppose it comes from an unknown target distribution $q$. Define the information-theoretically closest source domain as $i^\star := \arg\min_{i \in \{1, \cdots, I\}} D_{\mathrm{KL}}(q \, \| \, p_i)$.

**Theorem 2** *Let the prediction be based on the top-$k$ scores $s_i(x)$. For any $\alpha \in (0, 1]$ and $M_\alpha^j = \boldsymbol{\Sigma}_q^{-1} + \alpha \boldsymbol{\Sigma}_j^{-1} - \alpha \boldsymbol{\Sigma}_{i^\star}^{-1} \succ 0$, the probability that the LoRA $i^\star$ is excluded from the Top-$k$ set $\mathbb{K}$ satisfies:*

$$\Pr\left(i^\star \notin \mathbb{K}\right) \leq \frac{1}{k} \sum_{j \neq i^\star} C_\alpha^j \, |M_\alpha^j|^{-1/2} \exp\left(\frac{1}{2}(h_\alpha^j)^\top (M_\alpha^j)^{-1} h_\alpha^j - K_\alpha^j\right), \tag{6}$$

*where $h_\alpha^j = \boldsymbol{\Sigma}_q^{-1} \boldsymbol{\mu}_q + \alpha \boldsymbol{\Sigma}_j^{-1} \boldsymbol{\mu}_j - \alpha \boldsymbol{\Sigma}_{i^\star}^{-1} \boldsymbol{\mu}_{i^\star}$, $K_\alpha^j = \frac{1}{2} \boldsymbol{\mu}_q^\top \boldsymbol{\Sigma}_q^{-1} \boldsymbol{\mu}_q + \frac{\alpha}{2}(\boldsymbol{\mu}_j^\top \boldsymbol{\Sigma}_j^{-1} \boldsymbol{\mu}_j - \boldsymbol{\mu}_{i^\star}^\top \boldsymbol{\Sigma}_{i^\star}^{-1} \boldsymbol{\mu}_{i^\star})$, $C_\alpha^j = \exp(-\frac{\alpha}{2} \log|\boldsymbol{\Sigma}_j| + \frac{\alpha}{2} \log|\boldsymbol{\Sigma}_{i^\star}| - \frac{1}{2} \log|\boldsymbol{\Sigma}_q|)$.*

Here, $M_\alpha^j$ is a weighted precision matrix combining the covariance information of $q$, $j$, and $i^\star$, while the condition $M_\alpha^j \succ 0$ guarantees that the quadratic form is well-defined and divergence is finite; $h_\alpha^j$ is a mean–precision vector measuring the displacement of $q$ relative to $j$ and $i^\star$ under covariance-adjusted weighting; $K_\alpha^j$ is a correction term involving second-order statistics, capturing quadratic differences in alignment; $C_\alpha^j$ is a scale factor derived from covariance determinants, quantifying relative volume mismatch. Theorem 2 shows that for OOD inputs, the probability of excluding the closest LoRA decreases in two ways: (1) it drops exponentially when the unseen distribution $q$ is better aligned with $i^\star$ and more distinct from other source domains $j$; (2) it decreases proportionally with the size of the Top-$k$ set.

**Remarks.** Theorem 1 and Theorem 2 highlight two key insights. (i) When domains are well separated and the LoRA pool spans diverse tasks, the error bounds are tight, ensuring strong guarantees in both ID and OOD cases. This condition is often met in practice, as task domains are generally distinguishable, and open-source repositories already provide a rich collection of LoRAs across diverse tasks. (ii) Increasing $k$ tightens the bound, but excessively large values introduce redundancy and interference. To balance this trade-off, `HiLoRA` adaptively adjusts the size of the activated set based on input-LoRA similarity, retaining the corresponding or closest LoRA with high probability while avoiding unnecessary overhead and parameter interference. We further validate these theoretical assumptions empirically in Appendix D, showing that domain separability, divergence terms, and OOD conditions are consistently satisfied across tasks, reinforcing the practical relevance of our error bounds.

## 4 EXPERIMENTS

### 4.1 EXPERIMENTAL SETUP

**Datasets and Models.** We use a subset of tasks from FLAN-v2 (Wei et al., 2022), and organize them into ten clusters: Natural Language Inference (NLI), Question Answering (QA), Sentiment Analysis, Translation, Commonsense Reasoning, Paraphrase, Struct-to-Text, Coreference Resolution, Text Correction, and Word-level tasks, following the categorization in Wei et al. (2022). We construct the LoRA pool by downloading task-specific LoRAs for the selected tasks from HuggingFace. Since evaluation metrics vary across tasks, we adopt task-dependent measures including accuracy, F1 score, BLEU, and ROUGE-1, 2, L. Details of the selected tasks, their grouping, and metrics are provided in Appendix C.1. As backbone models, we use two representative LLMs: LLaMA2-7B, LLaMA2-13B (Touvron et al., 2023) and FLAN-T5-large (Chung et al., 2024).

**Baselines.** We compare `HiLoRA` with the following state-of-the-art methods. (i) `HiLoRA-GS`: a variant of `HiLoRA` that applies only sequence-level routing. (ii) `HiLoRA-ROC`: a variant of `HiLoRA` that applies only token-level routing by ranking all ROCs across LoRAs and selecting the top-$k$. (iii) `Retriever` (Zhao et al., 2024): a sequence-level method that retrieves the top-$k$ LoRAs based on cosine similarity between input and LoRA embedding. (iv) `LEGO` (Zhao et al., 2025a): a ROC-level merging method that clusters all ROCs into $k$ groups, merges each cluster into a new ROC, and applies the merged clusters to all tasks. (v) `Arrow` (Ostapenko et al., 2024): a token-level routing approach that builds gating vectors from the first right singular vector of the LoRA update $BA$. (vi) `Phatgoose` (Muqeeth et al., 2024): a token-level routing method where gating vectors are trained separately for each task. (vii) `Ensemble` (Mühlematter et al., 2024): an ensemble method that combines all LoRAs by averaging their outputs. (viii) `Merged` (Ostapenko et al., 2023): a method where all LoRAs are merged into a single module shared across tasks.

**Implementation Details.** We set the inference batch size to $32$. For each seen task, $m = 20$ domain-specific samples from the corresponding dataset are used to fit a Gaussian distribution. The sentence embedding model $E$ is implemented with instructor-base (Su et al., 2023), an instruction-tuned encoder that produces task-aware representations. The scaling factor $\gamma$ is fixed at $40\%$. Following (Zhao et al., 2024), we set the parameter $k = 3$ for all LoRA-level routing methods, and correspondingly $k = 24$ for all ROC-level routing methods. All experiments are conducted in PyTorch on a system with Ubuntu 22.04, Intel Xeon Platinum 8558P processors (192 CPUs), 2.0 TiB of memory, and NVIDIA H100 GPUs with 80GB memory.

Table 1: Performance on the NLI cluster using LLaMA2-7B, LLaMA2-13B and FLAN-T5-large. Tasks with a white background are set as *seen* tasks, while those with a gray background are set as *unseen* tasks. For each task, the best accuracy among all methods is in **bold**, and the second best is underlined.

| Methods | LoRA | HiLoRA | HiLoRA-GS | HiLoRA-ROC | Retriever | LEGO | Arrow | Phatgoose | Ensemble | Merged |
|---|---|---|---|---|---|---|---|---|---|---|
| *LLaMA2-7B* | | | | | | | | | | |
| ANLI-r1 | 46.40 | **45.00** | 42.10 | 38.90 | 36.10 | 37.00 | 38.90 | 37.00 | 35.80 | 31.70 |
| ANLI-r2 | 40.10 | **40.60** | 38.70 | 36.20 | 36.40 | 37.70 | 36.40 | 36.40 | 36.80 | 32.60 |
| ANLI-r3 | 36.92 | **37.67** | 36.17 | 35.92 | 35.25 | 34.75 | 36.25 | 35.42 | 34.50 | 31.08 |
| CB | 80.00 | 68.00 | 70.00 | 64.00 | 66.00 | 66.00 | 64.00 | **74.00** | 68.00 | 56.00 |
| MNLI | 77.66 | **76.33** | 74.06 | 70.78 | 74.22 | 71.91 | 60.51 | 62.58 | 67.66 | 39.92 |
| MNLI-mis | 79.69 | **78.59** | 74.69 | 69.38 | 75.78 | 71.80 | 60.82 | 62.34 | 68.75 | 40.59 |
| QNLI | 77.27 | **78.28** | 77.23 | 59.02 | 62.19 | 58.71 | 59.02 | 59.80 | 57.89 | 45.23 |
| RTE | 72.96 | 74.44 | **75.56** | 65.93 | 65.93 | 71.11 | 75.56 | 74.07 | 71.48 | 53.70 |
| SNLI | 67.42 | 69.45 | 68.13 | 69.34 | **70.94** | 67.46 | 59.06 | 57.89 | 62.58 | 35.27 |
| WNLI | 72.86 | **65.71** | 62.29 | 48.57 | 47.14 | 50.00 | 48.57 | 52.86 | 50.00 | 50.00 |
| Avg | 65.13 | **63.41** | 61.89 | 55.80 | 56.99 | 56.64 | 53.91 | 55.24 | 55.35 | 41.61 |
| *LLaMA2-13B* | | | | | | | | | | |
| Avg | 74.00 | **72.86** | 72.47 | 67.39 | 70.32 | 69.04 | 63.12 | 65.31 | 65.99 | 36.00 |
| *FLAN-T5-Large* | | | | | | | | | | |
| Avg | 67.81 | **67.70** | 64.85 | 66.53 | 66.76 | 56.20 | 57.81 | 55.29 | 56.19 | 53.03 |

### 4.2 MAIN RESULTS

Experimental results are reported under two evaluation settings: (i) the within-cluster setting evaluates performance when test tasks originate from the same cluster as the seen tasks, and (ii) the cross-cluster setting measures generalization to tasks from unseen clusters.

**Within-cluster Setting.** In this setting, experiments are conducted on ten NLI tasks, with half designated as seen tasks and the other half as unseen tasks. Results are summarized in Tab. 1, while per-task accuracy for LLaMA2-13B and T5-large is provided in the Appendix C.3 due to page limits. From the table, it can be observed that the proposed `HiLoRA` substantially outperforms all baselines on both *seen* and *unseen* tasks, improving average accuracy by 6-22% on LLaMA2-7B, up to 36% on LLaMA2-13B, and roughly 14% on T5-large. More specifically: (i) On *seen* tasks, `HiLoRA` achieves performance comparable to the oracle setting (LoRA in Tab. 1) where each input is served by its task-specific LoRA, and in some cases even surpasses it, *e.g.* ANLI-r3 and QNLI. This indicates that `HiLoRA` not only identifies the task-specific LoRA corresponding to the given input but also leverages useful ROCs from other LoRAs to further enhance performance. (ii) On *unseen* tasks, `HiLoRA` also delivers consistently strong results, demonstrating its ability to generalize by aligning inputs with semantically related LoRAs and refining predictions through selective ROC activation. (iii) The gains are particularly notable on LLaMA, which relies more heavily on LoRA adaptation than T5-large. Since T5-large has already been extensively pretrained on FLAN-style tasks, the relative contribution of LoRA adaptation is smaller compared to LLaMA. Methods such as `Retriever`, `Arrow`, `Phatgoose`, and `Ensemble` activate a fixed number of LoRAs (or even all of them) without accounting for conflicts or redundancies among ROCs, leading to parameter interference or insufficiency that ultimately degrades performance. `LEGO`, while incorporating ROC clustering and merging, remains input-agnostic and retains all clusters, thereby failing to eliminate parameter redundancy. The `Merged` baseline performs worst due to severe parameter interference when all LoRAs are combined into a single module. In contrast, `HiLoRA` employs a hierarchical routing strategy: at the sequence level, it prunes irrelevant LoRAs via Gaussian similarity sampling, and at the token level, it selects only the most effective ROCs. This design reduces parameter redundancy and prevents interference, and explains the consistent performance gains observed across both seen and unseen tasks.

Table 2: Performance of LLaMA2-7B and FLAN-T5-large under the cross-cluster setting. For tasks with multiple evaluation metrics, the average score across metrics is computed first, and the cluster score is then obtained by averaging over all tasks in the cluster. For each cluster, the best result among all methods is in **bold**, and the second best is underlined.

| Methods | LoRA | HiLoRA | HiLoRA-GS | HiLoRA-ROC | Retriever | LEGO | Arrow | Phatgoose | Ensemble | Merged |
|---|---|---|---|---|---|---|---|---|---|---|
| | | | | *LLaMA2-7B* | | | | | | |
| NLI | 63.13 | **46.54** | 44.23 | 45.00 | 43.78 | 42.89 | 42.29 | 43.78 | 43.57 | 11.69 |
| QA | 59.66 | **46.95** | 43.56 | 43.19 | 43.55 | 46.67 | 39.37 | 45.10 | 44.89 | 10.09 |
| Senti. | 59.87 | **54.43** | 49.88 | 54.00 | 50.12 | 52.93 | 40.76 | 53.00 | 50.26 | 4.19 |
| Trans. | 21.98 | 20.78 | **21.80** | 14.92 | 9.50 | 16.45 | 20.93 | 20.47 | 20.77 | 11.91 |
| Common. | 67.11 | **52.76** | 50.27 | 51.29 | 44.99 | 50.14 | 50.83 | 50.88 | 52.03 | 15.24 |
| Paraph. | 66.88 | 53.08 | 50.11 | 42.73 | **54.51** | 39.91 | 45.09 | 47.31 | 49.06 | 7.61 |
| StT | 44.51 | **28.31** | 28.18 | 24.86 | 27.32 | 15.89 | 27.71 | 28.01 | 27.21 | 24.94 |
| Corefe. | 47.95 | 61.59 | **62.04** | 59.30 | 59.02 | 58.79 | 61.04 | 58.23 | 60.70 | 6.98 |
| Text-Corr. | 54.73 | 30.98 | **33.21** | 25.73 | 26.14 | 24.04 | 29.35 | 29.58 | 29.93 | 6.34 |
| Word | 67.02 | 46.13 | 45.51 | 43.08 | **46.73** | 38.61 | 45.73 | 45.43 | 43.09 | 11.47 |
| | | | | *FLAN-T5-Large* | | | | | | |
| NLI | 67.81 | **63.49** | 58.65 | 63.21 | 62.04 | 52.18 | 50.59 | 62.08 | 52.75 | 49.11 |
| QA | 67.39 | **63.44** | 61.73 | 63.08 | 60.87 | 60.03 | 59.40 | 63.13 | 60.39 | 58.51 |
| Senti. | 59.18 | **58.55** | 58.14 | 58.49 | 57.73 | 58.11 | 57.96 | 58.13 | 58.00 | 57.94 |
| Trans. | 18.97 | 18.79 | 18.80 | 18.55 | **18.88** | 18.77 | 18.74 | 18.61 | 18.77 | 18.65 |
| Paraph. | 78.33 | **75.18** | 74.91 | 68.00 | 72.52 | 73.63 | 72.85 | 74.76 | 73.97 | 72.96 |
| StT | 60.18 | 59.85 | 59.83 | 59.42 | **59.88** | 59.80 | 59.79 | 59.76 | 59.79 | 59.75 |
| Corefe. | 63.13 | **63.89** | 61.63 | 63.61 | 62.04 | 60.95 | 60.95 | 62.07 | 62.04 | 60.68 |
| Text-Corr. | 54.91 | **54.83** | 54.21 | 53.68 | 54.01 | 54.56 | 54.45 | 54.68 | 54.63 | 54.21 |
| Word | 71.55 | 73.35 | 72.22 | 64.01 | 72.10 | 73.86 | 72.59 | **73.63** | 73.91 | 73.40 |

**Cross-cluster Setting.** In this setting, each cluster is treated as unseen in turn, while the remaining clusters serve as seen. For LLaMA2-7B and LLaMA2-13B, the LoRA pool contains all 50 task-specific modules, while for T5-large, only 33 modules are included due to the limited availability of community-provided LoRAs. Performance is evaluated on all tasks within the unseen cluster, with average results reported in Tab. 2 and detailed metrics provided in Appendix C.3. The results of LLaMA2-13B are also provided in Appendix C.3. This configuration is more challenging than the within-cluster settings, as unseen tasks may differ substantially in semantics from the seen ones. Nevertheless, `HiLoRA` achieves strong cross-domain generalization, yielding accuracy gains of up to 55% on LLaMA2-7B and 13% on T5-large. Although it does not always attain the highest score in every cluster, its performance is consistently within 2.5% of the best and remains superior to all baselines. These results highlight the routing capability of `HiLoRA`, which mitigates parameter redundancy and interference even when adapting to previously unseen clusters. Interestingly,

Ensemble performs relatively better in this setting than in the within-cluster case, since activating a larger number of LoRAs helps capture broader information, which is beneficial for serving tasks from unseen clusters. These observations further highlight the advantage of HiLoRA, which adaptively determines the number of activated LoRAs according to input-LoRA similarity, thereby preserving sufficient information while avoiding redundancy as formalized in Eq. (4).

### 4.3 FURTHER ANALYSIS

**Performance of Input Mapping.** To evaluate the input routing capability of HiLoRA, we visualize the similarities among task embeddings across different tasks. Fig. 4 presents a heatmap, where tasks from the same cluster are grouped by *green boxes*. Three observations can be made: (i) Task embeddings within the same domain exhibit higher similarity, indicating that HiLoRA effectively captures relationships across related tasks. (ii) The similarity values exhibit a substantially broader range ($-22$ to 5) compared with the narrower interval of $-1$ to 1 obtained by Retriever (Zhao et al., 2024) (see Appendix C.3). This broader contrast sharpens intra-cluster cohesion while maintaining clear separation across clusters, thereby improving task alignment and reducing the risk of mismatching semantically different tasks. (iii) Unlike other methods, HiLoRA adaptively determines the number of activated Lo-

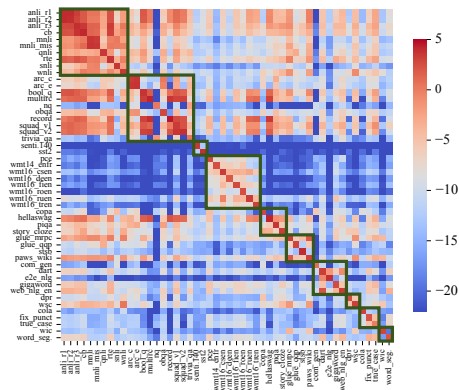

Figure 4: Input-LoRA similarity heatmap produced by HiLoRA, where tasks from the same cluster are enclosed within green boxes for clarity.

RAs based on input-LoRA similarity, (*i.e.*, Eq.(4)). As shown in Fig. 4, for easy cases such as seen tasks, HiLoRA activates only 1-2 LoRAs, whereas in the cross-cluster setting it scales up to 11 LoRAs to handle more dissimilar tasks. This dynamic adaptation and flexibility reduces redundancy while ensuring sufficient coverage. Consequently, to sustain robust performance, the LoRA pool requires a number of modules and reasonable task coverage.

Table 3: Performance sensitivity to sample size, synthetic samples, and embedding models. In within-cluster setting, tasks with a white background are set as *seen* tasks, while those with a gray background are set as *unseen* tasks.

| | Within-cluster setting | | | | | | Cross-cluster setting | | | | |
|---|---|---|---|---|---|---|---|---|---|---|---|
| Factors | 2-sample | 5-sample | 10-sample | 20-sample | AI-sample | MPNet | 2-sample | 5-sample | 10-sample | 20-sample | MPNet |
| ANLI_r1 | 43.30 | 45.00 | 45.00 | 45.00 | 35.80 | 45.00 | 29.40 | 30.70 | 30.50 | 30.70 | 31.90 |
| ANLI_r2 | 38.40 | 39.60 | 38.70 | 40.60 | 38.80 | 39.30 | 27.20 | 28.70 | 30.40 | 34.50 | 34.90 |
| ANLI_r3 | 38.75 | 37.83 | 37.75 | 37.67 | 34.75 | 36.75 | 30.67 | 29.50 | 28.58 | 31.67 | 31.75 |
| CB | 64.00 | 66.00 | 66.00 | 68.00 | 66.00 | 68.00 | 74.00 | 74.00 | 76.00 | 70.00 | 74.00 |
| MNLI | 71.80 | 78.36 | 78.44 | 76.33 | 67.66 | 73.24 | 51.84 | 53.24 | 53.28 | 50.74 | 48.24 |
| MNLI_mis | 73.16 | 80.86 | 81.05 | 78.59 | 66.76 | 73.55 | 52.03 | 53.91 | 53.36 | 51.29 | 49.57 |
| QNLI | 59.22 | 69.06 | 68.20 | 78.28 | 59.34 | 65.80 | 45.61 | 44.38 | 43.48 | 46.84 | 44.77 |
| RTE | 72.22 | 71.85 | 71.85 | 74.44 | 67.41 | 73.33 | 59.63 | 57.78 | 58.15 | 62.22 | 61.85 |
| SNLI | 69.06 | 69.77 | 69.88 | 69.45 | 68.55 | 70.20 | 42.97 | 42.27 | 42.77 | 40.31 | 39.30 |
| WNLI | 54.29 | 60.00 | 64.29 | 65.71 | 50.00 | 61.43 | 47.14 | 48.57 | 48.57 | 47.14 | 45.71 |
| Avg | 58.42 | 61.83 | 62.12 | 63.41 | 55.51 | 60.66 | 46.05 | 46.30 | 46.51 | 46.54 | 46.20 |

**Sensitivity of Input Mapping.** We evaluate the robustness of HiLoRA with respect to three factors that affect its input routing behavior: (i) the number of samples per LoRA $m$, 1 (ii) the use of synthetic proxy samples generated by GPT (with generation prompts provided in Appendix C.4), and (iii) the choice of embedding model $\mathbb{E}$. Experiments are conducted on NLI tasks under both within-cluster and cross-cluster settings, with results summarized in Tab. 3. The results indicate that HiLoRA remains robust across all tested conditions. More specifically: (i) Reducing $m$ leads to only a small accuracy drop (at most 5% on average), and the method remains competitive even with two samples, which is the minimum needed to fit a Gaussian distribution. The within-cluster setting is slightly more sensitive, consistent with the need for more accurate Gaussian fitting when tasks are highly similar. (ii) Synthetic samples yield lower accuracy than real training data, but HiLoRA still performs on par with or better than baselines. In practice, small amounts of task-related examples are typically available from public LoRA repositories, making this assumption reasonable. (iii) Substituting the instructor-tuned embedder with a standard model (MPNet-base-v2 (Song et al., 2020)) results in only a modest degradation ($\leq 3\%$), indicating that HiLoRA is not overly sensitive to the embedding backbone.

**Inference Throughput.** We further evaluate inference throughput by comparing `HiLoRA` with dynamic routing baselines under different numbers of seen tasks, ranging from 5 to 40. For each configuration, inference throughput is measured on a test set containing 5 seen tasks and 5 unseen tasks, numbers. As shown in Fig. 5, throughput decreases gradually as the number of seen tasks increases. Compared with some single-level routing

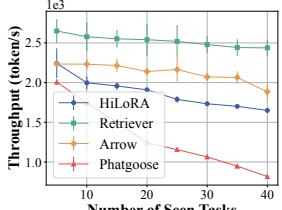

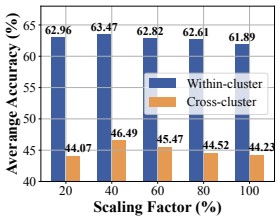

Figure 5: Inference throughput with different numbers of seen tasks.

Figure 6: Performance of `HiLoRA` with different value of scaling factor $\gamma$.

methods, `HiLoRA` incurs a throughput reduction of about 7-30%, but still achieves up to 90% higher throughput compared to `Phatgoose`. Considering the substantial performance gains observed in both within-cluster and cross-cluster settings, this moderate reduction in throughput is acceptable. In addition, we observe that the throughput of `HiLoRA` decreases more slowly as the number of seen tasks grows. This behavior arises from two factors. First, although the input–LoRA mapping latency increases with larger LoRA pools, this component constitutes only a small fraction of the total inference time. Second, the number of selected LoRAs can change in two characteristic ways: (i) If at least one LoRA yields a positive similarity score, the selected set expands only when additional positive-score LoRAs appear as more seen tasks are added. (ii) If all similarity scores are negative, the maximum negative score tends to decrease as more LoRAs are included, leading to a smaller selected set. Once a positive-score LoRA appears, the system transitions back to case (i). These two mechanisms together explain the gradual and occasionally non-monotonic latency patterns observed at the per-task level. However, because throughput is computed as an average over all tasks in the test set, the aggregated trend becomes smoother and exhibits a modest decline followed by slow stabilization as the LoRA pool continues to grow. Detailed latency breakdowns for each individual task are provided in Appendix C.3 (Tab. 8).

**Ablation Study.** Here, we conduct an ablation study on the scaling factor $\gamma$, which controls the number of total ROCs activated. Experiments are performed under both within-cluster and cross-cluster settings, and the evaluation covers all NLI tasks. As shown in Fig. 6, setting $\gamma = 40\%$ yields the best overall performance. To further examine the generality of this behavior, we also evaluate the effect of $\gamma$ on cross-cluster Translation and Struct-to-Text tasks. The complete results are provided in Appendix C.3 (Tab. 9). We observe that the optimal value of $\gamma$ differs across task families, reflecting variations in task complexity and the degree of semantic alignment between the target task and the LoRA pool. Larger values of $\gamma$ activate excessive ROCs, potentially introducing parameter redundancy and interference that reduce performance. Conversely, smaller values may exclude too many informative ROCs, typically leading to insufficient representation capacity and performance degradation. Overall, these results indicate that selecting an appropriate scaling factor is essential for achieving both efficiency and robust performance in `HiLoRA`.

## 5 CONCLUSION

In this paper, we present `HiLoRA`, a training-free framework for adaptive hierarchical routing over pools of task-specific LoRAs to support robust domain generalization. `HiLoRA` builds on structural insights into LoRA by treating each ROC as the minimal routing unit. At the sequence level, it adaptively selects candidate LoRAs and allocates ROCs using Gaussian likelihoods, narrowing the search space and improving robustness. At the token level, routing is further refined by selecting the most informative ROCs, which reduces redundancy and alleviates interference. Theoretical analysis and extensive experiments demonstrate that `HiLoRA` reliably identifies relevant LoRAs, substantially improves domain generalization, and maintains efficiency with only a moderate reduction in inference throughput.

Despite its strengths, `HiLoRA` has several limitations. It relies on a small number of task-specific samples to construct Gaussian representations, which may not always be accessible and could raise privacy concerns. Moreover, the token-level routing mechanism is empirically validated but lacks formal theoretical guarantees. The current routing strategy does not explicitly consider load balancing, which may affect efficiency under large-batch or large-pool scenarios. Future research could focus on addressing these limitations to broaden the practical applicability of the approach.

## REPRODUCIBILITY STATEMENT

We have made extensive efforts to ensure the reproducibility of our work. The formulation of `HiLoRA`, including the definition of rank-one components and the hierarchical routing framework, is described in detail in Sec. 3, with complete theoretical analyses and proofs provided in Appendix B. All datasets and task clusters are drawn from widely used public benchmarks, and the corresponding preprocessing steps and evaluation protocols are fully documented in Appendix C.1. Experimental configurations, including hyperparameter choices, routing parameters, and hardware settings, are reported in Sec. 4, and additional empirical results are provided in Appendix C.3. To further support reproducibility and enable reuse, we will release source code and scripts for dataset preparation upon publication.

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

## A    RELATED WORK

Recent advances in extending LoRA for cross-domain adaptation fall into two primary directions: MoE-style routing and LoRA merging.

**MoE-style Routing.** These methods extend LoRA adaptation by dynamically activating subsets of LoRAs through gating functions (Mao et al., 2025). At the sequence level, routing is performed using task-level similarity or global gating scores to select LoRA experts for the entire input, as in MoA (Feng et al., 2024) and MoLE (Wu et al., 2024). At the token level, methods such as LoRA-Switch (Kong et al., 2024) and Arrow (Ostapenko et al., 2024) introduce token-wise gating to activate different LoRAs for different positions. Hybrid strategies combine these two levels, *e.g.*, HMoRA (Liao et al., 2025) and MoLoRA (Hou et al., 2025), aiming to balance efficiency and flexibility. Beyond entire LoRA routing, rank-level routing has also been explored, where each rank is treated as a micro-expert and subsets are activated, as in SMoRA (Zhao et al., 2025b). Although these methods demonstrate the benefits of dynamic expert selection, they exhibit two key limitations: (i) they typically require training additional gating parameters, which undermines scalability and hinders deployment in training-free scenarios, and (ii) they impose a fixed activation budget, which reduces adaptability when handling diverse or unseen tasks. In contrast, our work introduces a hierarchical routing framework that performs training-free selection at the sequence level and further refines routing at the ROC level, enabling finer-grained control that reduces redundancy and improves robustness across both seen and unseen domains.

**LoRA Merging.** These methods aim to combine multiple task-specific LoRAs into a single unified module to support domain generalization (Huang et al., 2024; Coleman et al., 2024; Qorbani et al., 2025). ZipLoRA achieves effective style and subject composition by directly merging independently trained LoRAs (Shah et al., 2024) for vision and text generation. LoRA-LEGO introduces rank-wise clustering and re-assembly of LoRA ranks to construct merged adapters with adjustable capacity (Zhao et al., 2025a). Beyond heuristic merging, recent works explore more principled strategies: Closed-Form Merging (LoRM) derives analytical solutions for merging parameter-efficient modules in federated continual learning settings (Salami et al., 2025), while Adaptive LoRA Merge with Parameter Pruning further enhances robustness in low-resource domains by combining merging with pruning and lightweight fine-tuning (Miyano & Arase, 2025). While these approaches enhance cross-domain generalization by leveraging knowledge across tasks, they enforce a one-size-fits-all merged model. This limits flexibility and often degrades performance in scenarios involving diverse or unseen tasks. Our work addresses these limitations by designing an adaptive routing framework that adaptively selects LoRAs at the sequence level and refines the choice at the ROC level, providing task-aware composition while reducing redundancy and interference.

## B    THEORETICAL DEMONSTRATION

### B.1    ARIANCE NORMALIZATION PROPERTY

For completeness, we restate the variance normalization property, originally established as Theorem 3.1 in Zhao et al. (2025a). As the full proof is already provided in the cited work, we omit the derivation here and present the result in the form of a lemma below.

**Lemma 2 (Theorem 3.1 in (Zhao et al., 2025a))** *Let $\boldsymbol{A}_1 \in \mathbb{R}^{d \times r}$, $\boldsymbol{B}_1 \in \mathbb{R}^{r \times d}$, and $\boldsymbol{A}_2 \in \mathbb{R}^{d \times k}$, $\boldsymbol{B}_2 \in \mathbb{R}^{k \times d}$, where all entries are independently sampled from the standard normal distribution $\mathcal{N}(0,1)$. If the product $\boldsymbol{A}_2 \boldsymbol{B}_2$ is rescaled by the factor $\sqrt{r/k}$, then the variance of the entries in $\boldsymbol{A}_1 \boldsymbol{B}_1$ coincides with that of the normalized product:* $\mathrm{Var}(\boldsymbol{A}_1 \boldsymbol{B}_1) = \mathrm{Var}\big(\sqrt{\tfrac{r}{k}} \boldsymbol{A}_2 \boldsymbol{B}_2\big)$.

### B.2    PROOF OF LEMMA 1

The Bayes error for the optimal (MAP) decision rule is:

$$P_{\mathrm{err}}^{(2)}(i,j) = \int \min\{\pi_i p_i(\boldsymbol{z}),\ \pi_j p_j(\boldsymbol{z})\}\, d\boldsymbol{z} \ \leq\ \int \sqrt{\pi_i p_i(\boldsymbol{z})\, \pi_j p_j(\boldsymbol{z})}\, d\boldsymbol{z}$$

$$= \sqrt{\pi_i \pi_j} \int \sqrt{p_i(\boldsymbol{z})\, p_j(\boldsymbol{z})}\, d\boldsymbol{z} \ =\ \sqrt{\pi_i \pi_j}\, \rho(p_i, p_j),$$

where the first inequality follows from the elementary bound $\min a, b \leq \sqrt{ab}$ for $a, b \geq 0$, and $\rho(p_i, p_j) := \int \sqrt{p_i p_j}$ denotes the Bhattacharyya coefficient (affinity) between $p_i$ and $p_j$.

For $k \in \{i, j\}$, the Gaussian densities are:

$$p_k(\boldsymbol{z}) = (2\pi)^{-d/2} |\boldsymbol{\Sigma}_k|^{-1/2} \exp\left( -\tfrac{1}{2}(\boldsymbol{z} - \boldsymbol{\mu}_k)^\top \boldsymbol{\Sigma}_k^{-1}(\boldsymbol{z} - \boldsymbol{\mu}_k) \right).$$

Thus, we have:

$$\sqrt{p_i(\boldsymbol{z}) p_j(\boldsymbol{z})} = (2\pi)^{-d/2} |\boldsymbol{\Sigma}_i|^{-1/4} |\boldsymbol{\Sigma}_j|^{-1/4} \exp\left( -\tfrac{1}{2}(\boldsymbol{z} - \tilde{\boldsymbol{\mu}})^\top \tilde{\boldsymbol{\Sigma}}^{-1}(\boldsymbol{z} - \tilde{\boldsymbol{\mu}}) \right) e^{-C},$$

where $\tilde{\boldsymbol{\Sigma}}^{-1} = \frac{1}{2}(\boldsymbol{\Sigma}_i^{-1} + \boldsymbol{\Sigma}_j^{-1}), \tilde{\boldsymbol{\mu}} = \tilde{\boldsymbol{\Sigma}} \cdot \frac{1}{2}(\boldsymbol{\Sigma}_i^{-1}\boldsymbol{\mu}_i + \boldsymbol{\Sigma}_j^{-1}\boldsymbol{\mu}_j)$, with $C = \frac{1}{4}\left( \boldsymbol{\mu}_i^\top \boldsymbol{\Sigma}_i^{-1}\boldsymbol{\mu}_i + \boldsymbol{\mu}_j^\top \boldsymbol{\Sigma}_j^{-1}\boldsymbol{\mu}_j - 2\tilde{\boldsymbol{\mu}}^\top \tilde{\boldsymbol{\Sigma}}^{-1}\tilde{\boldsymbol{\mu}} \right)$.

Integration yields:

$$\rho(p_i, p_j) = |\boldsymbol{\Sigma}_i|^{-1/4} |\boldsymbol{\Sigma}_j|^{-1/4} e^{-C} |\tilde{\boldsymbol{\Sigma}}|^{1/2}.$$

Using standard matrix identities, we have:

$$C = \tfrac{1}{8}(\boldsymbol{\mu}_i - \boldsymbol{\mu}_j)^\top \left( \tfrac{\boldsymbol{\Sigma}_i + \boldsymbol{\Sigma}_j}{2} \right)^{-1} (\boldsymbol{\mu}_i - \boldsymbol{\mu}_j), \quad |\tilde{\boldsymbol{\Sigma}}|^{1/2} |\boldsymbol{\Sigma}_i|^{-1/4} |\boldsymbol{\Sigma}_j|^{-1/4} = exp\left( -\tfrac{1}{2} \log \frac{\left| \frac{\boldsymbol{\Sigma}_i + \boldsymbol{\Sigma}_j}{2} \right|}{\sqrt{|\boldsymbol{\Sigma}_i||\boldsymbol{\Sigma}_j|}} \right).$$

Substituting them into $\rho(p_i, p_j)$ gives:

$$\rho(p_i, p_j) = \exp\left( -\tfrac{1}{8}(\boldsymbol{\mu}_i - \boldsymbol{\mu}_j)^\top \left( \tfrac{\boldsymbol{\Sigma}_i + \boldsymbol{\Sigma}_j}{2} \right)^{-1} (\boldsymbol{\mu}_i - \boldsymbol{\mu}_j) - \tfrac{1}{2} \log \frac{\left| \frac{\boldsymbol{\Sigma}_i + \boldsymbol{\Sigma}_j}{2} \right|}{\sqrt{|\boldsymbol{\Sigma}_i||\boldsymbol{\Sigma}_j|}} \right) = \exp(-B_{ij}).$$

Therefore, we proved:

$$P_{\text{err}}^{(2)}(i, j) \leq \sqrt{\pi_i \pi_j}\, \rho(p_i, p_j) = \sqrt{\pi_i \pi_j}\, \exp\left( -B_{ij} \right).$$

### B.3 PROOF OF THEOREM 1

***Pairwise Error.*** For a given input $\boldsymbol{x}$ and $label(x) = t_i$, define the pairwise overtake events as $A_{ij} = \{p_j(\boldsymbol{z}) \geq p_i(\boldsymbol{z})\}, \quad j \neq i$. For $A_{ij}$, we have:

$$\Pr(A_{ij} \mid label(x) = t_i) = \int_{\{p_j(\boldsymbol{z}) \geq p_i(\boldsymbol{z})\}} p_i(\boldsymbol{z})\, d\boldsymbol{z} = \int_{\{p_j(\boldsymbol{z}) \geq p_i(\boldsymbol{z})\}} \min\{p_i(\boldsymbol{z}), p_j(\boldsymbol{z})\} d\boldsymbol{z} \quad (7)$$

$$\leq \int \min\{p_i, p_j\}\, d\boldsymbol{z} \leq \int \sqrt{p_i p_j}\, dx = \rho(p_i, p_j) = \exp(-B_{ij}),$$

where the first equality follows from the definition of the error event: under class $i$, misclassification occurs precisely when $p_j(\boldsymbol{z}) \geq p_i(\boldsymbol{z})$; the second equality holds because, on the region $\{p_j(\boldsymbol{z}) \geq p_i(\boldsymbol{z})\}$, we have $\min\{p_i(\boldsymbol{z}), p_j(\boldsymbol{z})\} = p_i(\boldsymbol{z})$; the inequality is obtained by extending the domain of integration; and the last two equality uses the Bhattacharyya coefficient, as established in Lemma 1.

***Top-$k$ Error.*** Let $N_1 = \sum_{j \neq i} \mathbf{1}_{A_{ij}}$ denote the number of rivals that beat $i$. Then the Top-$k$ error event under $label(\boldsymbol{x}) = i$ is $\{N_1 \geq k\}$. Now, we have the following analysis:

$$\Pr(N_1 \geq k \mid label(x) = t_i) \leq \frac{\mathbb{E}[N_1 \mid label(x) = t_i]}{k}$$

$$= \frac{1}{k} \sum_{j \neq i} \Pr(A_{ij} \mid label(x) = t_i)$$

$$\leq \frac{1}{k} \sum_{j \neq i} exp(-B_{i,j}),$$

where the first inequality applies the Markov's inequality; the equality follows from computing $\mathbb{E}[N_1]$ and substituting the pairwise error terms; and the final inequality then uses the bound in Eq. (7).

### B.4 PROOF OF THEOREM 2

For any competitor $j \neq i^\star$, consider the event $p_j(\boldsymbol{z}) \geq p_{i^\star}(\boldsymbol{z})$. For any $\alpha \in (0,1]$, the Markov-Chernoff technique gives:

$$\Pr_{\boldsymbol{z} \sim q}\big(p_j(\boldsymbol{z}) \geq p_{i^\star}(\boldsymbol{z})\big) = \Pr\left(\left(\tfrac{p_j(\boldsymbol{z})}{p_{i^\star}(\boldsymbol{z})}\right)^\alpha \geq 1\right) \leq \mathbb{E}_q\left[\left(\tfrac{p_j(\boldsymbol{z})}{p_{i^\star}(\boldsymbol{z})}\right)^\alpha\right],$$

where The first equality holds because, for any $\alpha > 0$, the event $\{p_j \geq p_{i^\star}\}$ is equivalent to $\left\{\left(\tfrac{p_j}{p_{i^\star}}\right)^\alpha \geq 1\right\}$; and the first inequality then follows from Markov's inequality: if a random variable $X \geq 0$, then for any $t > 0$, $\Pr(X \geq t) \leq \tfrac{\mathbb{E}[X]}{t}$.

**Lemma 3** *Let $q = \mathcal{N}(\boldsymbol{\mu}_q, \boldsymbol{\Sigma}_q)$, $p_j = \mathcal{N}(\boldsymbol{\mu}_j, \boldsymbol{\Sigma}_j)$, and $p_{i^\star} = \mathcal{N}(\boldsymbol{\mu}_{i^\star}, \boldsymbol{\Sigma}_{i^\star})$ be full-rank d-variate Gaussians. For any $\alpha \in (0,1]$, assume $M_\alpha^j := \boldsymbol{\Sigma}_q^{-1} + \alpha \boldsymbol{\Sigma}_j^{-1} - \alpha \boldsymbol{\Sigma}_{i^\star}^{-1} \succ 0$.*

*Then the $\alpha$-moment of the likelihood ratio admits the closed form as follows:*

$$\mathbb{E}_{\boldsymbol{z} \sim q}\left[\left(\tfrac{p_j(\boldsymbol{z})}{p_{i^\star}(\boldsymbol{z})}\right)^\alpha\right] = C_\alpha^j \, |M_\alpha^j|^{-1/2} \exp\big(\tfrac{1}{2}(h_\alpha^j)^\top (M_\alpha^j)^{-1} h_\alpha^j - K_\alpha^j\big),$$

*where $h_\alpha^j = \boldsymbol{\Sigma}_q^{-1} \boldsymbol{\mu}_q + \alpha \boldsymbol{\Sigma}_j^{-1} \boldsymbol{\mu}_j - \alpha \boldsymbol{\Sigma}_{i^\star}^{-1} \boldsymbol{\mu}_{i^\star}$, $K_\alpha^j = \tfrac{1}{2}\boldsymbol{\mu}_q^\top \boldsymbol{\Sigma}_q^{-1} \boldsymbol{\mu}_q + \tfrac{\alpha}{2}\big(\boldsymbol{\mu}_j^\top \boldsymbol{\Sigma}_j^{-1} \boldsymbol{\mu}_j - \boldsymbol{\mu}_{i^\star}^\top \boldsymbol{\Sigma}_{i^\star}^{-1} \boldsymbol{\mu}_{i^\star}\big)$, $C_\alpha^j = \exp\big(-\tfrac{\alpha}{2}\log|\boldsymbol{\Sigma}_j| + \tfrac{\alpha}{2}\log|\boldsymbol{\Sigma}_{i^\star}| - \tfrac{1}{2}\log|\boldsymbol{\Sigma}_q|\big)$.*

**Proof.** By Chernoff/Markov's trick (see (Chernoff, 1952)), we have:

$$\mathbb{E}_q\left[\left(\tfrac{p_j(\boldsymbol{z})}{p_{i^\star}(\boldsymbol{z})}\right)^\alpha\right] = \int_{\mathbb{R}^d} q(\boldsymbol{z}) \exp\Big(\alpha\big(\log p_j(\boldsymbol{z}) - \log p_{i^\star}(\boldsymbol{z})\big)\Big) d\boldsymbol{z}.$$

Write each log-density in quadratic form: $\log p(\boldsymbol{z}) = -\tfrac{d}{2}\log(2\pi) - \tfrac{1}{2}\log|\boldsymbol{\Sigma}| - \tfrac{1}{2}(\boldsymbol{z}-\boldsymbol{\mu})^\top \boldsymbol{\Sigma}^{-1}(\boldsymbol{z}-\boldsymbol{\mu})$. Collecting the constant (determinant) terms yields the prefactor $C_\alpha^{(q)}$. Collecting the quadratic and linear terms in $\boldsymbol{z}$ gives:

$$-\tfrac{1}{2}\boldsymbol{z}^\top M_\alpha \boldsymbol{z} + h_\alpha^\top \boldsymbol{z} - K_\alpha, \quad M_\alpha = \boldsymbol{\Sigma}_q^{-1} + \alpha \boldsymbol{\Sigma}_j^{-1} - \alpha \boldsymbol{\Sigma}_{i^\star}^{-1},$$

with $h_\alpha, K_\alpha$ as stated. Completing the square and using the multivariate Gaussian integral $\int \exp(-\tfrac{1}{2}\boldsymbol{z}^\top A \boldsymbol{z} + b^\top \boldsymbol{z}) d\boldsymbol{z} = (2\pi)^{d/2}|A|^{-1/2}\exp(\tfrac{1}{2}b^\top A^{-1}b)$ (valid for $A \succ 0$), and noticing that $(2\pi)^{d/2}$ cancels with the corresponding factor in $q$, we obtain the claimed closed form.

If $q = p_{i^\star}$ (i.e., $\boldsymbol{\mu}_q = \boldsymbol{\mu}_{i^\star}$ and $\boldsymbol{\Sigma}_q = \boldsymbol{\Sigma}_{i^\star}$), then we have:

$$\mathbb{E}_{\boldsymbol{z} \sim p_{i^\star}}\left[\left(\tfrac{p_j(\boldsymbol{z})}{p_{i^\star}(\boldsymbol{z})}\right)^\alpha\right] = \int p_{i^\star}(\boldsymbol{z})^{1-\alpha} p_j(\boldsymbol{z})^\alpha \, d\boldsymbol{z} = \rho_\alpha\big(p_{i^\star}, p_j\big),$$

where the right-hand side is the standard multivariate Gaussian Chernoff $\alpha$-coefficient:

$$\rho_\alpha(p_{i^\star}, p_j) = \frac{|\boldsymbol{\Sigma}_j|^{\alpha/2}\,|\boldsymbol{\Sigma}_{i^\star}|^{(1-\alpha)/2}}{\big|\alpha\boldsymbol{\Sigma}_j + (1-\alpha)\boldsymbol{\Sigma}_{i^\star}\big|^{1/2}} \exp\left(-\frac{\alpha(1-\alpha)}{2}(\boldsymbol{\mu}_j - \boldsymbol{\mu}_{i^\star})^\top\big(\alpha\boldsymbol{\Sigma}_j + (1-\alpha)\boldsymbol{\Sigma}_{i^\star}\big)^{-1}(\boldsymbol{\mu}_j - \boldsymbol{\mu}_{i^\star})\right),$$

as given in Nielsen (2014, Eq. (35)).

Let $N_2 = \sum_{j \neq i^\star} \mathbf{1}_{p_j \geq p_{i^\star}}$. denote the number of rivals that beat $i^\star$. Then the Top-$k$ error event under $\boldsymbol{z} \sim q$ is $\{N_q \geq k\}$. Similar to the proof of Theorem 1, we have:

$$
\begin{aligned}
\Pr(N_2 \geq k \mid \boldsymbol{z} \sim q) &\leq \frac{\mathbb{E}[N_2 \mid \boldsymbol{z} \sim q]}{k} \\
&= \frac{1}{k}\sum_{j \neq i^\star} \Pr(p_j(\boldsymbol{z}) \geq p_{i^\star}(\boldsymbol{z}) \mid \boldsymbol{z} \sim q]) \\
&\leq \frac{1}{k}\sum_{j \neq i^\star} \mathbb{E}_q\left[\left(\tfrac{p_j(\boldsymbol{z})}{p_{i^\star}(\boldsymbol{z})}\right)^\alpha\right], \\
&= \frac{1}{k}\sum_{j \neq i^\star} C_\alpha^j \, |M_\alpha^j|^{-1/2}\exp\big(\tfrac{1}{2}(h_\alpha^j)^\top (M_\alpha^j)^{-1} h_\alpha^j - K_\alpha^j\big)
\end{aligned}
$$

# C  EXPERIMENTAL SUPPLEMENTARY MATERIAL

## C.1  DETAILS OF EVALUATION DATASETS AND METRICS

We employ a subset of the FLAN-v2 datasets (Wei et al., 2022) for domain generation. FLAN-v2 datasets is a large-scale instruction-tuning corpus that integrates diverse Natural Language Understanding (NLU) and Natural Language Generation (NLG) tasks into an instruction–response format. A detailed summary of the selected datasets together with their associated evaluation metrics is provided below.

***Natural Language Inference.*** Natural language inference tasks require models to determine logical relations (entailment, contradiction, or neutrality) between pairs of sentences. We use the following datasets: (1) ANLI (v1-v3); (2) CB; (3) MNLI (matched, mismatched); (4) QNLI; (5) RTE; (6) SNLI; (7) WNLI. All datasets in this cluster are evaluated using accuracy as the metric.

***Question Answering.*** Question answering tasks evaluate the ability to retrieve or generate correct answers from passages or knowledge bases. We use the following datasets: (1) ARC (Challenge, Easy); (2) BoolQ; (3) MultiRC; (4) NaturalQuestions; (5) OpenBookQA; (6) ReCoRD; (7) SQuAD (v1-v2); (8) TriviaQA. For ARC, BoolQ, OpenBookQA, and ReCoRD, accuracy is used as the evaluation metric. For the remaining datasets, both accuracy and F1 score are reported.

***Sentiment Analysis.*** Sentiment analysis tasks involve classifying the polarity or emotional tone of text, such as positive or negative sentiment. We use the following datasets: (1) Sentiment140; (2) SST2. For Sentiment140, both accuracy and F1 score are reported, while SST2 is evaluated using accuracy only.

***Translation.*** Translation tasks test the capacity to generate fluent and semantically correct text across different languages. We use the following datasets: (1) ParaCrawl_EnEs; (2) WMT14_EnFr; (3) WMT16_CsEn; (4) WMT16_DeEn; (5) WMT16_FiEn; (6) WMT16_RoEn; (7) WMT16_RuEn; (8) WMT16_TrEn. All translation tasks are evaluated using BLEU, which measures $n$-gram overlap between system outputs and reference translations.

***Commonsense Reasoning.*** Commonsense reasoning tasks require leveraging everyday knowledge and logical inference to choose or generate plausible answers. We use the following datasets: (1) COPA; (2) HellaSwag; (3) PIQA; (4) StoryCloze. All datasets in this cluster are evaluated using accuracy as the metric.

***Paraphrase.*** Paraphrase tasks assess whether two sentences express the same underlying meaning, despite differences in wording. We use the following datasets: (1) GLUE_MRPC; (2) GLUE_QQP; (3) STS-B; (4) PAWS_Wiki. MRPC and QQP are evaluated with both accuracy and F1, while STS-B and PAWS_Wiki are evaluated using accuracy.

***Struct-to-Text Generation.*** These tasks focus on converting structured data, such as triples or tables, into coherent natural language text. We use the following datasets: (1) CommonGen; (2) DART; (3) E2E_NLG; (4) Gigaword; (5) WebNLG_En. All datasets in this cluster are evaluated using ROUGE (ROUGE-1,2,L) and BLEU, since $n$-gram overlap captures the informativeness and fluency of generated text.

***Coreference Resolution.*** Coreference tasks require identifying expressions in text that refer to the same entity. We use the following datasets: (1) Definite Pronoun Resolution; (2) WSC. Both datasets are evaluated using accuracy.

***Text Correction.*** Text correction tasks involve detecting and fixing grammatical errors or inconsistencies in sentences. We use the following datasets: (1) CoLA; (2) FixPunct; (3) TrueCase. All datasets in this cluster are evaluated using accuracy.

***Word-level Tasks.*** Word-level tasks examine lexical semantics and basic text processing such as contextual meaning and segmentation. We use the following datasets: (1) WiC; (2) Word_Segment. WiC is evaluated using accuracy, while Word_Segment is evaluated using both accuracy and F1.

Accuracy is sufficient when tasks have clear-cut, single-label predictions, such as classification or multiple-choice settings, where each prediction is either entirely correct or incorrect. In contrast, tasks with span-based, multi-label, or imbalanced data distributions may yield partially correct pre-

dictions. In these cases, F1 score is reported alongside accuracy, as it balances precision and recall and provides a more sensitive evaluation of partial correctness.

## C.2 EXTENDED VISUALIZATION OF LORA PROJECTIONS

In the main text, we reported scatter plots of the first two principal components obtained from vectors in LoRA projection matrices fine-tuned on five NLI tasks, focusing on two representative layers. To further validate these findings, Fig. 7 presents results from additional layers, which reveal consistent structural patterns across model depth. We additionally extend the analysis to tasks drawn from different clusters (As shown in Fig. 8), where similar trends are observed. Taken together, these results provide stronger empirical support for the key observations discussed in Sec. 3.1.

Furthermore, Fig. 9–12 provide a comparison between the cosine similarity distributions of LoRA's down-projection matrices and up-projection matrices under within-cluster and cross-cluster settings. For down-projection matrices, the within-cluster and cross-cluster distributions are nearly identical: both are centered at zero with indistinguishable variance and tail mass. This suggests that the down-projection matrices behave almost like random projections and do not encode task-specific information. In contrast, the similarity distributions of up-projection matrices are noticeably more concentrated. Under the within-cluster setting, the up-projection matrices exhibit higher cosine similarity than in the cross-cluster setting, indicating that the up-projection matrices share more structure within the same cluster. This further confirms the distinct functional roles of the down- and up-projection matrices: down-projection matrices behave closer to task-agnostic random projection, whereas up-projection matrices capture meaningful cluster-level structure.

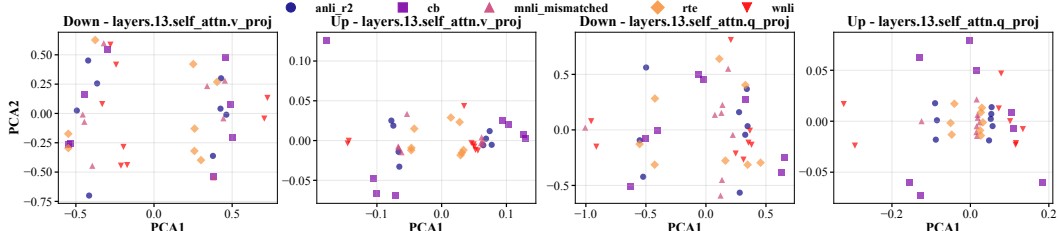

Figure 7: Scatter plots of the first two principal components derived from vectors in LoRA query and value projection matrices in layer 13 across five NLI tasks.

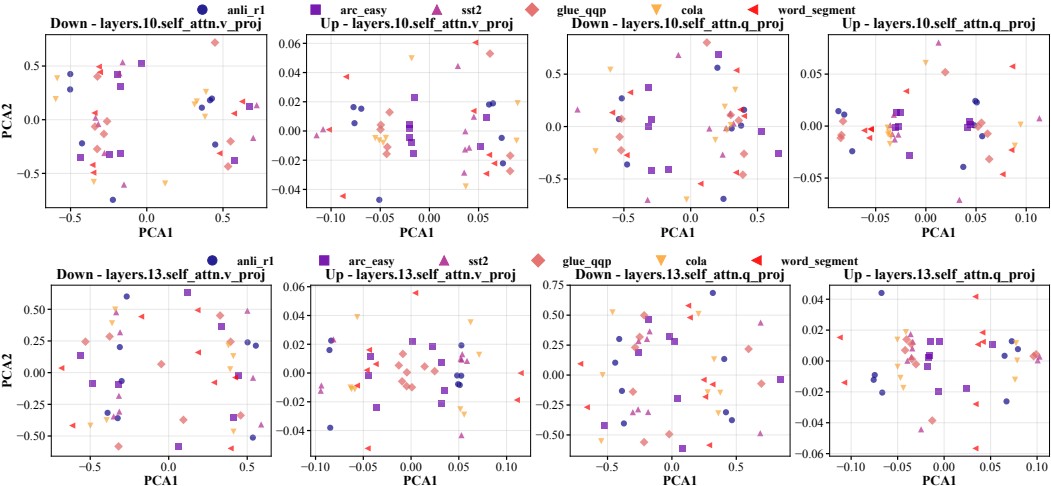

Figure 8: Scatter plots of the first two principal components derived from vectors in LoRA query and value projection matrices in layers 10 and 13 across six tasks from different clusters.

## C.3 FULL EXPERIMENTAL RESULTS

Tab. 4 reports the per-task accuracy of different methods on the NLI cluster using LLaMA2-13B and FLAN-T5-large. Comprehensive performance under the unseen cluster setting is reported for all tasks, including detailed metrics for each task and evaluation criterion. Results for LLaMA2-7B and

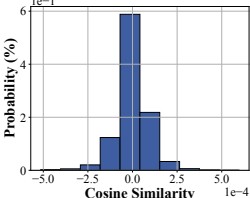
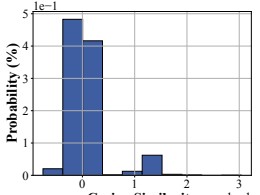
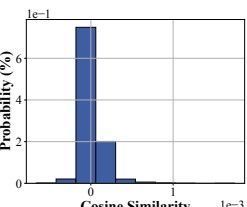

Figure 9: Cosine similarity distribution of LoRA down-projection matrices across NLI tasks.

Figure 10: Cosine similarity distribution of LoRA up-projection matrices across NLI tasks.

Figure 11: Cosine similarity distribution of LoRA down-projection matrices across NLI and QA tasks.

Figure 12: Cosine similarity distribution of LoRA up-projection matrices across NLI and QA tasks.

LLaMA2-13B are shown in Tab. 5 and Tab. 6, and results for FLAN-T5-large are shown in Tab. 7. Fig. 13 shows a heatmap of cosine similarities produced by `Retriever` across tasks, with tasks from the same cluster grouped by green boxes. Tab. 8 reports the per-batch input-mapping latency and inference latency across all NLI tasks under different LoRA-pool sizes. Tab. 9 summarizes the performance of `HiLoRA` under different settings of the scaling factor $\gamma$ across various task types.

Table 4: Detailed performance on the NLI cluster using LLaMA2-13B and FLAN-T5-large. Tasks with a white background are set as *seen* tasks, while those with a gray background are set as *unseen* tasks. For each task, the best accuracy among all methods is in **bold**, and the second best is underlined.

| Methods | LoRA | HiLoRA | HiLoRA-GS | HiLoRA-ROC | Retriever | LEGO | Arrow | Phatgoose | Ensemble | Merged |
|---|---|---|---|---|---|---|---|---|---|---|
| *LLaMA2-13B* | | | | | | | | | | |
| ANLI_r1 | 60.30 | **62.70** | 60.60 | 58.80 | 61.80 | 58.30 | 47.40 | 55.60 | 52.00 | 24.70 |
| ANLI_r2 | 47.30 | **48.60** | 47.00 | 46.20 | 48.50 | 44.3 | 041.00 | 43.70 | 41.40 | 24.30 |
| ANLI_r3 | 49.92 | 48.92 | 47.75 | **49.08** | 49.75 | 47.17 | 44.92 | 47.33 | 46.42 | 26.42 |
| CB | 88.00 | **86.00** | 84.00 | 80.00 | 82.00 | 84.00 | 80.00 | 76.00 | 84.00 | 38.00 |
| MNLI | 87.97 | **87.58** | 87.34 | 82.58 | 86.56 | 85.47 | 70.00 | 76.45 | 75.35 | 37.30 |
| MNLI_mis | 89.80 | **89.49** | 88.63 | 83.79 | 88.87 | 85.70 | 71.02 | 75.94 | 76.41 | 36.64 |
| QNLI | 82.66 | **83.13** | 82.66 | 56.29 | 77.89 | 68.95 | 67.97 | 70.98 | 69.96 | 44.73 |
| RTE | 80.74 | 75.19 | 75.56 | 73.33 | 74.81 | 72.59 | **78.89** | 73.33 | 76.30 | 52.59 |
| SNLI | 81.91 | 81.84 | **81.99** | 76.72 | 81.60 | 79.65 | 67.19 | 75.23 | 72.38 | 32.46 |
| WNLI | 71.43 | **67.14** | **67.14** | **67.14** | 51.43 | 64.29 | 62.86 | 58.57 | 65.71 | 42.8 |
| AVG | 74.00 | **72.86** | 72.47 | 67.39 | 70.32 | 69.04 | 63.12 | 65.31 | 65.99 | 36.00 |
| *FLAN-T5-Large* | | | | | | | | | | |
| ANLI-r1 | 60.20 | 60.40 | 61.20 | 57.90 | 60.70 | 60.60 | 60.80 | **61.30** | 60.70 | 60.80 |
| ANLI-r2 | 43.30 | 42.20 | 42.70 | 41.00 | 42.80 | 42.50 | **43.70** | 42.90 | 43.50 | 43.40 |
| ANLI-r3 | 44.50 | 43.08 | 44.42 | 42.42 | 44.25 | 44.17 | **45.25** | 44.67 | 44.33 | 44.33 |
| CB | 78.00 | 78.00 | 78.00 | 78.00 | 78.00 | 78.00 | **80.00** | 78.00 | 78.00 | 78.00 |
| MNLI | 88.59 | **89.12** | 84.77 | 89.00 | 88.16 | 64.69 | 61.76 | 66.52 | 63.52 | 58.87 |
| MNLI-mis | 89.14 | **88.91** | 82.07 | 88.52 | 88.67 | 62.93 | 68.79 | 64.10 | 61.05 | 56.76 |
| QNLI | 82.54 | **82.70** | 82.54 | **82.70** | 82.38 | 81.95 | 81.02 | 82.38 | 82.38 | 80.66 |
| RTE | 78.89 | 61.15 | 61.11 | 61.48 | 62.59 | 62.96 | **63.70** | 60.00 | **63.70** | 57.04 |
| SNLI | 60.08 | **80.00** | 60.31 | 80.00 | 68.44 | 13.75 | 17.85 | 10.14 | 13.28 | 6.13 |
| WNLI | 52.86 | 51.43 | 51.43 | 44.29 | 51.29 | 50.00 | **54.29** | 44.29 | 51.43 | 44.29 |
| Avg | 67.81 | **67.70** | 64.85 | 66.53 | 66.76 | 56.20 | 57.81 | 55.29 | 56.19 | 53.03 |

## C.4 Synthetic Sample Generation

In addition to using real training samples, we also evaluate a setting where the sequence-level routing in `HiLoRA` is based on synthetic task examples generated by a large language model (LLM). The goal is to approximate the instruction format and semantic characteristics of each task without accessing its original training data.

For each task (e.g., QNLI), we prompt the LLM to generate $m = 20$ synthetic examples that follow the same input style as the corresponding FLAN instruction, while avoiding any direct use of the original dataset. The prompt we use is conceptually as follows:

" Following the format below, generate 20 synthetic samples for the QNLI task *without* relying on the original dataset. Each sample should be provided as a JSON object with a single `"inputs"` field that contains the full instruction-style text. The overall JSON structure should be:

Table 5: Per-task performance of LLaMA2-7B under the cross-cluster setting.

| Tasks/Methods | Metric | LoRA | HiLoRA | HiLoRA-GS | HiLoRA-ROC | Retriever | LEGO | Arrow | Phatgoose | Ensemble | Merged |
|---|---|---|---|---|---|---|---|---|---|---|---|
| ANLI_r1 | ACC | 46.40 | 30.70 | 27.30 | 27.70 | 30.00 | 28.20 | 31.30 | 29.40 | 29.90 | 21.40 |
| ANLI_r2 | ACC | 40.10 | 34.50 | 32.50 | 30.20 | 34.30 | 30.30 | 32.20 | 31.50 | 33.00 | 22.00 |
| ANLI_r3 | ACC | 36.92 | 31.67 | 30.42 | 30.67 | 31.33 | 29.42 | 30.58 | 30.08 | 29.75 | 16.17 |
| CB | ACC | 80.00 | 70.00 | 62.00 | 72.00 | 66.00 | 60.00 | 52.00 | 64.00 | 64.00 | 4.00 |
| MNLI | ACC | 77.66 | 50.74 | 49.65 | 49.92 | 48.95 | 49.34 | 45.12 | 47.85 | 45.35 | 0.94 |
| MNLI_mis | ACC | 79.69 | 51.29 | 49.45 | 50.59 | 49.45 | 50.82 | 45.66 | 48.05 | 46.37 | 1.21 |
| QNLI | ACC | 77.27 | 46.84 | 42.62 | 46.56 | 42.23 | 41.41 | 47.97 | 44.80 | 47.19 | 8.59 |
| RTE | ACC | 52.96 | 62.22 | 57.41 | 52.96 | 72.59 | 50.74 | 58.15 | 55.93 | 56.30 | 39.26 |
| SNLI | ACC | 67.42 | 40.31 | 41.09 | 40.23 | 15.82 | 40.08 | 34.18 | 39.06 | 36.68 | 0.51 |
| WNLI | ACC | 72.86 | 47.14 | 49.86 | 49.14 | 47.14 | 48.57 | 45.71 | 47.14 | 47.14 | 2.86 |
| AVG_NLI | | 63.13 | 46.54 | 44.23 | 45.00 | 43.78 | 42.89 | 42.29 | 43.78 | 43.57 | 11.69 |
| ARC_C | ACC | 40.43 | 34.74 | 30.78 | 30.43 | 34.31 | 32.33 | 30.00 | 32.76 | 31.90 | 0.43 |
| ARC_E | ACC | 40.17 | 48.64 | 44.28 | 45.89 | 47.50 | 46.74 | 41.57 | 46.06 | 45.42 | 0.55 |
| Bool_Q | ACC | 85.23 | 78.36 | 75.86 | 64.34 | 77.27 | 75.43 | 64.45 | 76.41 | 76.09 | 21.21 |
| MultiRC | ACC | 62.15 | 36.48 | 34.49 | 40.82 | 33.36 | 39.57 | 32.58 | 34.41 | 35.08 | 3.52 |
| | F1 | 65.86 | 38.21 | 36.10 | 42.99 | 34.93 | 41.28 | 34.94 | 36.58 | 36.84 | 13.15 |
| NaturalQuestions | ACC | 18.95 | 12.03 | 9.38 | 5.90 | 8.59 | 13.40 | 9.88 | 13.48 | 11.80 | 0.35 |
| | F1 | 29.82 | 20.39 | 18.45 | 11.33 | 17.13 | 21.42 | 17.82 | 21.37 | 19.65 | 9.13 |
| OpenBookQA | ACC | 58.20 | 45.60 | 44.60 | 45.40 | 38.60 | 45.80 | 43.00 | 42.80 | 45.20 | 0.20 |
| RecoRD | ACC | 92.87 | 69.81 | 63.51 | 66.45 | 72.30 | 68.19 | 66.26 | 65.18 | 62.78 | 35.42 |
| SQuAD_v1 | ACC | 55.20 | 44.65 | 40.90 | 39.80 | 40.35 | 45.78 | 25.51 | 43.40 | 42.97 | 2.42 |
| | F1 | 74.91 | 64.25 | 59.47 | 58.99 | 60.01 | 63.75 | 41.89 | 61.98 | 62.36 | 21.43 |
| SQuAD_v2 | ACC | 64.34 | 26.45 | 24.57 | 24.41 | 19.26 | 26.48 | 14.77 | 24.06 | 25.78 | 0.35 |
| | F1 | 73.80 | 36.03 | 34.01 | 35.56 | 30.42 | 36.51 | 23.35 | 33.99 | 35.16 | 9.06 |
| TriviaQA | ACC | 54.02 | 47.19 | 41.13 | 43.95 | 37.27 | 49.18 | 42.77 | 47.46 | 46.99 | 3.91 |
| | F1 | 60.27 | 58.96 | 54.73 | 54.95 | 49.81 | 59.14 | 53.35 | 58.79 | 58.33 | 22.88 |
| AVG_QA | | 59.66 | 46.95 | 43.56 | 43.19 | 43.55 | 46.67 | 39.37 | 45.10 | 44.89 | 10.09 |
| Sentiment140 | ACC | 43.06 | 43.27 | 40.61 | 44.49 | 39.59 | 42.73 | 31.84 | 42.22 | 40.41 | 4.49 |
| | F1 | 44.70 | 44.26 | 41.91 | 45.51 | 41.34 | 43.68 | 33.05 | 43.67 | 41.08 | 12.05 |
| SST2 | ACC | 75.86 | 63.10 | 58.51 | 62.99 | 59.77 | 62.64 | 49.08 | 63.06 | 59.77 | 0.11 |
| AVG_Sentiment | | 59.87 | 54.43 | 49.88 | 54.00 | 50.12 | 52.93 | 40.76 | 53.00 | 50.26 | 4.19 |
| ParaCrawl_EnEs | BLUE | 29.05 | 27.18 | 28.61 | 18.15 | 15.25 | 19.31 | 27.01 | 26.11 | 25.34 | 11.12 |
| WMT14_EnFr | BLUE | 30.49 | 30.23 | 30.95 | 23.20 | 12.57 | 25.79 | 29.67 | 29.83 | 29.91 | 15.64 |
| WMT16_CsEn | BLUE | 19.67 | 18.56 | 19.37 | 12.55 | 7.52 | 14.10 | 18.26 | 18.04 | 18.29 | 11.76 |
| WMT16_DeEn | BLUE | 26.72 | 27.26 | 27.40 | 20.06 | 11.22 | 21.67 | 26.57 | 26.19 | 26.61 | 16.12 |
| WMT16_FiEn | BLUE | 14.58 | 14.63 | 15.36 | 10.31 | 5.40 | 11.20 | 14.58 | 14.23 | 14.50 | 9.04 |
| WMT16_RoEn | BLUE | 24.91 | 22.80 | 22.87 | 16.19 | 12.01 | 17.57 | 22.47 | 22.13 | 22.33 | 13.72 |
| WMT16_RuEn | BLUE | 22.27 | 17.84 | 21.54 | 13.33 | 10.08 | 16.82 | 20.99 | 19.46 | 21.31 | 12.50 |
| WMT16_TrEn | BLUE | 8.11 | 7.74 | 8.33 | 5.54 | 1.91 | 5.12 | 7.93 | 7.79 | 7.88 | 5.39 |
| AVG_Translation | | 21.98 | 20.78 | 21.80 | 14.92 | 9.50 | 16.45 | 20.93 | 20.47 | 20.77 | 11.91 |
| COPA | ACC | 72.00 | 65.00 | 66.00 | 66.00 | 71.00 | 68.00 | 59.00 | 62.00 | 63.00 | 20.00 |
| HellaSwag | ACC | 71.76 | 28.87 | 22.42 | 26.41 | 22.85 | 23.55 | 28.98 | 23.91 | 26.99 | 0.00 |
| PIQA | ACC | 61.75 | 53.72 | 50.98 | 48.58 | 53.22 | 47.70 | 51.75 | 52.90 | 51.64 | 1.80 |
| StoryCloze | ACC | 62.94 | 63.48 | 61.66 | 64.17 | 32.89 | 61.28 | 63.58 | 64.71 | 66.47 | 39.14 |
| AVG_Commonsense | | 67.11 | 52.76 | 50.27 | 51.29 | 44.99 | 50.14 | 50.83 | 50.88 | 52.03 | 15.24 |
| GLUE_MRPC | ACC | 68.00 | 65.25 | 65.75 | 53.50 | 58.50 | 36.75 | 53.25 | 59.50 | 58.75 | 17.50 |
| | F1 | 68.00 | 65.25 | 65.75 | 53.50 | 58.50 | 36.75 | 53.25 | 59.50 | 58.75 | 19.50 |
| GLUE_QQP | ACC | 76.13 | 64.80 | 67.66 | 53.44 | 68.12 | 58.71 | 63.09 | 56.80 | 64.73 | 2.23 |
| | F1 | 76.13 | 64.80 | 67.66 | 53.44 | 68.12 | 58.71 | 63.09 | 56.80 | 64.73 | 3.01 |
| STSB | ACC | 34.82 | 17.83 | 20.33 | 16.78 | 19.78 | 15.95 | 17.27 | 16.78 | 16.43 | 0.21 |
| PAWS_Wiki | ACC | 88.55 | 64.45 | 46.68 | 47.19 | 71.64 | 48.24 | 46.76 | 56.17 | 56.33 | 9.10 |
| AVG_Paraphrase | | 66.88 | 53.08 | 50.11 | 42.73 | 54.51 | 39.91 | 45.09 | 47.31 | 49.06 | 7.61 |
| CommonGen | ROUGE-1 | 54.60 | 43.27 | 44.89 | 43.08 | 43.03 | 24.14 | 39.70 | 38.95 | 37.43 | 35.44 |
| | ROUGE-2 | 23.18 | 14.05 | 13.88 | 2.21 | 8.65 | 1.47 | 10.48 | 12.26 | 11.26 | 9.68 |
| | ROUGE-L | 47.82 | 36.84 | 36.30 | 28.31 | 32.51 | 21.70 | 32.57 | 33.79 | 32.47 | 27.93 |
| | BLEU | 11.95 | 6.74 | 7.21 | 0.27 | 3.90 | 0.08 | 4.20 | 5.60 | 5.08 | 3.71 |
| DART | ROUGE-1 | 72.16 | 52.74 | 52.89 | 51.78 | 53.03 | 35.26 | 49.97 | 50.33 | 50.61 | 42.92 |
| | ROUGE-2 | 47.78 | 25.66 | 25.48 | 23.79 | 24.49 | 16.03 | 26.08 | 26.08 | 26.19 | 21.06 |
| | ROUGE-L | 56.07 | 39.76 | 39.68 | 38.74 | 39.44 | 28.32 | 39.27 | 39.19 | 39.45 | 33.72 |
| | BLEU | 36.44 | 14.99 | 14.60 | 13.19 | 13.20 | 6.39 | 14.55 | 15.74 | 15.26 | 10.43 |
| E2E_NLG | ROUGE-1 | 73.04 | 56.60 | 58.55 | 47.01 | 51.66 | 18.73 | 59.00 | 57.95 | 56.15 | 54.62 |
| | ROUGE-2 | 44.84 | 29.79 | 29.29 | 21.20 | 27.65 | 9.24 | 33.25 | 32.76 | 31.78 | 29.99 |
| | ROUGE-L | 52.67 | 41.28 | 41.45 | 33.49 | 39.45 | 16.72 | 43.66 | 43.81 | 42.31 | 40.50 |
| | BLEU | 31.79 | 17.99 | 16.56 | 7.75 | 13.89 | 0.28 | 21.17 | 18.94 | 18.24 | 20.17 |
| Gigaword | ROUGE-1 | 36.28 | 25.28 | 25.24 | 22.26 | 25.18 | 24.64 | 25.43 | 26.50 | 26.31 | 22.92 |
| | ROUGE-2 | 16.07 | 8.58 | 8.57 | 7.46 | 8.31 | 8.27 | 8.64 | 9.50 | 9.55 | 7.76 |
| | ROUGE-L | 32.90 | 21.67 | 21.70 | 19.11 | 21.63 | 21.21 | 21.90 | 22.90 | 22.90 | 19.54 |
| | BLEU | 10.32 | 3.30 | 3.37 | 2.82 | 3.08 | 3.42 | 3.37 | 4.09 | 4.03 | 3.19 |
| WebNLG_En | ROUGE-1 | 78.37 | 48.01 | 48.29 | 52.02 | 53.28 | 32.22 | 45.27 | 45.01 | 42.99 | 43.94 |
| | ROUGE-2 | 56.19 | 24.95 | 25.53 | 25.93 | 26.66 | 15.57 | 24.09 | 24.60 | 22.90 | 23.76 |
| | ROUGE-L | 62.52 | 39.14 | 39.16 | 42.23 | 42.80 | 28.15 | 37.65 | 37.52 | 35.71 | 35.89 |
| | BLEU | 45.18 | 15.60 | 15.02 | 14.60 | 14.62 | 5.92 | 13.94 | 14.73 | 13.65 | 11.58 |
| AVG_Text_Generation | | 44.51 | 28.31 | 28.18 | 24.86 | 27.32 | 15.89 | 27.71 | 28.01 | 27.21 | 24.94 |
| DPR | ACC | 45.89 | 60.18 | 61.07 | 56.61 | 58.04 | 58.57 | 61.07 | 59.46 | 58.39 | 6.96 |
| WSC | ACC | 50.00 | 63.00 | 63.00 | 62.00 | 60.00 | 59.00 | 61.00 | 57.00 | 63.00 | 7.00 |
| AVG_coreference | | 47.95 | 61.59 | 62.04 | 59.30 | 59.02 | 58.79 | 61.04 | 58.23 | 60.70 | 6.98 |
| CoLA | ACC | 62.24 | 55.49 | 55.30 | 53.18 | 54.82 | 55.78 | 55.78 | 55.20 | 55.30 | 5.20 |
| FixPunct | ACC | 34.69 | 22.85 | 22.85 | 17.93 | 21.09 | 15.94 | 21.13 | 21.09 | 21.17 | 6.60 |
| TrueCase | ACC | 67.27 | 14.61 | 21.48 | 6.09 | 2.50 | 0.39 | 11.13 | 14.45 | 13.32 | 7.23 |
| AVG_Text_Correct | | 54.73 | 30.98 | 33.21 | 25.73 | 26.14 | 24.04 | 29.35 | 29.58 | 29.93 | 6.34 |
| WIC | ACC | 57.78 | 49.68 | 49.68 | 49.68 | 50.63 | 49.37 | 51.49 | 50.32 | 48.95 | 0.16 |
| Word_Segment | ACC | 62.46 | 25.70 | 23.44 | 18.87 | 19.57 | 14.57 | 24.07 | 24.06 | 23.78 | 10.74 |
| | F1 | 90.07 | 60.04 | 59.24 | 54.07 | 66.07 | 41.13 | 55.88 | 57.03 | 58.67 | 34.80 |
| AVG_Word | | 67.02 | 46.28 | 45.51 | 43.08 | 46.73 | 38.61 | 45.73 | 45.43 | 43.09 | 11.47 |

Table 6: Per-task performance of LLaMA-13B under the cross-cluster setting.

| Tasks/Methods | Metric | LoRA | HiLoRA | HiLoRA-GS | HiLoRA-ROC | Retriever | LEGO | Arrow | Phatgoose | Ensemble | Merged |
|---|---|---|---|---|---|---|---|---|---|---|---|
| ANLI_r1 | ACC | 60.30 | 41.00 | 38.50 | 30.80 | 37.10 | 32.30 | 44.20 | 36.80 | 42.80 | 0.00 |
| ANLI_r2 | ACC | 47.30 | 41.00 | 36.80 | 29.80 | 35.80 | 31.00 | 40.80 | 38.20 | 39.70 | 0.10 |
| ANLI_r3 | ACC | 49.92 | 40.17 | 37.50 | 32.42 | 37.25 | 31.92 | 40.58 | 37.42 | 40.33 | 0.00 |
| CB | ACC | 88.00 | 82.00 | 70.00 | 74.00 | 74.00 | 68.00 | 74.00 | 74.00 | 72.00 | 0.00 |
| MNLI | ACC | 87.97 | 62.42 | 60.94 | 40.74 | 60.98 | 45.66 | 61.80 | 55.82 | 62.19 | 0.04 |
| MNLI_mis | ACC | 89.80 | 62.93 | 61.37 | 39.92 | 61.91 | 46.02 | 63.01 | 56.33 | 63.09 | 0.00 |
| QNLI | ACC | 82.66 | 61.84 | 57.66 | 43.55 | 52.03 | 42.19 | 61.68 | 55.27 | 63.95 | 0.00 |
| RTE | ACC | 80.74 | 72.22 | 64.07 | 50.37 | 72.22 | 53.33 | 73.70 | 65.93 | 72.96 | 0.00 |
| SNLI | ACC | 81.91 | 53.59 | 51.21 | 30.94 | 35.82 | 39.30 | 56.21 | 53.75 | 56.09 | 0.00 |
| WNLI | ACC | 71.43 | 61.43 | 67.14 | 44.29 | 52.86 | 47.14 | 58.57 | 45.71 | 57.14 | 24.29 |
| AVG_NLI | | 74.00 | 57.86 | 54.52 | 41.68 | 52.00 | 43.69 | 57.46 | 51.92 | 57.03 | 2.44 |
| ARC_C | ACC | 48.88 | 49.48 | 48.79 | 47.76 | 47.41 | 41.03 | 46.98 | 48.1 | 48.53 | 0.00 |
| ARC_E | ACC | 58.98 | 58.77 | 58.9 | 58.26 | 58.22 | 54.92 | 57.2 | 58.22 | 58.69 | 0/00 |
| Bool_Q | ACC | 89.34 | 85.27 | 84.84 | 84.8 | 82.34 | 84.45 | 83.71 | 83.09 | 83.36 | 28.09 |
| MultiRC | ACC | 67.93 | 43.44 | 43.71 | 43.75 | 48.75 | 40.62 | 40.66 | 35.55 | 39.88 | 0.00 |
| | F1 | 71.58 | 45.19 | 45.17 | 45.22 | 51.04 | 42.47 | 43.33 | 38.51 | 42.64 | 12.25 |
| NaturalQuestions | ACC | 19.92 | 14.96 | 14.30 | 2.97 | 12.38 | 17.11 | 16.80 | 16.25 | 16.80 | 0.55 |
| | F1 | 30.87 | 26.00 | 25.15 | 15.53 | 23.83 | 27.78 | 28.86 | 25.64 | 26.84 | 9.93 |
| OpenBookQA | ACC | 63.80 | 53.20 | 52.20 | 51.60 | 49.60 | 51.00 | 53.20 | 54.20 | 53.60 | 0.20 |
| Record | ACC | 95.19 | 80.21 | 80.58 | 80.6 | 81.35 | 77.41 | 76.4 | 75.9 | 78.17 | 36.88 |
| SQuAD_v1 | ACC | 57.34 | 49.49 | 45.04 | 45.27 | 45.31 | 45.27 | 48.20 | 47.73 | 48.12 | 0.04 |
| | F1 | 75.76 | 68.52 | 64.90 | 65.44 | 66.08 | 64.59 | 68.19 | 68.32 | 68.56 | 15.29 |
| SQuAD_v2 | ACC | 70.78 | 34.87 | 31.99 | 31.95 | 29.61 | 28.20 | 32.11 | 31.52 | 31.56 | 0.00 |
| | F1 | 80.71 | 45.38 | 42.82 | 42.81 | 41.31 | 38.89 | 42.63 | 41.95 | 42.36 | 5.66 |
| TriviaQA | ACC | 60.78 | 44.84 | 54.73 | 17.07 | 50.86 | 55.20 | 57.27 | 57.15 | 57.38 | 2.03 |
| | F1 | 67.42 | 70.04 | 68.11 | 35.71 | 64.72 | 67.11 | 69.61 | 68.41 | 69.47 | 23.68 |
| AVG_QA | | 55.33 | 54.33 | 49.59 | 27.39 | 53.59 | 52.24 | 54.03 | 53.50 | 54.10 | 9.99 |
| Sentiment140 | ACC | 42.04 | 44.69 | 44.30 | 35.10 | 41.22 | 44.90 | 43.06 | 43.06 | 43.47 | 0.20 |
| | F1 | 43.59 | 46.09 | 45.63 | 39.48 | 43.56 | 45.92 | 45.01 | 44.84 | 45.38 | 9.47 |
| SST2 | ACC | 76.32 | 74.94 | 74.83 | 75.52 | 61.84 | 74.6 | 74.71 | 74.37 | 74.48 | 0.00 |
| AVG_Sentiment | | 59.57 | 60.17 | 59.95 | 56.40 | 52.12 | 60.00 | 59.37 | 59.16 | 59.45 | 2.42 |
| COPA | ACC | 72.00 | 76.00 | 75.00 | 67.00 | 72.00 | 68.00 | 74.00 | 72.00 | 74.00 | 3.00 |
| HellaSwag | ACC | 90.78 | 45.90 | 39.45 | 45.59 | 44.41 | 27.19 | 34.10 | 35.94 | 39.26 | 0.00 |
| PIQA | ACC | 66.72 | 55.52 | 56.34 | 56.07 | 57.60 | 51.75 | 55.68 | 54.92 | 57.81 | 0.77 |
| StoryCloze | ACC | 77.70 | 70.37 | 72.41 | 70.11 | 39.36 | 66.79 | 73.80 | 73.16 | 74.01 | 1.60 |
| AVG_Commonsense | | 76.80 | 61.95 | 60.80 | 53.69 | 53.34 | 53.43 | 59.40 | 59.01 | 61.27 | 1.34 |
| GLUE_MRPC | ACC | 89.25 | 71.75 | 70.00 | 63.75 | 69.25 | 68.50 | 70.75 | 68.25 | 69.50 | 18.00 |
| | F1 | 89.25 | 71.75 | 70.00 | 63.75 | 69.25 | 68.50 | 70.75 | 68.25 | 69.50 | 20.75 |
| GLUE_QQP | ACC | 84.88 | 75.86 | 76.17 | 45.47 | 69.02 | 55.62 | 57.89 | 43.95 | 51.33 | 6.41 |
| | F1 | 84.88 | 75.86 | 76.17 | 45.65 | 69.02 | 55.62 | 57.89 | 43.95 | 51.33 | 34.10 |
| STSB | ACC | 44.99 | 16.09 | 18.38 | 8.70 | 18.94 | 14.62 | 17.06 | 16.71 | 17.62 | 0.00 |
| PAWS_Wiki | ACC | 93.79 | 80.78 | 79.18 | 62.42 | 68.67 | 53.95 | 59.45 | 51.41 | 57.15 | 10.08 |
| AVG_Paraphrase | | 78.23 | 61.12 | 60.93 | 45.11 | 56.47 | 48.17 | 51.29 | 45.08 | 48.90 | 12.43 |
| CommonGen | ROUGE-1 | 55.16 | 44.13 | 46.71 | 34.99 | 47.15 | 34.02 | 43.99 | 41.94 | 44.24 | 33.74 |
| | ROUGE-2 | 25.11 | 14.14 | 15.38 | 1.56 | 2.69 | 5.04 | 14.79 | 13.99 | 14.70 | 9.96 |
| | ROUGE-L | 48.78 | 36.58 | 39.41 | 23.50 | 31.35 | 26.87 | 37.47 | 36.53 | 37.77 | 27.09 |
| | BLEU | 13.05 | 7.82 | 8.36 | 0.51 | 0.24 | 1.78 | 7.24 | 6.52 | 7.11 | 2.68 |
| DART | ROUGE-1 | 74.34 | 53.06 | 53.37 | 49.68 | 52.10 | 31.15 | 53.66 | 49.81 | 54.22 | 46.32 |
| | ROUGE-2 | 50.87 | 26.48 | 26.46 | 22.47 | 23.73 | 14.91 | 29.11 | 26.69 | 29.33 | 24.21 |
| | ROUGE-L | 58.92 | 40.34 | 40.36 | 37.25 | 38.43 | 26.31 | 41.99 | 40.12 | 42.14 | 36.00 |
| | BLEU | 41.53 | 15.66 | 16.08 | 12.18 | 12.31 | 3.32 | 16.90 | 13.33 | 17.37 | 9.61 |
| E2E_NLG | ROUGE-1 | 72.92 | 63.45 | 60.32 | 54.20 | 62.77 | 13.31 | 65.19 | 62.38 | 65.40 | 54.60 |
| | ROUGE-2 | 45.55 | 34.42 | 31.70 | 24.28 | 33.41 | 5.38 | 36.35 | 34.37 | 36.35 | 29.90 |
| | ROUGE-L | 53.34 | 46.14 | 44.15 | 37.42 | 44.67 | 11.99 | 47.01 | 45.57 | 47.06 | 39.75 |
| | BLEU | 32.50 | 20.73 | 19.46 | 8.06 | 20.85 | 0.04 | 21.26 | 17.51 | 20.93 | 18.72 |
| Gigaword | ROUGE-1 | 36.99 | 25.80 | 25.35 | 22.46 | 24.88 | 25.15 | 26.79 | 28.27 | 27.00 | 21.03 |
| | ROUGE-2 | 16.38 | 8.82 | 8.50 | 7.29 | 8.17 | 8.40 | 9.67 | 10.47 | 9.61 | 6.85 |
| | ROUGE-L | 33.25 | 22.13 | 21.82 | 19.26 | 21.30 | 21.49 | 23.17 | 24.77 | 23.32 | 18.03 |
| | BLEU | 10.38 | 3.63 | 3.32 | 2.61 | 3.05 | 3.21 | 4.27 | 4.99 | 4.25 | 2.58 |
| WebNLG_En | ROUGE-1 | 81.11 | 51.50 | 54.92 | 50.51 | 53.78 | 32.95 | 50.09 | 48.44 | 50.83 | 44.55 |
| | ROUGE-2 | 60.23 | 27.70 | 30.25 | 25.08 | 26.62 | 16.71 | 28.79 | 28.38 | 28.99 | 24.45 |
| | ROUGE-L | 65.37 | 42.50 | 44.49 | 41.10 | 43.08 | 30.17 | 41.25 | 41.25 | 42.31 | 36.40 |
| | BLEU | 52.29 | 16.26 | 18.92 | 14.21 | 14.21 | 2.63 | 15.36 | 12.04 | 16.48 | 9.53 |
| AVG_Text_Generation | | 46.40 | 30.06 | 30.47 | 24.43 | 28.24 | 15.74 | 30.76 | 29.37 | 30.97 | 24.80 |
| DPR | ACC | 90.54 | 64.11 | 64.46 | 61.61 | 63.39 | 61.61 | 64.46 | 63.21 | 64.11 | 0.18 |
| WSC | ACC | 67.00 | 59.00 | 55.00 | 52.00 | 65.00 | 45.00 | 51.00 | 43.00 | 47.00 | 2.00 |
| AVG_Coreference | | 78.77 | 61.56 | 59.73 | 56.80 | 64.20 | 53.31 | 57.73 | 53.11 | 55.55 | 1.09 |
| CoLA | ACC | 69.08 | 63.78 | 62.52 | 53.76 | 58.19 | 55.78 | 63.49 | 63.01 | 63.20 | 0.00 |
| FixPunct | ACC | 45.2 | 22.66 | 22.7 | 12.23 | 23.32 | 16.21 | 22.27 | 22.11 | 23.16 | 1.37 |
| TrueCase | ACC | 72.85 | 34.34 | 29.41 | 6.76 | 10.74 | 5.16 | 31.99 | 32.89 | 41.41 | 1.21 |
| AVG_Text_Correct | | 62.38 | 40.26 | 38.21 | 24.25 | 30.75 | 25.72 | 39.25 | 39.34 | 42.59 | 0.86 |
| WIC | ACC | 73.65 | 58.57 | 55.87 | 54.60 | 52.86 | 51.43 | 55.00 | 57.14 | 58.41 | 0.16 |
| Word_Segment | ACC | 74.06 | 30.35 | 31.02 | 23.87 | 22.38 | 13.55 | 28.41 | 26.46 | 37.55 | 3.20 |
| | F1 | 93.34 | 60.61 | 62.37 | 54.18 | 65.42 | 35.96 | 62.52 | 63.54 | 62.71 | 39.59 |
| AVG_Word | | 78.68 | 52.03 | 51.28 | 46.81 | 48.38 | 38.09 | 50.23 | 51.07 | 51.77 | 10.78 |

Table 7: Per-task performance of FLAN-T5-large under the cross-cluster setting.

| Tasks/Methods | Metric | LoRA | HiLoRA | HiLoRA-GS | HiLoRA-ROC | Retriever | LEGO | Arrow | Phatgoose | Ensemble | Merged |
|---|---|---|---|---|---|---|---|---|---|---|---|
| ANLI_r1 | ACC | 60.20 | 57.90 | 59.20 | 54.00 | 60.60 | 60.70 | 60.50 | 60.00 | 60.60 | 60.60 |
| ANLI_r2 | ACC | 43.30 | 42.70 | 42.20 | 41.40 | 42.00 | 42.80 | 43.10 | 42.90 | 43.00 | 42.70 |
| ANLI_r3 | ACC | 44.50 | 43.42 | 44.50 | 41.83 | 43.83 | 44.00 | 44.00 | 44.25 | 43.92 | 44.00 |
| CB | ACC | 78.00 | 78.00 | 78.00 | 80.00 | 80.00 | 78.00 | 78.00 | 82.00 | 78.00 | 78.00 |
| MNLI | ACC | 88.59 | 81.76 | 82.50 | 83.48 | 83.40 | 58.24 | 56.52 | 79.49 | 59.10 | 50.78 |
| MNLI_mis | ACC | 89.14 | 82.66 | 83.40 | 83.87 | 84.77 | 55.62 | 50.20 | 79.10 | 56.37 | 48.20 |
| QNLI | ACC | 82.54 | 81.09 | 80.78 | 82.38 | 82.23 | 77.93 | 75.78 | 80.98 | 78.44 | 74.10 |
| RTE | ACC | 78.89 | 74.07 | 67.04 | 74.81 | 73.33 | 59.26 | 53.33 | 72.96 | 60.00 | 52.59 |
| SNLI | ACC | 60.08 | 36.13 | 10.35 | 38.87 | 10.27 | 3.79 | 1.64 | 23.40 | 3.83 | 1.52 |
| WNLI | ACC | 52.86 | 57.14 | 38.57 | 51.43 | 60.00 | 41.43 | 42.86 | 55.71 | 44.29 | 38.57 |
| AVG_NLI | | 67.81 | 63.49 | 58.65 | 63.21 | 62.04 | 52.18 | 50.59 | 62.08 | 52.75 | 49.11 |
| Bool_Q | ACC | 87.27 | 79.22 | 76.09 | 81.60 | 80.70 | 74.73 | 73.05 | 80.04 | 75.23 | 70.39 |
| MultiRC | ACC | 55.31 | 54.02 | 53.48 | 52.42 | 53.20 | 52.97 | 53.59 | 54.02 | 53.24 | 52.77 |
| | F1 | 58.71 | 57.19 | 56.76 | 55.64 | 56.53 | 56.30 | 56.94 | 57.12 | 56.53 | 56.11 |
| SQuAD_v1 | ACC | 42.66 | 36.21 | 34.30 | 36.33 | 27.38 | 30.27 | 29.45 | 35.00 | 30.94 | 29.92 |
| | F1 | 64.25 | 59.92 | 57.50 | 57.56 | 51.23 | 53.92 | 53.01 | 58.28 | 54.31 | 53.29 |
| SQuAD_v2 | ACC | 66.56 | 65.39 | 64.26 | 64.34 | 62.97 | 63.09 | 62.46 | 64.77 | 63.09 | 61.91 |
| | F1 | 77.13 | 76.39 | 75.36 | 75.16 | 74.24 | 74.27 | 73.67 | 75.78 | 74.53 | 73.30 |
| AVG_QA | | 67.39 | 63.44 | 61.73 | 63.08 | 60.87 | 60.03 | 59.40 | 63.13 | 60.39 | 58.51 |
| Sentiment140 | ACC | 42.04 | 41.84 | 41.63 | 41.63 | 41.63 | 41.63 | 41.63 | 41.43 | 41.63 | 41.63 |
| | F1 | 43.18 | 43.41 | 43.10 | 43.21 | 43.09 | 43.01 | 43.10 | 42.82 | 43.02 | 43.02 |
| SST2 | ACC | 75.75 | 74.48 | 73.91 | 76.55 | 73.10 | 73.91 | 73.56 | 74.14 | 73.68 | 73.56 |
| AVG_Sentiment | | 59.18 | 58.55 | 58.14 | 58.49 | 57.73 | 58.11 | 57.96 | 58.13 | 58.00 | 57.94 |
| ParaCrawl_EnEs | BLEU | 27.15 | 26.78 | 26.80 | 26.30 | 26.93 | 26.71 | 26.61 | 26.66 | 26.72 | 26.53 |
| WMT16_RoEn | BLEU | 20.81 | 20.79 | 20.93 | 20.51 | 20.83 | 20.87 | 20.97 | 20.79 | 20.88 | 20.88 |
| WMT16_TrEn | BLEU | 8.94 | 8.81 | 8.66 | 8.84 | 8.86 | 8.74 | 8.65 | 8.37 | 8.71 | 8.55 |
| AVG_Translation | | 18.97 | 18.79 | 18.80 | 18.55 | 18.88 | 18.77 | 18.74 | 18.61 | 18.77 | 18.65 |
| GLUE_MRPC | ACC | 89.00 | 81.75 | 82.00 | 76.50 | 77.00 | 80.75 | 80.25 | 81.50 | 81.00 | 81.25 |
| | F1 | 89.00 | 81.75 | 82.00 | 76.50 | 77.00 | 80.75 | 80.25 | 81.50 | 81.00 | 81.25 |
| GLUE_QQP | ACC | 85.43 | 83.36 | 82.42 | 79.45 | 82.19 | 78.40 | 78.32 | 82.19 | 78.32 | 70.78 |
| | F1 | 85.43 | 83.40 | 82.89 | 79.45 | 82.19 | 81.41 | 80.16 | 82.19 | 81.60 | 80.51 |
| STSB | ACC | 44.29 | 41.99 | 41.23 | 31.89 | 37.95 | 40.39 | 39.42 | 41.57 | 41.30 | 41.02 |
| PAQS_Wiki | ACC | 94.61 | 93.59 | 93.75 | 84.14 | 92.93 | 93.48 | 93.63 | 93.79 | 93.63 | 93.95 |
| AVG_Paraphrase | | 78.33 | 75.18 | 74.91 | 68.00 | 72.52 | 73.63 | 72.85 | 74.76 | 73.97 | 72.96 |
| DART | ROUGE-1 | 76.03 | 75.82 | 75.75 | 75.95 | 75.85 | 75.77 | 75.79 | 75.66 | 75.72 | 75.60 |
| | ROUGE-2 | 54.63 | 54.39 | 54.22 | 54.27 | 54.38 | 54.21 | 54.32 | 54.17 | 54.19 | 54.11 |
| | ROUGE-L | 61.99 | 61.76 | 61.52 | 61.52 | 61.78 | 61.62 | 61.66 | 61.50 | 61.48 | 61.43 |
| | BLEU | 46.56 | 46.66 | 46.73 | 45.78 | 46.57 | 46.65 | 46.76 | 46.53 | 46.68 | 46.72 |
| E2E_NLG | ROUGE-1 | 73.08 | 73.48 | 73.27 | 73.95 | 73.55 | 73.23 | 73.09 | 73.29 | 73.19 | 73.14 |
| | ROUGE-2 | 46.57 | 46.73 | 46.62 | 46.92 | 46.77 | 46.63 | 46.47 | 46.66 | 46.57 | 46.53 |
| | ROUGE-L | 54.16 | 54.21 | 54.04 | 54.40 | 54.25 | 54.15 | 53.96 | 54.12 | 54.04 | 53.97 |
| | BLEU | 35.27 | 35.43 | 35.37 | 35.02 | 35.44 | 35.33 | 35.12 | 35.38 | 35.28 | 35.23 |
| WebNLG_En | ROUGE-1 | 83.34 | 82.50 | 82.71 | 81.77 | 82.61 | 82.70 | 82.78 | 82.49 | 82.75 | 82.71 |
| | ROUGE-2 | 64.51 | 63.24 | 63.39 | 62.52 | 63.35 | 63.29 | 63.35 | 63.26 | 63.36 | 63.27 |
| | ROUGE-L | 69.18 | 68.12 | 68.19 | 67.77 | 68.25 | 68.04 | 67.98 | 68.02 | 68.03 | 67.96 |
| | BLEU | 56.89 | 55.83 | 56.18 | 53.07 | 55.72 | 55.98 | 56.26 | 56.11 | 56.16 | 56.36 |
| AVG_Text_Generation | | 60.18 | 59.85 | 59.83 | 59.42 | 59.88 | 59.80 | 59.79 | 59.76 | 59.79 | 59.75 |
| DPR | ACC | 86.25 | 76.79 | 76.25 | 78.21 | 76.07 | 75.89 | 75.89 | 77.14 | 76.07 | 75.36 |
| WSC | ACC | 40.00 | 51.00 | 47.00 | 57.00 | 48.00 | 46.00 | 46.00 | 47.00 | 48.00 | 46.00 |
| AVG_Coreference | | 63.12 | 63.89 | 61.62 | 63.61 | 62.04 | 60.95 | 60.95 | 62.07 | 62.04 | 60.68 |
| CoLA | ACC | 64.26 | 58.00 | 56.55 | 59.25 | 56.17 | 56.94 | 56.36 | 56.84 | 56.94 | 56.45 |
| FixPunct | ACC | 41.25 | 39.45 | 40.16 | 37.27 | 39.61 | 40.47 | 40.55 | 40.39 | 40.62 | 40.62 |
| TrueCase | ACC | 59.22 | 67.03 | 65.94 | 64.53 | 66.25 | 66.29 | 66.45 | 66.80 | 66.33 | 65.55 |
| AVG_Text_Correct | | 54.91 | 54.83 | 54.21 | 53.68 | 54.01 | 54.56 | 54.45 | 54.68 | 54.63 | 54.21 |
| WIC | ACC | 66.98 | 65.71 | 65.40 | 48.25 | 65.40 | 66.03 | 63.81 | 64.92 | 66.19 | 65.40 |
| Word_Segment | ACC | 63.91 | 67.15 | 67.34 | 66.72 | 67.07 | 70.62 | 70.20 | 71.53 | 70.55 | 70.35 |
| | F1 | 88.33 | 90.83 | 90.74 | 92.82 | 90.52 | 92.76 | 92.56 | 93.15 | 92.72 | 92.47 |
| AVG_Word | | 71.55 | 73.35 | 72.22 | 64.01 | 72.10 | 73.86 | 72.59 | 73.63 | 73.91 | 73.40 |

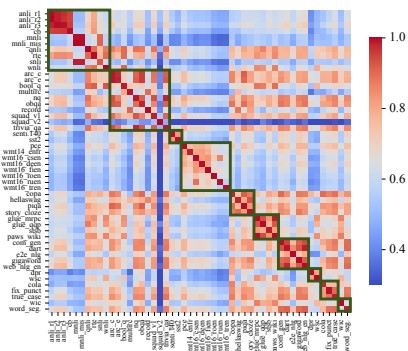

Figure 13: Input-LoRA similarity heatmap produced by `Retriever`, where tasks from the same cluster are enclosed within green boxes for clarity.

Table 8: Per-batch input-mapping and inference latency across tasks under varying pool sizes (different number of seen tasks).

| Type | Pool size | ANLI_r1 | ARC_C | Sentiment140 | ParaCrawl | COPA | GLUE_MRPC | COMMON_GEN | DPR | COLA | WIC |
|------|-----------|---------|-------|--------------|-----------|------|-----------|------------|-----|------|-----|
| Mapping | 5 | 0.0612 | 0.0596 | 0.0266 | 0.0578 | 0.0259 | 0.0490 | 0.0291 | 0.0286 | 0.0356 | 0.0445 |
| | 10 | 0.0597 | 0.0595 | 0.0440 | 0.0617 | 0.0225 | 0.0449 | 0.0304 | 0.0316 | 0.0391 | 0.0493 |
| | 15 | 0.0705 | 0.0576 | 0.0340 | 0.0637 | 0.0303 | 0.0465 | 0.0333 | 0.0451 | 0.0423 | 0.0481 |
| | 20 | 0.0807 | 0.0630 | 0.0494 | 0.0652 | 0.0396 | 0.0537 | 0.0398 | 0.0389 | 0.0474 | 0.0486 |
| | 25 | 0.0821 | 0.0749 | 0.0424 | 0.0664 | 0.0304 | 0.0525 | 0.0423 | 0.0417 | 0.0487 | 0.0683 |
| | 30 | 0.0886 | 0.0789 | 0.0490 | 0.0668 | 0.0336 | 0.0599 | 0.0378 | 0.0401 | 0.0431 | 0.0605 |
| | 35 | 0.0910 | 0.0759 | 0.0580 | 0.0627 | 0.0343 | 0.0672 | 0.0408 | 0.0466 | 0.0420 | 0.0673 |
| | 40 | 0.1026 | 0.0809 | 0.0566 | 0.0862 | 0.0390 | 0.0725 | 0.0453 | 0.0404 | 0.0411 | 0.0615 |
| Inference | 5 | 4.0765 | 2.2196 | 1.8765 | 14.5083 | 1.0900 | 0.5494 | 2.1211 | 1.2808 | 0.8343 | 0.8477 |
| | 10 | 10.0089 | 1.4993 | 1.7331 | 19.8633 | 1.0506 | 0.6800 | 2.1986 | 1.6111 | 0.8586 | 1.4085 |
| | 15 | 10.6356 | 1.5529 | 1.9547 | 21.3176 | 1.1875 | 0.4262 | 2.3208 | 1.5352 | 0.9120 | 1.0494 |
| | 20 | 7.1220 | 1.5537 | 1.9695 | 19.8872 | 1.1717 | 0.4234 | 2.1853 | 1.4684 | 0.9324 | 1.0237 |
| | 25 | 8.6094 | 1.5760 | 1.9752 | 21.2259 | 1.2098 | 0.4206 | 2.4156 | 1.4883 | 0.9311 | 1.1112 |
| | 30 | 9.7096 | 1.5021 | 1.9194 | 20.3829 | 1.1304 | 0.4111 | 2.1553 | 1.4695 | 0.9009 | 0.9758 |
| | 35 | 10.1859 | 1.7774 | 1.9127 | 20.1784 | 1.1404 | 0.3770 | 2.2162 | 1.4004 | 0.8978 | 0.9948 |
| | 40 | 10.8998 | 1.7849 | 1.8663 | 19.9678 | 1.2508 | 0.3633 | 2.1733 | 1.4737 | 0.8520 | 0.9701 |

```
{
    "model_name": "<task_name>",
    "sample": [
        {
            "inputs": "<instruction-style input 1>"
        },
        {
            "inputs": "<instruction-style input 2>"
        },
        ...
    ]
}
"
```

# D  EMPIRICAL VALIDATION OF THEORETICAL GUARANTEES

To complement the theoretical analysis in Sec. 3, we empirically examine whether the assumptions required by Theorem 1 and Theorem 2 hold in practice, and whether the resulting bounds behave as predicted.

**Domain Separability.** We begin by evaluating the separability of task domains by computing the pairwise KL divergence between the Gaussian distributions fitted for each LoRA. As shown in Fig.14(a), most task pairs exhibit large KL values, with an average divergence of 1432. These results indicate that the task domains are well separated in practice. This observation confirms that the key assumption required by Theorem 1 and Theorem 2 is satisfied, since greater inter-domain divergence corresponds to a lower probability of routing error.

Table 9: Performance of `HiLoRA` under different values of the scaling factor $\gamma$ across multiple task types.

| $\gamma$ | 20% | 40% | 60% | 80% | 100% |
|---|---|---|---|---|---|
| Within-NLI | 62.9569 | 63.4682 | 62.8199 | 62.6141 | 61.8909 |
| Cross-NLI | 44.0750 | 46.4861 | 45.4724 | 44.5162 | 44.2294 |
| Cross-Trans. | 19.5152 | 20.7552 | 21.2560 | 21.5565 | 21.8029 |
| Cross-StT | 27.5563 | 28.1187 | 28.7055 | 28.6063 | 28.1832 |

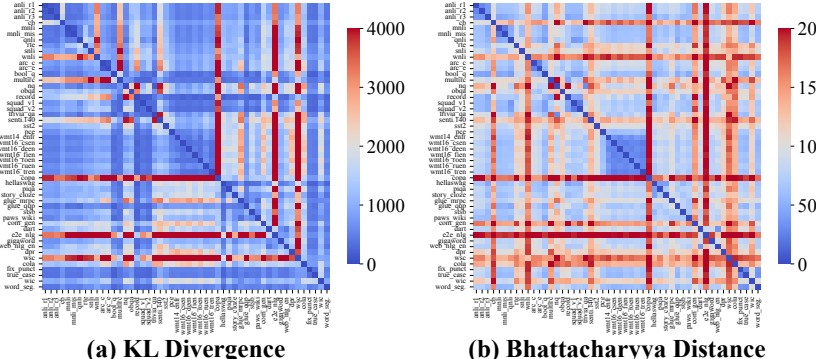

**(a) KL Divergence**  **(b) Bhattacharyya Distance**

Figure 14: (a) KL divergence and (b) Bhattacharyya distance computed across all task pairs used in our experiments.

**Verification of Theorem 1.** For Theorem 1, we analyze $B_{ij}$, the Bhattacharyya distance between Gaussian distributions of task pairs, which determines the exponential decay term in the error bound. As shown in Fig.14(b), $B_{ij}$ is strictly positive across all pairs and typically large, with an average value of $108.35$. These results indicate that task domains are well separated in practice. Such substantial divergence ensures that the bound in Theorem 1 is operationally meaningful, as greater domain separability significantly reduces the probability that the correct LoRA is excluded from the Top-$k$ set.

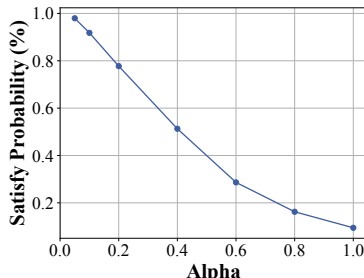

Figure 15: Satisfaction rate of the positive-definiteness condition $M_\alpha^j \succ 0$ in Theorem 2 across task pairs under varying $\alpha$.

**Verification of Theorem 2.** To validate Theorem 2, we examine the feasibility condition $M_\alpha^j = \Sigma q^{-1} + \alpha \Sigma j^{-1} - \alpha \Sigma i^{\star-1} \succ 0$. Since each covariance inverse ($\Sigma^{-1}$) is positive definite, the matrix $M\alpha^j$ remains positive definite when $\alpha$ is sufficiently small, because the sum of positive-definite matrices is positive-definite and the subtraction term is scaled down by $\alpha$. Thus, from a theoretical standpoint, the condition is expected to hold with high probability for small and moderate $\alpha$. We empirically verify this by evaluating the proportion of task pairs satisfying $M_\alpha^j \succ 0$ across different values of $\alpha$. As shown in Fig. 15, the condition indeed holds with high probability when $\alpha$ is small or moderate, confirming that the assumptions required by our OOD error bound are realistic in our experimental setting.

Taken together, these results show that the assumptions under which our theoretical bounds become tight are frequently met in practice. The fitted task distributions are well separated, the key ID

divergence term $B_{i,j}$ is sufficiently large, and the OOD conditions hold with high probability. These empirical findings validate that `HiLoRA`'s theoretical guarantees are not merely abstract but translate into reliable behavior in real-world applications.

## E  LLM USAGE

Large Language Models (LLMs) were used solely to aid in the writing and polishing of the manuscript. LLMs, specifically ChatGPT, were employed exclusively as writing assistants in the preparation of this manuscript. Their role was limited to improving the presentation quality of the text, including tasks such as rephrasing sentences, correcting grammar, enhancing readability, and improving the overall flow of exposition. The use of LLMs was confined to linguistic refinement, and they were not involved in generating, verifying, or shaping any scientific ideas.

All research contributions, including the formulation of research questions, algorithmic design, theoretical derivations, and experimental studies, were conceived and executed entirely by the authors. Their contribution was restricted to stylistic and grammatical adjustments, with no bearing on the substance of the research.

The authors retain full responsibility for the entire content of this work, including any text improved with LLM assistance. We have carefully ensured that the usage of LLMs complies with ethical standards and does not introduce plagiarism, fabrication, or other forms of scientific misconduct.

