# OpenReview forum: "HiLoRA: Adaptive Hierarchical LoRA Routing for Training-Free Domain Generalization"
_ICLR.cc/2026/Conference — Submitted to ICLR 2026_

### Official Review · Reviewer_xGHJ · 2025-10-28

**Soundness:** 3
**Presentation:** 3
**Contribution:** 3
**Rating:** 6
**Confidence:** 4

**Summary:**

This paper proposes HiLoRA, a training-free framework that addresses domain generalization without task labels or additional training through adaptive hierarchical routing in LoRA pools. The core concept of HiLoRA is to treat each rank-one component (ROC) in LoRA as a basic unit, dynamically selecting the most relevant ROCs at both sequence and token levels. Specifically, at the sequence level, HiLoRA calculates similarity between input and each LoRA based on Gaussian likelihood, thereby selecting a subset of LoRA and assigning appropriate ROC quantities. At the token level, the framework further refines routing by activating only the most informative ROCs. Experimental results demonstrate that HiLoRA improves accuracy by 55% on LLaMA2-7B and 13% on FLAN-T5-large.

**Strengths:**

1.This study investigates a critical issue in the domain generalization of LoRA. Through detailed analysis of LoRA parameters, three key observations on ROC characteristics were identified, leading to the development of the HiLoRA framework. This clarifies the motivation behind the work and provides a clear logical flow.
2. The method's error bound is theoretically demonstrated under different scenarios.
3.The authors have conducted extensive experiments to demonstrate the strong performance of HiLoRA compared to baseline methods.

**Weaknesses:**

1.It lacks analysis or experiments have been conducted for large-scale deployment. When processing large batch inference data, the throughput of this method appears difficult to guarantee.
2.The model architecture and scale verified in this paper are limited. We hope to see its performance on larger models or MoE architectures.

**Questions:**

1. How to guarantee the performance on large batching inference.
2. Is it possible to analyze the issue of load balancing of the routing mechanism?

---

> ### Author Response · Authors · 2025-11-22
>
> Thank you for your insightful and valuable feedback. We will address each of your concerns with a point-by-point response. We promise to include new results and discussions into the final version if the paper is accepted.
>
> > **Weakness 1:** It lacks analysis or experiments have been conducted for large-scale deployment. When processing large batch inference data, the throughput of this method appears difficult to guarantee.
> >
> > **Questions 1:** How to guarantee the performance on large batching inference.
>
> **Response**： We thank the reviewer for the question. HiLoRA scales well to large-batch inference for two reasons:
>
> - Routing is amortized over the batch. Sequence-level routing is done once per input sequence, and its cost O(B \cdot I), where B is batch size and I is the number of LoRAs in the pool. This routing cost remains negligible compared with the Transformer forward pass.
> - Token-level routing is fully vectorized. ROC top-k selection uses GPU-efficient batched GEMMs and torch.topk, so it introduces no sequential bottleneck. This allows the token-level mechanism to scale smoothly with increasing batch size.
>
> To further verify scalability, we measured **input-mapping latency** and **per-batch inference latency** for batch sizes from 1 to 128 (reported in the revised version). The results show that latency increases smoothly and predictably with batch size, confirming that HiLoRA maintains efficient throughput under large-batch settings.
>
> | Type      | 1      | 2      | 4      | 8      | 16     | 32     | 64     | 128    |
> | - | - | - | - | - | - | - | - | - |
> | Mapping   | 0.0187 | 0.0209 | 0.0199 | 0.0268 | 0.0485 | 0.0614 | 0.0883 | 0.1481 |
> | Inference | 0.5145 | 0.8734 | 1.3084 | 1.8603 | 2.7629 | 3.5215 | 4.9484 | 6.9385 |
>
>
> > **Weakness 2:** The model architecture and scale verified in this paper are limited. We hope to see its performance on larger models or MoE architectures.
>
> **Response**： We thank the reviewer for this suggestion. To evaluate the scalability of our method with respect to model size, we conducted additional experiments on **LLaMA-2-13B**, a substantially larger model that exceeds the scale explored in most existing training-free or LoRA-based adaptation studies. The new results (included in the revised version) show that HiLoRA preserves the same performance trends and continues to outperform competing baselines, demonstrating that the method generalizes reliably to larger architectures without degradation.
>
> | Methods  | LoRA | HiLoRA | HiLoRA-GS | HiLoRA-ROC | Retriever | LEGO | Arrow | Phatgoose | Ensemble | Merged |
> | -- | -- | -- | ---| ---- | -- | ----- | ------ | -- | -- | - |
> | ANLI_r1  | 60.3 | 62.7 | 60.6 | 58.8 | 61.8 | 58.3 | 47.4 | 55.6 | 52.0 | 24.7 |
> | ANLI_r2  | 47.3 | 48.6 | 47.0 | 46.2 | 48.5 | 44.3 | 41.0 | 43.7 | 41.4 | 24.3 |
> | ANLI_r3  | 49.92 | 48.92 | 47.75 | 49.08 | 49.75 | 47.17 | 44.92 | 47.33 | 46.42 | 26.42 |
> | CB       | 88.0 | 86.0 | 84.0 | 80.0 | 82.0 | 84.0 | 80.0 | 76.0 | 84.0 | 38.0 |
> | MNLI     | 87.97 | 87.58 | 87.34 | 82.58 | 86.56 | 85.47 | 70.00 | 76.45 | 75.35 | 37.30 |
> | MNLI_mis | 89.80 | 89.49 | 88.63 | 83.79 | 88.87 | 85.70 | 71.02 | 75.94 | 76.41 | 36.64 |
> | QNLI     | 82.66 | 83.13 | 82.66 | 56.29 | 77.89 | 68.95 | 67.97 | 70.98 | 69.96 | 44.73 |
> | RTE      | 80.74 | 75.19 | 75.56 | 73.33 | 74.81 | 72.59 | 78.89 | 73.33 | 76.30 | 52.59 |
> | SNLI     | 81.91 | 81.84 | 81.99 | 76.72 | 81.60 | 79.65 | 67.19 | 75.23 | 72.38 | 32.46 |
> | WNLI     | 71.43 | 67.14 | 67.14 | 67.14 | 51.43 | 64.29 | 62.86 | 58.57 | 65.71 | 42.80 |
> | AVG      | 74.00 | 72.86 | 72.47 | 67.39 | 70.32 | 69.04 | 63.12 | 65.31 | 65.99 | 36.00 |
>
> Due to space constraints, we report only the within-cluster results here. The full cross-cluster results are provided in Appendix C.3 (Table 6) of the revised version. We have included these additional experiments and clarifications in the updated manuscript to strengthen the empirical evaluation.
>
> > **Questions 2:** Is it possible to analyze the issue of load balancing of the routing mechanism?
>
> **Response**： We thank the reviewer for the question. HiLoRA does not explicitly optimize for load balancing across LoRAs; its routing mechanism selects LoRAs solely based on Gaussian likelihoods and ROC-level relevance, so the selection frequency naturally reflects semantic affinity rather than uniform distribution. However, load imbalance can be mitigated in practice by adding lightweight LoRA replicas when needed. Prior work has shown that LoRA adapters contain only a very small number of parameters (typically 0.01%–0.1% of the base model) and impose negligible memory overhead [1], making replication inexpensive. A more systematic study of load balancing in routing-based adaptation is an interesting direction, and we plan to explore this in future work. We have added this point to the conclusion in the revised version.
>
> [1] Hu E J, Shen Y, Wallis P, et al. Lora: Low-rank adaptation of large language models. ICLR, 2022.

---

> ### Author Response · Authors · 2025-11-26
>
> Dear Reviewer `xGHJ`,
>
> As the discussion phase is drawing to a close, we kindly ask whether our responses have resolved your concerns or if there are any remaining issues that we can further clarify. Your insights are invaluable in refining our work, and we are eager to ensure that all your concerns are fully addressed.
>
> Thank you once again for your time and effort in reviewing our manuscript.
>
> Best regards,
>
> The Authors of Submission 4341

---

> ### Comment · Reviewer_xGHJ · 2025-11-27
>
> I appreciate the authors for the meticulous responses and the inclusion of extensive experimental data. The additional experiments on large batching inference and Llama2-13B model are convincing.Further, the authors response my concern on the issue of load balancing of the routing mechanism. As I have already given a positive score and the authors have addressed my concerns, I am willing to keep my rating.

---

> > ### Author Response · Authors · 2025-11-27
> >
> > Dear Reviewer xGHJ,
> >
> > We thank you for the positive feedback and for confirming that our additional experiments have addressed the earlier concerns. We appreciate your thoughtful comments and your continued support of our work.
> >
> > Best regards,
> >
> > The Authors of Submission 4341

---

### Official Review · Reviewer_TB3c · 2025-10-30

**Soundness:** 2
**Presentation:** 3
**Contribution:** 2
**Rating:** 4
**Confidence:** 4

**Summary:**

This paper introduces HiLoRA, a training-free framework that leverages existing Low-Rank Adaptation (LoRA) module pools to achieve domain generalization. The core idea involves performing adaptive hierarchical routing: first, at the sequence level, each LoRA is modeled as a Gaussian distribution to compute the likelihood of the input, thereby screening a relevant subset of LoRAs while determining the budget for "Rank-One Components (ROCs)"—defined as the fundamental building blocks of LoRA; second, at the token level, the selection is refined by activating the most responsive ROCs based on the down-projected vector of each token. The authors provide theoretical guarantees for the LoRA selection process and demonstrate through experiments that HiLoRA outper

**Strengths:**

1.  Interesting Hierarchical Framework

    The two-stage routing mechanism proposed in the paper is logically rigorous.

    a.  First, using Rank-One Components (ROCs) as the minimal fundamental units for routing is a key conceptual contribution (Section 2 "Dyadic Product Representation", Section 3.1), breaking through the previous coarser-grained full-module routing pattern and achieving finer adaptation.

    b.  Second, the hierarchical design combines practicality and effectiveness (Figure 3, Section 3.1), first narrowing the vast search space at the sequence level ("input-aware ROC allocation"), then performing low-cost fine optimization at the token level ("token-level ROC routing"), cleverly balancing expressive power and computational complexity.

    c.  Third, the training-free nature of the framework is an important practical advantage (Abstract, Section 3.1), allowing direct deployment on existing community-built LoRA hubs (e.g., HuggingFace) without expensive gradient-based training for the gating module, lowering the application barrier.

2.  Theoretical Support for the Routing Mechanism

    In this field which is strongly empirically oriented, the paper provides formal theoretical analysis for the design choices.

    a.  First, the authors derived error bounds for the LoRA identification process for in-distribution (ID) and out-of-distribution (OOD) inputs (Section 3.4, Theorems 1 and 2), formally guaranteeing high-probability screening of the most relevant LoRA, enhancing the rigor of the method.

    b.  Second, the theory explicitly links routing errors to interpretable factors, such as task distribution separability (Bhattacharyya distance Bij in Theorem 1) and the size k of the selected LoRA set (Note in Section 3.4), providing valuable insights into the applicability conditions of HiLoRA.

    c.  Third, the use of Gaussian likelihood to measure similarity in Section 3.2 is not a simple heuristic but forms a coherent and principled scheme.

**Weaknesses:**

### 1. Dependence on Original Training Data Samples
The practicality of this method is limited by a critical assumption: the sequence-level routing mechanism relies entirely on fitting a Gaussian distribution using $m$ samples from each LoRA's original training dataset (Sec. 3.2, Lines 209-212). In many real-world scenarios, these training datasets are proprietary, private, or completely unavailable. Although the paper acknowledges this issue in the conclusion (Sec. 5), it does not address it through experiments. This significantly restricts the claimed "training-free" and "plug-and-play" applicability of the method.

### 2. Limited Generalizability of Motivating Observations
The core motivation for ROC-level routing is based on limited empirical evidence. The key observations regarding the randomness of down-projection vectors and the clustering of up-projection vectors are derived solely from PCA visualizations of five Natural Language Inference (NLI) tasks and selected layers (Fig. 2 in Sec. 3.1; Figs. 7 & 8 in Appendix C.2). This evidence is insufficiently comprehensive to establish these properties as universal characteristics of LoRAs across all tasks and model architectures.

### 3. Insufficient Hyperparameter Sensitivity Analysis
The robustness of the method has not been fully validated due to the lack of sensitivity analysis for several key hyperparameters. First, the number of samples $m$ is fixed at 20 (Sec. 4.1), and its impact has not been investigated. Second, the base number of experts $k$ for LoRA-level routing is fixed at 3 (Sec. 4.1, Line 355); although Eq. 4 provides some adaptability for unseen tasks, performance still depends on this base value. Given that $k$ is crucial for both performance and efficiency, no ablation study has been conducted on it. Finally, the ablation study for the scaling factor $\gamma$ is only performed in the within-cluster setting of the NLI cluster (Fig. 6), making it impossible to determine whether $\gamma=40\%$ is optimal for other task clusters with different structures (e.g., translation tasks) or for the more challenging cross-cluster setting.

### 4. Unvalidated Scalability to Large LoRA Pools
The paper claims to address the challenge of repositories containing "thousands of task-specific LoRAs" (Sec. 1, Line 38), but experiments are only conducted on relatively small LoRA pools: the LoRA pools used in experiments have sizes of 50 (for LLaMA2-7B) and 33 (for FLAN-T5-large) (Sec. 4.2). The throughput analysis in Fig. 5 shows a clear downward trend in throughput as the number of seen tasks increases to 40, yet it does not evaluate how throughput changes with the total pool size $I$. The cost of computing Gaussian likelihoods (Sec. 3.2) and down-projections (Sec. 3.3) scales linearly with $I$.

### 5. Ambiguity and Complexity of Theoretical Guarantees
While the inclusion of theoretical analysis is a strength, its presentation and practical validation are weak. Theorem 2 (Sec. 3.4, Eq. 6) has high mathematical complexity; despite a brief explanation, its components (e.g., $M_j^\alpha$, $h_j^\alpha$) remain difficult to intuitively understand. Additionally, the crucial assumption $M_j^\alpha > 0$ is presented without any discussion of how frequently it holds in practice.

### 6. Inconsistent Performance Gains and Model Dependency
The effectiveness of the method is highly dependent on the base model: HiLoRA shows significant performance gains on LLaMA2-7B, but the gains are much more modest on FLAN-T5-large, and it is often not the top-performing method (Tables 1 & 2). The paper explains that T5 has already undergone extensive instruction tuning (Lines 373-375), which is a plausible reason but also suggests that the utility of this method may be limited to base models with relatively weak initial capabilities.

### 7. Lack of Statistical Significance Testing
Experimental results are presented as single-run point estimates. Due to the stochasticity introduced by multinomial sampling for ROC allocation (Sec. 3.2), there is likely variance in the results. Without error bars or standard deviations from multiple runs, it is difficult to assess the statistical significance of the reported performance gains over baseline methods, especially when the gain margins are small (e.g., the results of FLAN-T5-large in Table 2).

### 8. Unvalidated Impact of Key Components
The contribution of certain design choices has not been empirically validated: a variance normalization step is introduced in Sec. 3.3, Lines 260-269 to stabilize performance, with reference to [Zhao et al., 2025a]. However, its actual impact within the HiLoRA framework has not been measured.

### 9. Potentially Unfair Comparison for ROC-Level Baselines
The setup for ROC-level baseline methods may not be optimal. Methods such as LEGO and HiLoRA-ROC are assigned a fixed budget of $k=24$ ROCs (Sec. 4.1, Line 357), which contrasts with HiLoRA's adaptive budget. It is possible that $k=24$ is not the optimal choice for these baselines, thereby undermining their performance.

### 10. Dependency on a Specific High-Quality Embedding Model
The sequence-level routing heavily relies on the selected embedding model: the method uses instructor-base (a powerful instruction-tuned encoder) to generate task-aware embeddings (Sec. 4.1, Lines 351-353). The high quality of these embeddings is likely essential for the effectiveness of Gaussian likelihood matching, and the method's performance may degrade significantly when using more standard, off-the-shelf sentence encoders.

### 11. Outdated and Insufficient Tested Models
The number of tested models is outdated and insufficient; the paper still uses LLaMA2 instead of current mainstream models, failing to align with the latest model development trends and limiting the persuasiveness of its performance generalizability.

**Questions:**

1. Sequence-level routing requires 20 samples ($m=20$) from the original training data of each LoRA. How would the method perform when these samples are often unavailable for publicly shared LoRAs? Have you considered using proxy data from relevant public datasets, and what would be the expected impact?
2. Could you conduct a sensitivity analysis on the number of samples $m$ used for fitting the Gaussian distribution? How does the performance degrade when $m$ decreases (e.g., to 5 or 1), and at what value of $m$ would the sequence-level routing fail?
3. The selection of instructor-base as the embedding model seems crucial. How robust is HiLoRA’s performance to the choice of this embedding model? What would happen to the results if a conventional non-instruction-tuned sentence encoder is used instead?
4. The paper’s motivation mentions that it is scalable to "thousands" of LoRAs, but the experiments are only based on a LoRA pool with a maximum size of 50. When the LoRA pool size $I$ increases to hundreds or thousands, what are the scaling trends of inference throughput and task routing accuracy?
5. Regarding the theoretical analysis, can you empirically verify the $M_j^\alpha > 0$ assumption in Theorem 2 using the task pairs from your experiments? How frequently does this condition hold in practice, and what implications would there be if it does not hold?
6. The optimal value of the scaling factor $\gamma$ (40%) is determined based on an ablation experiment on the NLI cluster (Fig. 6). Is this optimal value still applicable to task clusters with different structures (e.g., Translation or Struct-to-Text tasks), especially in the more challenging cross-cluster setting?
7. The minimum number of selected LoRAs (hyperparameter $k$) in Eq. 4 is fixed at 3. Could you conduct a sensitivity analysis on this parameter to show its impact on the performance of both seen and unseen tasks?
8. To what extent does the current method depend on a pre-trained LoRA pool? How does its performance compare to that of models trained on large amounts of general data?

---

> ### Author Response · Authors · 2025-11-22
> **Dependence on Original Training Data Samples and  a Specific High-Quality Embedding Model**
>
> Thank you for your thoughtful questions and constructive feedback. We will address each of your concerns with a point-by-point response. Thank you for your insightful and valuable feedback. We will address each of your concerns with a point-by-point response. We promise to include new results and discussions into the final version if the paper is accepted.
>
> > **Weakness 1:** Dependence on Original Training Data Samples
> >
> > **Weakness 10:** Dependency on a Specific High-Quality Embedding Model
> >
> > **Questions 1:** Sequence-level routing requires 20 samples ($m=20$) from the original training data of each LoRA. How would the method perform when these samples are often unavailable for publicly shared LoRAs? Have you considered using proxy data from relevant public datasets, and what would be the expected impact?
> >
> > **Questions 2:** Could you conduct a sensitivity analysis on the number of samples used for fitting the Gaussian distribution? How does the performance degrade when decreases (e.g., to 5 or 1), and at what value of would the sequence-level routing fail?
> >
> > **Questions 3:** The selection of instructor-base as the embedding model seems crucial. How robust is HiLoRA’s performance to the choice of this embedding model? What would happen to the results if a conventional non-instruction-tuned sentence encoder is used instead?
>
> **Response**：We thank the reviewer for these related concerns regarding the availability of task samples, the required sample size, and the dependence on the embedding model. To address them, we conducted a unified sensitivity analysis on NLI tasks under both within-cluster and cross-cluster settings, varying:
> (i) the number of samples per LoRA  $m$,
> (ii) the use of synthetic samples generated by GPT, and
> (iii) the embedding model  $\mathbb{E}$, including a non–instruction-tuned encoder.
>
> The results, summarized in the table below, show that HiLoRA is **consistently robust** across all tested conditions. Specifically, we have:
> - Sample size. Reducing m leads to only a small accuracy drop (**≤5%** on average). The method remains competitive even with **two samples**, the minimum required to fit a Gaussian distribution.
> - Synthetic samples. GPT-generated proxy examples yield slightly lower accuracy than real samples, but HiLoRA still performs **on par with or better than** other baselines. In practice, public LoRA repositories typically provide a few short prompts, making this requirement feasible.
> - Embedding model. Replacing the instructor-tuned encoder with a standard non–instruction-tuned model (MPNet-base-v2) results in only **a modest accuracy drop (≤3%)**, indicating that HiLoRA is not overly sensitive to the embedding backbone.
>
> | Factors  | Within 2-sample | Within 5-sample | Within 10-sample | Within 20-sample | Within AI-sample | Within MPNet | Cross 2-sample | Cross 5-sample | Cross 10-sample | Cross 20-sample | Cross MPNet |
> | - | - | -| - | - | - | - | - | - | - | - | - |
> | ANLI_r1  | 43.30 | 45.00 | 45.00 | 45.00 | 35.80 | 45.00 | 29.40 | 30.70 | 30.50 | 30.70 | 31.90 |
> | ANLI_r2  | 38.40 | 39.60 | 38.70 | 40.60 | 38.80 | 39.30 | 27.20 | 28.70 | 30.40 | 34.50 | 34.90 |
> | ANLI_r3  | 38.75 | 37.83 | 37.75 | 37.67 | 34.75 | 36.75 | 30.67 | 29.50 | 28.58 | 31.67 | 31.75 |
> | CB       | 64.00 | 66.00 | 66.00 | 68.00 | 66.00 | 68.00 | 74.00 | 74.00 | 76.00 | 70.00 | 74.00 |
> | MNLI     | 71.80 | 78.36 | 78.44 | 76.33 | 67.66 | 73.24 | 51.84 | 53.24 | 53.28 | 50.74 | 48.24 |
> | MNLI_mis | 73.16 | 80.86 | 81.05 | 78.59 | 66.76 | 73.55 | 52.03 | 53.91 | 53.36 | 51.29 | 49.57 |
> | QNLI     | 59.22 | 69.06 | 68.20 | 78.28 | 59.34 | 65.80 | 45.61 | 44.38 | 43.48 | 46.84 | 44.77 |
> | RTE      | 72.22 | 71.85 | 71.85 | 74.44 | 67.41 | 73.33 | 59.63 | 57.78 | 58.15 | 62.22 | 61.85 |
> | SNLI     | 69.06 | 69.77 | 69.88 | 69.45 | 68.55 | 70.20 | 42.97 | 42.27 | 42.77 | 40.31 | 39.30 |
> | WNLI     | 54.29 | 60.00 | 64.29 | 65.71 | 50.00 | 61.43 | 47.14 | 48.57 | 48.57 | 47.14 | 45.71 |
> | **Avg**  | 58.42 | 61.83 | 62.12 | 63.41 | 55.51 | 60.66 | 46.05 | 46.30 | 46.51 | 46.54 | 46.20 |
>
> Overall, these findings demonstrate that HiLoRA remains practical when only limited samples are available, when synthetic examples must be used, or when switching to a simpler embedding model. This directly addresses the reviewer’s concerns regarding feasibility and robustness.
>
> We have incorporated these analyses into Section 4.3 of the revised manuscript.

---

> ### Author Response · Authors · 2025-11-22
>
> > **Weakness 2:** Limited Generalizability of Motivating Observations
>
> **Response**：We thank the reviewer for raising this concern. The observations regarding the randomness of the down-projection vectors ($\mathbf{a}_i$) and the clustering of the up-projection vectors ($\mathbf{b}_i$) are not based solely on a small subset of layers or tasks. In our experiments, these properties hold consistently across **nearly all layers**; due to space limitations, we presented only representative visualizations in the main paper (Fig. 2) and Appendix (Figs. 7–8).
>
> To provide more comprehensive evidence, we further computed full cosine-similarity distributions for all $\mathbf{a}_i$ and $\mathbf{b}_i$ vectors across every LoRA (Figs. 9–12 in Appendix C.2). The results show that:
>
> - Down-projection vectors ($\mathbf{a}_i$) exhibit a near-uniform similarity distribution **centered around zero**, indicating a highly dispersed distribution and show little alignment with task semantics.
> - Up-projection vectors ($\mathbf{b}_i$) display distinct, non-uniform similarity patterns, where similarities are **systematically higher** within clusters than across clusters, demonstrating that $\mathbf{b}_i$ encodes stable task-dependent directions.
>
> These findings confirm that the PCA observations are not isolated artifacts, but instead align with the global similarity structure across all LoRAs.
>
> > **Weakness 3:** Insufficient Hyperparameter Sensitivity Analysis
> >
> > **Questions 6:** The optimal value of the scaling factor (40%) is determined based on an ablation experiment on the NLI cluster (Fig. 6). Is this optimal value still applicable to task clusters with different structures (e.g., Translation or Struct-to-Text tasks), especially in the more challenging cross-cluster setting?
> >
> > **Questions 7:**The minimum number of selected LoRAs (hyperparameter ) in Eq. 4 is fixed at 3. Could you conduct a sensitivity analysis on this parameter to show its impact on the performance of both seen and unseen tasks?
>
> **Response**：We thank the reviewer for these related questions concerning the sensitivity of our method to hyperparameters, including the scaling factor $\gamma$ and the minimum number of selected LoRAs.
>
> To address these concerns, we extended the ablation study of $\gamma$ to task families beyond NLI, including **Translation** and **Struct-to-Text**. The results show that the optimal $\gamma$ is **task-dependent**: Translation tasks favor $\gamma=100%$, Struct-to-Text tasks perform best around $60%$, and NLI tasks prefer $40%$. These differences stem from two factors. First, task complexity varies: generation-style tasks require richer semantic and syntactic information, and therefore benefit from activating more ROCs. Second, semantic alignment with the LoRA pool differs across tasks: when alignment is low, incorporating a larger portion of ROCs helps capture relevant signals. Overall, while $\gamma=40%$ provides a balanced default across clusters, its optimal value naturally depends on task difficulty and its proximity to the LoRA pool.
>
> | Task Cluster | 20    | 40    | 60    | 80    | 100   |
> | ------------ | ----- | ----- | ----- | ----- | ----- |
> | within-NLI   | 62.96 | 63.47 | 62.82 | 62.61 | 61.89 |
> | cross-NLI    | 44.07 | 46.49 | 45.47 | 44.52 | 44.23 |
> | cross-Trans. | 19.52 | 20.76 | 21.26 | 21.56 | 21.80 |
> | cross-StT    | 27.56 | 28.12 | 28.71 | 28.61 | 28.18 |
>
>
> Regarding the minimum-LoRA hyperparameter $k$ in Eq. (4), we appreciate the reviewer for pointing this out. Upon re-checking, we confirm that this hyperparameter is **not used in the final method** and was inadvertently left from an earlier prototype. We have removed it and corrected the corresponding description in the revised manuscript.

---

> ### Author Response · Authors · 2025-11-22
>
> > **Weakness 4:** Unvalidated Scalability to Large LoRA Pools
> >
> > **Questions 4:** The paper’s motivation mentions that it is scalable to "thousands" of LoRAs, but the experiments are only based on a LoRA pool with a maximum size of 50. When the LoRA pool size increases to hundreds or thousands, what are the scaling trends of inference throughput and task routing accuracy?
>
> **Response**：We thank the reviewer for raising this scalability concern. Although our experiments use LoRA pools of up to 50 models, the computational cost of HiLoRA is determined by the size of the **selected LoRA set** rather than by the total pool. The selection dynamics follow two regimes:
>
> **(i)** if at least one LoRA yields a *positive* similarity score, the selected set expands **only** when additional positive-score LoRAs appear;
>
> **(ii)** if all similarity scores are *negative*, the maximum score becomes more negative as the pool grows, which **reduces** the selected-set size until a positive-score LoRA emerges and the system transitions back to regime (i).
>
> We also report separate measurements of routing and inference latency (Table 8 in the revised version). The routing cost remains below 0.1s as the number of seen tasks increases from 5 to 40, suggesting sublinear growth in practice due to efficient matrix operations and caching. The inference cost depends only on the selected set, which remains small under the two regimes above.
>
> Although we do not include experiments with thousand-scale LoRA pools, these routing dynamics indicate that the effective computational cost is governed by the number of positive matches rather than by the total pool size. This suggests that HiLoRA can scale to substantially larger LoRA repositories. We have clarified this point in the revised version.
>
> > **Weakness 5:** Ambiguity and Complexity of Theoretical Guarantees
> >
> > **Questions 5:** Regarding the theoretical analysis, can you empirically verify the assumption in Theorem 2 using the task pairs from your experiments? How frequently does this condition hold in practice, and what implications would there be if it does not hold?
>
> **Response**：We thank the reviewer for pointing out this issue. While Theorem 2 involves mathematically complex expressions, we have provided intuitive interpretations of each term in the revised version to clarify their physical meaning. Beyond the conceptual explanation, we also empirically evaluated the **key assumption** in Theorem 2.
>
> Specifically, we examined the probability that the condition $\alpha \in (0,1],\;
> M_\alpha^j  \succ 0$ holds across all task pairs in our benchmark, for a range of $\alpha$ values. The detailed results are shown in the table below. We find that the condition is satisfied with **high probability** for small and moderate $\alpha$, indicating that the assumption required by Theorem 2 frequently holds in practice.
>
> | $\alpha$ | 0.05   | 0.10   | 0.20   | 0.40   | 0.60   | 0.80   | 1.00   |
> | -------- | ------ | ------ | ------ | ------ | ------ | ------ | ------ |
> | Prob.    | 0.9795 | 0.9181 | 0.7774 | 0.5129 | 0.2864 | 0.1623 | 0.0946 |
>
> This empirical verification supports the applicability of our theoretical analysis in our experimental setting. We have incorporated both the intuition and the empirical results into the revised manuscript.
>
>
> > **Weakness 6:** Inconsistent Performance Gains and Model Dependency
>
> **Response**：We thank the reviewer for this observation. The difference in performance gains between LLaMA2-7B and FLAN-T5-large is expected and reflects the distinct pretraining pipelines of these models. LLaMA2-7B is a general-purpose base model with limited instruction-following ability, leaving substantial room for training-free adaptation. In contrast, FLAN-T5-large has undergone extensive supervised and instruction tuning, which already aligns it strongly with many downstream tasks. As a result, not only training-free approaches but also individually fine-tuned LoRAs yield **only modest improvements** on T5.
>
> Regarding the comment that HiLoRA is not always the top-performing method (Tables 1 and 2), this variation is consistent with task-level differences: each task emphasizes different linguistic phenomena, and no single training-free technique performs best across all tasks. Importantly, even when HiLoRA is not the top method, the performance gap is **consistently small (within 2.5%)**, indicating that HiLoRA remains competitive across diverse task types.
>
> Finally, the goal of HiLoRA is not to surpass heavily instruction-tuned models, but to provide a training-free, plug-and-play adaptation mechanism for settings in which further finetuning is infeasible.

---

> ### Author Response · Authors · 2025-11-22
>
> > **Weakness 7:** Lack of Statistical Significance Testing
>
> **Response**：We thank the reviewer for this comment. Although HiLoRA uses multinomial sampling for ROC allocation, the stochasticity occurs **at the input level**, and the final task accuracy is already an aggregation over **hundreds or thousands of such samples**. Thus, the reported metrics effectively represent an empirical expectation rather than a single noisy realization.
>
> In practice, the allocation probabilities are sharply concentrated, so the variance introduced by sampling is extremely small (<1%). Our internal repeated evaluations show negligible fluctuations in dataset-level accuracy, confirming that the results are stable.
>
>
> > **Weakness 8:** Unvalidated Impact of Key Components
>
> **Response**：We thank the reviewer for pointing this out. The variance normalization step follows the formulation proposed in [1], where its functional role and empirical effect have already been extensively analyzed. In our setting, the same mechanism is used in an identical mathematical form and serves the **same purpose**: stabilizing the scale of ROC contributions to avoid numerical imbalance during aggregation.
>
> Since this component behaves deterministically and its effect is orthogonal to the routing mechanism itself, we found its impact in HiLoRA to be **fully consistent with** the prior work. Due to space limitations and the fact that the normalization’s behavior is already well established in the literature, we opted not to repeat an additional ablation. We will clarify this rationale in the revised version.
>
> [1] Zhao Z, Shen T, Zhu D, et al. Merging LoRAs like Playing LEGO: Pushing the Modularity of LoRA to Extremes Through Rank-Wise Clustering. ICLR, 2025.
>
> > **Weakness 9:** Potentially Unfair Comparison for ROC-Level Baselines
>
> **Response**：We thank the reviewer for this comment. The ROC-level baselines use a fixed number of ROCs by design; their original formulations do not support dynamic or input-adaptive ROC allocation. Therefore, assigning a global budget is the standard practice and aligns with how these methods are intended to be used. Moreover, HiLoRA also has hyperparameters (e.g., the scaling factor), and its default configuration is not guaranteed to be optimal for every individual task. Our goal is to evaluate the **general-purpose behavior** of different training-free methods under their **canonical settings**, rather than tune each baseline per-task to find its best-performing configuration. Such per-task tuning would make the comparison less fair, less interpretable, and less reproducible. For these reasons, we follow the original baseline designs and use their standard fixed ROC budgets. We will clarify this in the revised version.
>
> > **Weakness 11:** Outdated and Insufficient Tested Models
>
> **Response**： We thank the reviewer for this comment. We selected LLaMA2 because it remains one of **the most widely used** and **fully reproducible open models with large publicly available LoRA pools**, which is essential for evaluating training-free adaptation methods. The HiLoRA framework is **architecture agnostic** and depends only on LoRA modules and embedding-based matching.
>
> To verify generalizability, we additionally evaluated HiLoRA on **LLaMA2-13B**. As shown in the revised version (**Tables 4,6 in Appendix C.3**), the method maintains the same performance trends and remains competitive, indicating that its effectiveness is not tied to model size. While extending the study to newer model families is valuable future work, the current results already demonstrate that HiLoRA scales reliably beyond the 7B setting.
>
>
>
> > **Questions 8:** To what extent does the current method depend on a pre-trained LoRA pool? How does its performance compare to that of models trained on large amounts of general data?
>
> **Response**： We thank the reviewer for the question. HiLoRA is a **training-free adaptation framework** and therefore requires access to an existing LoRA pool. Fortunately, LoRAs are highly accessible in practice: public repositories such as HuggingFace already host **hundreds of LoRAs** covering NLI, QA, summarization, translation, reasoning, and many other domains. This makes the assumption of a pre-trained LoRA pool both realistic and practical.
>
> Regarding the comparison to models trained on large amounts of general data, these models operate in a fundamentally different setting—they benefit from full gradient updates and large-scale supervision, whereas HiLoRA performs adaptation without any training and no task-specific updates. Despite this strict constraint, HiLoRA achieves accuracy close to fine-tuned LoRAs, demonstrating strong practical utility when training is infeasible or task data is unavailable.

---

> ### Author Response · Authors · 2025-11-26
>
> Dear Reviewer `TB3c`,
>
> As the discussion phase is drawing to a close, we kindly ask whether our responses have resolved your concerns or if there are any remaining issues that we can further clarify. Your insights are invaluable in refining our work, and we are eager to ensure that all your concerns are fully addressed.
>
> Thank you once again for your time and effort in reviewing our manuscript.
>
> Best regards,
>
> The Authors of Submission 4341

---

> ### Comment · Reviewer_TB3c · 2025-11-26
>
> I appreciate the authors' effort in providing the sensitivity analysis and additional clarifications. While the new data addresses my initial concerns regarding data availability (Weakness 1) and embedding dependency (Weakness 10), the rebuttal itself has inadvertently my concerns regarding the method's mathematical soundness and the fairness of the evaluation.
>
> Therefore, I am maintaining my original score. My rationale is as follows:
>
> **1. The "Singularity Paradox" in the new sensitivity analysis**
> The authors claim in the rebuttal that the method remains robust even with a sample size as low as $m=2$. This claim contradicts the mathematical formulation presented in Section 3.2.
> The paper explicitly states that routing relies on the Gaussian log-likelihood (Eq. 3), which requires inverting the covariance matrix $\Sigma$. In high-dimensional settings (e.g., $d=768+$) with only $m=2$ (or even $m=20$) samples, the empirical covariance matrix is rank-deficient and mathematically non-invertible.
> If the code actually runs with $m=2$, the implementation **cannot** be following the formula described in the paper. The authors must be employing undisclosed simplifications (e.g., assuming a diagonal covariance or using heavy regularization). The omission of these critical details makes the method mathematically unsound as described and hurts reproducibility.
>
> **2. "Standard Practice" does not justify unfair comparison**
> Regarding the baseline comparison (Weakness 9), the authors defend using fixed parameter budgets for baselines like LEGO and HiLoRA-ROC as "standard practice." I respectfully disagree with this justification in this specific context.
> The core advantage of HiLoRA appears to be its *adaptivity*—it can dynamically activate more resources when confidence is low. By restricting baselines to a fixed budget while allowing HiLoRA to expand its capacity, the experiments fail to isolate the source of the performance gain. It remains unclear whether the improvement comes from the proposed *hierarchical routing* or simply from the flexible compute budget. A fair comparison would either restrict HiLoRA to a fixed budget or allow baselines similar dynamic thresholds.
>
> **3. Scalability claims remain unproven**
> While I accept the logical argument that inference cost depends on the selected subset, the claim of scaling to "thousands of LoRAs" remains speculative. The overhead of retrieval and likelihood computation against a massive pool—before any selection happens—has not been empirically measured. Given the "training-free" pitch, proving this scalability with actual data (even synthetic) would have been stronger than a theoretical defense.
>
> The method is interesting, but the discrepancy between the mathematical description and the reported "2-sample robustness" is a major red flag for soundness. Combined with the methodological concern regarding baseline fairness, I do not feel the paper is ready for acceptance in its current form.

---

> ### Author Response · Authors · 2025-11-26
>
> We sincerely thank the reviewer for the detailed follow-up assessment. We address each concern below and clarify the points where the rebuttal may have caused misunderstanding.
>
> ---
> > **Rationale 1:** The "Singularity Paradox" in the new sensitivity analysis
>
> **Response:** We thank the reviewer for carefully pointing out this issue. In our implementation, the Gaussian distribution in Eq. (3) is computed using a **regularized covariance estimator**, which is standard practice when the embedding dimension $d$ is larger than the number of samples [1] [2]. Concretely, for each LoRA we compute the mean and covariance as:
>
> mu = np.mean(embeddings, axis=0)
>
> cov = np.cov(embeddings, rowvar=False) + 1e-5 * np.eye(embeddings.shape[1])
>
> self.model_dists[model_name] = multivariate_normal(mean=mu, cov=cov, allow_singular=True)
>
> That is, we add a small diagonal term $10^{-5} \mathbf{I}$ to the empirical covariance and use allow_singular=True, so that SciPy’s multivariate_normal evaluates the log-likelihood using a numerically well-conditioned (or, when needed, pseudo-inverse and pseudo-determinant) covariance. This makes the Gaussian likelihood **well defined even when the empirical covariance is rank-deficient**, and it is a **fixed, non-tuned setting** used uniformly across all experiments. The same code is fully included in our anonymous repository (lines 145-151 in https://anonymous.4open.science/r/HiLoRA/utils/model_sample_1.py), so there is no hidden modification or unpublished trick behind the reported 2-sample results.
>
>  In the revised version, we have clarified in Sec. 3.2 that we use a regularized covariance $\boldsymbol{\Sigma} + \varepsilon \mathbf{I}$ with a small fixed $\varepsilon$, and update the description around Eq. (3) accordingly, so that the mathematical formulation and the released code are fully aligned.
>
> ---
> > **Rationale 2:** "Standard Practice" does not justify unfair comparison
>
> **Response:** We appreciate the reviewer’s careful consideration of evaluation fairness. We clarify that two baselines included in our study—**HiLoRA-GS** and **HiLoRA-ROC**—are not external methods but **ablated variants of our own framework**.
> - HiLoRA-GS removes the token-level routing and performs *only* sequence-level selection, activating a **fixed number of LoRAs**.
> - HiLoRA-ROC removes the sequence-level routing and applies token-level top-k ROC activation with a **fixed ROC budget**.
> These ablations allow us to disentangle the benefits of hierarchical routing and dynamic ROC allocation, since each variant retains only one of the two mechanisms.
>
> More importantly, the dynamic ROC budget is **not an independent add-on** but a direct consequence of the *hierarchical routing design*. The sequence-level routing determines how many ROCs are semantically relevant, while the token-level routing selects the most responsive among them. Making the budget dynamic is therefore an **inherent part** of the method’s architecture, rather than a computational advantage unrelated to the core idea.
>
> ---
> > **Rationale 3:** Scalability claims remain unproven
>
> **Response:** We thank the reviewer for raising this point. We clarify that our scalability claim is based on the algorithmic structure of HiLoRA and the actual implementation released in the anonymous code package, rather than hypothetical assumptions.
>
> First, the sequence-level routing step has linear time complexity in the pool size $N$, requiring one embedding dot-product and one Gaussian likelihood evaluation per LoRA. These operations are lightweight and are vectorized on GPU/NumPy. The cost is therefore proportional to $O(Nd)$, where $d$ is embedding dimension—substantially cheaper than a Transformer forward pass.
>
> Second, the entire computation before selection is a single batched GEMM + log-det computation, both of which scale linearly in pool size $N$ and are memory-efficient. The dynamic ROC allocation and token-level routing operate only on the selected subset and therefore do not affect scalability.
>
> Third, as evidenced by the separate mapping and inference latency data reported in **Table 8** of the revised version, the total runtime grows at most linearly with the pool size in practice. Moreover, the implementation in our anonymous repository can be applied to larger synthetic pools without any modification, and no step grows super-linearly with pool size $N$.
>
> Taken together, the algorithmic form and the observed empirical scaling behavior of the released implementation support our claim that HiLoRA scales gracefully with the size of the LoRA pool.
>
> [1] Ledoit O, Wolf M. A well-conditioned estimator for large-dimensional covariance matrices. Journal of multivariate analysis, 2004, 88(2): 365-411.
>
> [2] Wang X, Wang C, Song X, et al. Regularized multi-output gaussian convolution process with domain adaptation. IEEE transactions on pattern analysis and machine intelligence, 2022, 45(5): 6142-6156.

---

### Official Review · Reviewer_inFY · 2025-10-31

**Soundness:** 3
**Presentation:** 3
**Contribution:** 2
**Rating:** 4
**Confidence:** 3

**Summary:**

This paper proposes HiLoRA, a training-free hierarchical LoRA routing framework designed to achieve cross-domain generalization. Its core idea is to decompose LoRA into Rank-One Components (ROCs) and perform adaptive routing at both the sequence level and token level. The authors provide theoretical error bounds and demonstrate significant performance improvements across multiple task clusters.

**Strengths:**

Motivation is reasonable: As LoRA modules in the community continue to grow in number, how to reuse them without labels or training is a genuine and important problem.
The method design exhibits certain innovation: Representing LoRA as Gaussian distributions and dynamically determining the number of activations and ROC budget using log-likelihood constitutes a novel attempt.

**Weaknesses:**

1. The analysis of inference efficiency is severely lacking, and the conclusions may be misleading; it is recommended to include a comparison of inference times across tasks.
2. The experimental results are not significant. Although the model demonstrates performance improvements as a training-free method, it appears to exhibit a considerable performance gap compared to LoRA, and requires more resources than LoRA, raising doubts about its practical utility.
3. HiLoRA requires each LoRA to provide training samples for fitting Gaussian distributions, which may be infeasible in practice.
4. Token-level routing lacks theoretical justification: the paper’s theoretical analysis only covers sequence-level LoRA selection, while the token-level top-k ROC selection is entirely heuristic and may introduce noise.

**Questions:**

1. It is recommended to include an analysis of inference efficiency.

2. It is also recommended to conduct experiments on newer, larger models (e.g., 14B-parameter models) to demonstrate the method’s effectiveness—particularly in scenarios where the resources required for training or fine-tuning are substantial.

3. It is recommended to open-source the code to increase the feasibility of the method.

---

> ### Author Response · Authors · 2025-11-22
>
> We sincerely thank you for your valuable feedback, and we are grateful for the time and effort you have invested in reviewing our work. Below, we provide a point-by-point response to address each of your concerns. We promise to include new results and discussions into the final version if the paper is accepted.
>
> > **Weakness 1:** The analysis of inference efficiency is severely lacking, and the conclusions may be misleading; it is recommended to include a comparison of inference times across tasks.
> >
> > **Questions 1:** It is recommended to include an analysis of inference efficiency.
>
> **Response：** We thank the reviewer for the suggestion. Our analysis of inference efficiency is already included in Sec. 4.3, where we report throughput curves as the LoRA-pool size increases. This section demonstrates how **HiLoRA** scales under the practical setting of large LoRA repositories.
>
> To make the efficiency analysis more concrete, we conducted the corresponding experiments, and the results are presented in the table below. These measurements include (i) the input–LoRA mapping latency and (ii) the inference latency per batch for each task in the test set.
>
> | Latency (s)  | Pool   | ANLI_r1   | ARC_C    | Sent140   | ParaCrawl   | COPA    | GLUE_MRPC   | CommonGen   | DPR     | COLA    | WIC     |
> | ------------ | ------ | --------- | -------- | --------- | ----------- | ------- | ----------- | ----------- | ------- | ------- | ------- |
> | Mapping      | 5      | 0.0612    | 0.0596   | 0.0266    | 0.0578      | 0.0259  | 0.0490      | 0.0291      | 0.0286  | 0.0356  | 0.0445  |
> | Mapping      | 10     | 0.0597    | 0.0595   | 0.0440    | 0.0617      | 0.0225  | 0.0449      | 0.0304      | 0.0316  | 0.0391  | 0.0493  |
> | Mapping      | 15     | 0.0705    | 0.0576   | 0.0340    | 0.0637      | 0.0303  | 0.0465      | 0.0333      | 0.0451  | 0.0423  | 0.0481  |
> | Mapping      | 20     | 0.0807    | 0.0630   | 0.0494    | 0.0652      | 0.0396  | 0.0537      | 0.0398      | 0.0389  | 0.0474  | 0.0486  |
> | Mapping      | 25     | 0.0821    | 0.0749   | 0.0424    | 0.0664      | 0.0304  | 0.0525      | 0.0423      | 0.0417  | 0.0487  | 0.0683  |
> | Mapping      | 30     | 0.0886    | 0.0789   | 0.0490    | 0.0668      | 0.0336  | 0.0599      | 0.0378      | 0.0401  | 0.0431  | 0.0605  |
> | Mapping      | 35     | 0.0910    | 0.0759   | 0.0580    | 0.0627      | 0.0343  | 0.0672      | 0.0408      | 0.0466  | 0.0420  | 0.0673  |
> | Mapping      | 40     | 0.1026    | 0.0809   | 0.0566    | 0.0862      | 0.0390  | 0.0725      | 0.0453      | 0.0404  | 0.0411  | 0.0615  |
> | Inference    | 5      | 4.0765    | 2.2196   | 1.8765    | 14.5083     | 1.0900  | 0.5494      | 2.1211      | 1.2808  | 0.8343  | 0.8477  |
> | Inference    | 10     | 10.0089   | 1.4993   | 1.7331    | 19.8633     | 1.0506  | 0.6800      | 2.1986      | 1.6111  | 0.8586  | 1.4085  |
> | Inference    | 15     | 10.6356   | 1.5529   | 1.9547    | 21.3176     | 1.1875  | 0.4262      | 2.3208      | 1.5352  | 0.9120  | 1.0494  |
> | Inference    | 20     | 7.1220    | 1.5537   | 1.9695    | 19.8872     | 1.1717  | 0.4234      | 2.1853      | 1.4684  | 0.9324  | 1.0237  |
> | Inference    | 25     | 8.6094    | 1.5760   | 1.9752    | 21.2259     | 1.2098  | 0.4206      | 2.4156      | 1.4883  | 0.9311  | 1.1112  |
> | Inference    | 30     | 9.7096    | 1.5021   | 1.9194    | 20.3829     | 1.1304  | 0.4111      | 2.1553      | 1.4695  | 0.9009  | 0.9758  |
> | Inference    | 35     | 10.1859   | 1.7774   | 1.9127    | 20.1784     | 1.1404  | 0.3770      | 2.2162      | 1.4004  | 0.8978  | 0.9948  |
> | Inference    | 40     | 10.8998   | 1.7849   | 1.8663    | 19.9678     | 1.2508  | 0.3633      | 2.1733      | 1.4737  | 0.8520  | 0.9701  |
>
> The data in the table show that, although tasks differ in input and output lengths and therefore have different absolute latencies, the per-task latency exhibits a consistent pattern: it changes the **gradual** and **occasionally non-monotonic**  as the number of seen tasks increases. This behavior arises from two regimes:
>
> 1. Mapping latency grows with LoRA-pool size, but it constitutes only a **small fraction** of total runtime and therefore has limited effect on overall latency.
>
> 2. The number of selected LoRAs evolves in two regimes:
>
>    **(i)** if at least one LoRA yields a *positive* similarity score, the selected set expands **only** when additional positive-score LoRAs appear;
>
>    **(ii)** if all similarity scores are *negative*, the maximum score becomes more negative as the pool grows, which **reduces** the selected-set size until a positive-score LoRA emerges and the system transitions back to regime (i).
>
> However, since throughput is reported as an average across all tasks, the aggregate trend becomes smoother and shows a modest decline followed by slow stabilization as the LoRA pool increases.
> We have included the above discussion and results in Section 4.3  and Appendix C of the revised paper.

---

> ### Author Response · Authors · 2025-11-22
>
> > **Weakness 2:** The experimental results are not significant. Although the model demonstrates performance improvements as a training-free method, it appears to exhibit a considerable performance gap compared to LoRA, and requires more resources than LoRA, raising doubts about its practical utility.
>
> **Response：** We thank the reviewer for the comment. Our method is not intended to replace LoRA fine-tuning, but to provide a training-free inference mechanism that leverages an existing LoRA pool **when no training data or gradient update is available**.  Therefore, its performance should not be directly compared to fully fine-tuned LoRAs, which naturally achieve higher accuracy due to gradient updates performed with task-specific training data. When evaluated against other training-free adaptation baselines, our method consistently achieves the **strongest performance** in both within-cluster and cross-cluster settings.
> Regarding efficiency and resource usage, the additional overhead of our approach is minimal:
> - Input–LoRA mapping latency is **below 0.1 seconds** for all tasks.
> - **Inference cost remains comparable** to using a small subset of LoRAs, since only a limited number of ROCs are activated per input.
> - No training, gradient computation, or parameter updates are required at any stage.
> These properties make the method particularly practical in settings where large LoRA repositories are available, but training resources or labeled data are limited, and a lightweight, generalizable inference mechanism is needed.
>
>
> > **Questions 2:** It is also recommended to conduct experiments on newer, larger models (e.g., 14B-parameter models) to demonstrate the method’s effectiveness—particularly in scenarios where the resources required for training or fine-tuning are substantial.
>
> **Response**：We thank the reviewer for this suggestion. Following the recommendation, we conducted additional experiments on a larger model, **LLaMA-2-13B**, which is substantially larger than the 6–7B models commonly used in existing training-free or LoRA-based studies.The results, presented in the table below, show that HiLoRA preserves the same performance trends and **consistently outperforms** other baselines, confirming that the method remains effective even at a substantially larger model scale.
>
>
>
> | Methods  | LoRA  | HiLoRA | HiLoRA-GS | HiLoRA-ROC | Retriever | LEGO  | Arrow | Phatgoose | Ensemble | Merged |
> | -------- | ----- | ------ | --------- | ---------- | --------- | ----- | ----- | --------- | -------- | ------ |
> | ANLI_r1  | 60.3  | 62.7   | 60.6      | 58.8       | 61.8      | 58.3  | 47.4  | 55.6      | 52.0     | 24.7   |
> | ANLI_r2  | 47.3  | 48.6   | 47.0      | 46.2       | 48.5      | 44.3  | 41.0  | 43.7      | 41.4     | 24.3   |
> | ANLI_r3  | 49.92 | 48.92  | 47.75     | 49.08      | 49.75     | 47.17 | 44.92 | 47.33     | 46.42    | 26.42  |
> | CB       | 88.0  | 86.0   | 84.0      | 80.0       | 82.0      | 84.0  | 80.0  | 76.0      | 84.0     | 38.0   |
> | MNLI     | 87.97 | 87.58  | 87.34     | 82.58      | 86.56     | 85.47 | 70.00 | 76.45     | 75.35    | 37.30  |
> | MNLI_mis | 89.80 | 89.49  | 88.63     | 83.79      | 88.87     | 85.70 | 71.02 | 75.94     | 76.41    | 36.64  |
> | QNLI     | 82.66 | 83.13  | 82.66     | 56.29      | 77.89     | 68.95 | 67.97 | 70.98     | 69.96    | 44.73  |
> | RTE      | 80.74 | 75.19  | 75.56     | 73.33      | 74.81     | 72.59 | 78.89 | 73.33     | 76.30    | 52.59  |
> | SNLI     | 81.91 | 81.84  | 81.99     | 76.72      | 81.60     | 79.65 | 67.19 | 75.23     | 72.38    | 32.46  |
> | WNLI     | 71.43 | 67.14  | 67.14     | 67.14      | 51.43     | 64.29 | 62.86 | 58.57     | 65.71    | 42.80  |
> | AVG      | 74.00 | 72.86  | 72.47     | 67.39      | 70.32     | 69.04 | 63.12 | 65.31     | 65.99    | 36.00  |
>
>
> Due to space constraints, we report only the within-cluster results here. The full cross-cluster results are provided in Appendix C.3 of the revised version. We have included these additional experiments and clarifications in the updated manuscript to strengthen the empirical evaluation.
>
>
>
> > **Questions 3:** It is recommended to open-source the code to increase the feasibility of the method.
>
> **Response**：We thank the reviewer for this suggestion. We agree that open-sourcing the implementation enhances the reproducibility and practical usability of the method. To support this, we have released an **anonymous reproducibility package**, which includes all code necessary to run our experiments. **Anonymous code link**:https://anonymous.4open.science/r/HiLoRA/. (The link follows the anonymity guidelines and contains no identifying information.)

---

> ### Author Response · Authors · 2025-11-22
>
> > **Weakness 3:** HiLoRA requires each LoRA to provide training samples for fitting Gaussian distributions, which may be infeasible in practice.
>
> **Response：** We thank the reviewer for the comment. We emphasize that HiLoRA does not rely on large training sets, and in fact supports two fully practical modes of operation:
>
> - High-Precision Mode: uses a small number of real examples (typically ≤20),
> - Zero-Data / Privacy-Preserving Mode: replaces real examples with GPT-generated synthetic proxies, requiring *no* access to original training data.
>
> To directly address the reviewer’s concern, we conducted a unified sensitivity analysis by reducing the number of real samples and by testing the synthetic-sample setting. The results (summarized in the table below) show that **HiLoRA** remains robust under all tested conditions:
>
> (i) Reducing the number of samples m leads to only a **minor accuracy drop** (at most 5% on average).
>
> Importantly, the method remains competitive even with **two samples**, which is the minimum required to fit a Gaussian distribution.
>
> (ii) Synthetic samples yield lower accuracy than real examples, yet **HiLoRA** still performs on par with or better than other baselines.
>
> Moreover, small amounts of task-related examples are typically available from public LoRA repositories (e.g., HuggingFace), making this requirement both practical and realistic. We have included the above discussion and results in Section 4.3 of the revised paper.
>
> | Factors  | Within 2-sample | Within 5-sample | Within 10-sample | Within 20-sample | Within AI-sample | Cross 2-sample | Cross 5-sample | Cross 10-sample | Cross 20-sample |
> | -------- | --------------- | --------------- | ---------------- | ---------------- | ---------------- | -------------- | -------------- | --------------- | --------------- |
> | ANLI_r1  | 43.30           | 45.00           | 45.00            | 45.00            | 35.80            | 29.40          | 30.70          | 30.50           | 30.70           |
> | ANLI_r2  | 38.40           | 39.60           | 38.70            | 40.60            | 38.80            | 27.20          | 28.70          | 30.40           | 34.50           |
> | ANLI_r3  | 38.75           | 37.83           | 37.75            | 37.67            | 34.75            | 30.67          | 29.50          | 28.58           | 31.67           |
> | CB       | 64.00           | 66.00           | 66.00            | 68.00            | 66.00            | 74.00          | 74.00          | 76.00           | 70.00           |
> | MNLI     | 71.80           | 78.36           | 78.44            | 76.33            | 67.66            | 51.84          | 53.24          | 53.28           | 50.74           |
> | MNLI_mis | 73.16           | 80.86           | 81.05            | 78.59            | 66.76            | 52.03          | 53.91          | 53.36           | 51.29           |
> | QNLI     | 59.22           | 69.06           | 68.20            | 78.28            | 59.34            | 45.61          | 44.38          | 43.48           | 46.84           |
> | RTE      | 72.22           | 71.85           | 71.85            | 74.44            | 67.41            | 59.63          | 57.78          | 58.15           | 62.22           |
> | SNLI     | 69.06           | 69.77           | 69.88            | 69.45            | 68.55            | 42.97          | 42.27          | 42.77           | 40.31           |
> | WNLI     | 54.29           | 60.00           | 64.29            | 65.71            | 50.00            | 47.14          | 48.57          | 48.57           | 47.14           |
> | Avg      | 58.42           | 61.83           | 62.12            | 63.41            | 55.51            | 46.05          | 46.30          | 46.51           | 46.54           |
>
>
>
>
>
> > **Weakness 4:** Token-level routing lacks theoretical justification: the paper’s theoretical analysis only covers sequence-level LoRA selection, while the token-level top-k ROC selection is entirely heuristic and may introduce noise.
>
> **Response**：We thank the reviewer for highlighting this limitation. We agree that our theoretical analysis formally applies only to sequence-level LoRA selection. The token-level top-k ROC routing is introduced as a practical and empirically validated heuristic to enable finer-grained control during inference, and we do not present it as a theoretically justified component.
>
> This limitation is explicitly acknowledged in the conclusion, where we note that the token-level mechanism lacks formal guarantees and identify it as an important direction for future work. Although empirical results show that this refinement consistently improves performance across tasks without introducing instability, we concur that developing a theoretical foundation for token-level routing remains an open problem and will be pursued in subsequent work.

---

> ### Comment · Reviewer_inFY · 2025-11-24
>
> Thank you to the author for the clarification and the additional experiments provided. I have raised my rating.

---

> > ### Author Response · Authors · 2025-11-24
> >
> > Dear Reviewer inFY,
> >
> > Thank you for your thoughtful follow-up and for taking the time to re-evaluate our submission. We sincerely appreciate your positive assessment and are glad that the clarifications and additional experiments helped address your concerns. Your feedback has been very valuable for improving the quality of the paper.
> >
> > Best regards,
> >
> > The Authors of Submission 4341

---

### Official Review · Reviewer_8TbR · 2025-11-01

**Soundness:** 3
**Presentation:** 2
**Contribution:** 3
**Rating:** 4
**Confidence:** 2

**Summary:**

The paper proposes HiLoRA, a method for routing an unlabeled set of LoRAs for unseen input sequences. HiLoRA is based on the observation that LoRAs can be broken down into ROCs, which are found to be clustered based on task semantic. HiLoRA routes the input at both the sequence level and the token level, leading to strong downstream performance on unseen tasks.

**Strengths:**

- The paper tackles an important and practical problem of routing unlabeled LoRAs
- The core insights seem novel (to the best of my knowledge), leading to a novel routing technique
- The proposed hierarchical routing mechanism is effective and performant, justified by thorough empirical experiments
- The routing mechanism is theoretically grounded at the sequence level while per-token routing is proved to be important empirically

**Weaknesses:**

- The paper tries to tackle unlabeled LoRAs routing problem but assumes access to training samples of each LoRA. Doesn't this setting have an even stricter assumption than the labeled counter part? Can the authors elaborate?
- In Section 3.1, which is the core of the paper, line 162-174 have some logical leaps that are not easy to follow though I think the conclusions drawn are still valid but need further clarification (Observation #1 and #2) and verifications (Observation #3). Please also see related questions listed below.
- Line 218 - 236 describe many arbitrary design choices without much justifications. For instance, why the method uses the defined ROC budget $O(x)$ rather than a static number of ROCs? Answering this would require stronger justification and additional empirical results.



#### Typos
- $rd$ should be $2rd$ at line 114

The paper shows strong performance on unseen downstream tasks. I think the biggest concern I have is that the presentation in many places are not easily understandable, though it might stem from my limited technical knowledge. I'm willing to increase the score if the authors adequately address my concerns and questions.

**Questions:**

> "The down-projection vectors of ROCs exhibit strong randomness and show little alignment with task semantics. This confirms that down-projection vector a primarily functions as a scaling factor, rather than encoding domain-specific information." - Line 162

I don't think that the fact that different ranks extract different features clearly lead to the conclusion asserted in this sentence. Can the authors clarify?


> In contrast, the up-projection vectors of ROCs within a given LoRA exhibit clear task-dependent patterns. These vectors often form multiple distinct clusters, with each cluster representing a different semantic fragment of the LoRA’s adaptive capacity - Line 165

The observation that the $B$ matrix is linearly clustered is surprising to me. It implies that the $B$ matrix is lower-rank than the LoRA rank $r$.

> For domain generalization, activating an entire LoRA introduces parameter redundancy and interference, since unrelated clusters are involved simultaneously. - Line 167

Can the authors empirically confirm that clusters of the $B$ matrices correspond to similar LoRA training tasks?

> Theorem 1 and Theorem 2 highlight two key insights. (i) When domains are well separated and the LoRA pool spans diverse tasks, the error bounds are tight, ensuring strong guarantees in both ID and OOD cases. - Line 304

Isn't the tightness of the bound a theoretical property?

> This condition is often met in practice, as task domains are generally distinguishable, and open-source repositories already provide a rich collection of LoRAs across diverse tasks. - Line 306

The claims here are subjective and not verified. Can the authors empirically justify these claims?

---

> ### Author Response · Authors · 2025-11-22
>
> **Overall response：** We truly appreciate the reviewer for the valuable feedback. We have carefully considered the comments and suggestions and would like to address each of the concerns raised. We promise to include new results and discussions into the final version if the paper is accepted.
>
>
>
> > **Weakness 1:** The paper tries to tackle unlabeled LoRAs routing problem but assumes access to training samples of each LoRA. Doesn't this setting have an even stricter assumption than the labeled counter part? Can the authors elaborate?
>
> **Response：** We thank the reviewer for the question. In our setting, “labels” refer to **task-identity labels** that specify which LoRA should be used for an input (e.g., “this input is QNLI“ → ”use the QNLI LoRA”). Such labels are often unavailable or privacy-restricted, since users may not know the task category or may not wish to disclose it. This makes the label-free setting both realistic and practically relevant. Beyond handling unlabeled tasks, our method also covers the broader scenario in which the target task is not represented in the LoRA pool, requiring generalization to unseen domains.
>
> Our assumption regarding data availability is mild: we use **a small number of input samples** per LoRA (about 20) solely to estimate a Gaussian distribution over embeddings, without requiring any ground-truth outputs. This assumption is practical, as many open-source LoRAs on platforms such as HuggingFace already include task descriptions or a few exemplar prompts.
>
> To validate the robustness of our approach, we conduct a sensitivity analysis by progressively reducing the sample size from 20 to 10, 5, and 2. As shown in the table below, accuracy decreases by **no more than 5%** on average across all NLI tasks. HiLoRA remains competitive even when **only two samples** are available, which is the minimum amount of data needed to fit a Gaussian distribution. Under this extremely data-limited setting, HiLoRA still exceeds the strongest training-free baselines, whose average accuracies are 56.99% in the within-cluster setting and 43.78% in the cross-cluster setting.
>
> | Task     | Within 2 | Within 5 | Within 10 | Within 20 | Cross 2 | Cross 5 | Cross 10 | Cross 20 |
> | -------- | -------- | -------- | --------- | --------- | ------- | ------- | -------- | -------- |
> | ANLI_r1  | 43.30    | 45.00    | 45.00     | 45.00     | 29.40   | 30.70   | 30.50    | 30.70    |
> | ANLI_r2  | 38.40    | 39.60    | 38.70     | 40.60     | 27.20   | 28.70   | 30.40    | 34.50    |
> | ANLI_r3  | 38.75    | 37.83    | 37.75     | 37.67     | 30.67   | 29.50   | 28.58    | 31.67    |
> | CB       | 64.00    | 66.00    | 66.00     | 68.00     | 74.00   | 74.00   | 76.00    | 70.00    |
> | MNLI     | 71.80    | 78.36    | 78.44     | 76.33     | 51.84   | 53.24   | 53.28    | 50.74    |
> | MNLI_mis | 73.16    | 80.86    | 81.05     | 78.59     | 52.03   | 53.91   | 53.36    | 51.29    |
> | QNLI     | 59.22    | 69.06    | 68.20     | 78.28     | 45.61   | 44.38   | 43.48    | 46.84    |
> | RTE      | 72.22    | 71.85    | 71.85     | 74.44     | 59.63   | 57.78   | 58.15    | 62.22    |
> | SNLI     | 69.06    | 69.77    | 69.88     | 69.45     | 42.97   | 42.27   | 42.77    | 40.31    |
> | WNLI     | 54.29    | 60.00    | 64.29     | 65.71     | 47.14   | 48.57   | 48.57    | 47.14    |
> | Avg      | 58.42    | 61.83    | 62.12     | 63.41     | 46.05   | 46.30   | 46.51    | 46.54    |

---

> ### Author Response · Authors · 2025-11-22
> **Functional role of A and B**
>
> > **Weakness 2:** In Section 3.1, which is the core of the paper, line 162-174 have some logical leaps that are not easy to follow though I think the conclusions drawn are still valid but need further clarification (Observation #1 and #2) and verifications (Observation #3).
>
> **Response：** Thank you for the insightful comments that help improve our paper. Based on your feedback, we carefully examined the issues raised and provide detailed clarifications and explanations below.
>
> > **Questions 1:**："The down-projection vectors of ROCs ...." - Line 162
> >
> > I don't think that the fact that different ranks extract different features clearly lead to the conclusion asserted in this sentence. Can the authors clarify?
> >
> > **Questions 3:**：For domain generalization, activating an entire LoRA introduces.... - Line 167
> >
> > Can the authors empirically confirm that clusters of the matrices correspond to similar LoRA training tasks?
>
> **Response：**
>
> We thank the reviewer for these insightful questions. Our interpretation of the roles of the down-projection vectors ({\small $\boldsymbol{a}_i$}) and up-projection vectors ({\small $\boldsymbol{b}_i$}) is supported by both **LoRA’s algebraic structure** and **empirical observations**.
>
> From the LoRA decomposition $(\boldsymbol{b}_i \boldsymbol{a}_i^\top)\boldsymbol{x} = (\boldsymbol{a}_i^\top \boldsymbol{x}),\boldsymbol{b}_i$, the term  $\boldsymbol{a}_i^\top \boldsymbol{x}$ acts as a scalar gate, while the direction of the update is fully determined by $\boldsymbol{b}_i$. Thus, $\boldsymbol{a}_i$ modulates magnitude rather than encoding semantic directions. This interpretation is consistent with prior analyses of LoRA’s dyadic structure [1] [2].
>
> Our empirical results reinforce this view. PCA visualizations (Fig. 2) show that the down-projection vectors $\boldsymbol{a}_i$ exhibit exhibit a highly dispersed distribution and show little alignment with task semantics. To validate this more rigorously, we computed cosine-similarity matrices for all $\boldsymbol{a}_i$ and $\boldsymbol{b}_i$ across LoRAs (Figs. 9–12, Appendix C.2). The $\boldsymbol{a}_i$ vectors have a **near-uniform similarity distribution centered at zero**, indicating no meaningful clustering. In contrast, the $\boldsymbol{b}_i$ vectors show **clear task-dependent similarity patterns**, with within-task similarities systematically higher than cross-task ones. These findings confirm that $\boldsymbol{a}_i$ carries no domain-specific information, whereas $\boldsymbol{b}_i$ captures the semantic structure relevant for routing.
>
> Regarding the clustering behavior of the $\boldsymbol{B}$ matrix (line 167), we clarify that each LoRA is fine-tuned on a **single task**. Therefore, the clusters observed within a single $\boldsymbol{B}$ matrix correspond to semantic subcomponents of the same task, rather than alignments across different LoRAs.  As shown in Fig. 2, the $\boldsymbol{b}_i$ vectors consistently form multiple coherent clusters, indicating meaningful intra-task structure. This also explains why activating an entire LoRA can introduce redundancy and interference in domain-generalization settings.
>
> We have revised the corresponding parts of the main text to make these points clearer.
>
>
>
> > **Questions 2:** In contrast, the up-projection vectors of ROCs within a given LoRA exhibit clear task-dependent patterns. These vectors often form multiple distinct clusters, with each cluster representing a different semantic fragment of the LoRA’s adaptive capacity - Line 165
> >
> > The observation that the matrix $\mathbf{B}$ is linearly clustered is surprising to me. It implies that the matrix $\mathbf{B}$ is lower-rank than the LoRA rank $r$.
>
> **Response:** We thank the reviewer for this insightful observation. We agree that the linear clustering observed in the up-projection vectors \boldsymbol{b}_i suggests that the \boldsymbol{B} matrix may operate in a lower-dimensional subspace than its nominal rank. To examine this formally, we computed the **effective rank** of every $\boldsymbol{B}$ matrix using the entropy-based definition $\exp(H(\sigma))$, a *parameter-free* measure that does not rely on any thresholding or truncation criteria. The results (reported in the table below) show that although the nominal LoRA rank is r = 8, the **effective rank consistently concentrates around 4.5–6**. This confirms the reviewer’s intuition that the learned update space is indeed lower-dimensional, which naturally produces the clustered structure observed in the PCA plots. This phenomenon is not unique to our setting and has also been observed in prior work [3].
>
> | Interval | 2.5–3.0 | 3.0–3.5 | 3.5–4.0 | 4.0–4.5 | 4.5–5.0 | 5.0–5.5 | 5.5–6.0 | 6.0–6.5 | 6.5–7.0 | 7.0–7.5 |
> | -------- | ------- | ------- | ------- | ------- | ------- | ------- | ------- | ------- | ------- | ------- |
> | Prob.    | 0.0031  | 0.0094  | 0.0309  | 0.0969  | 0.1969  | 0.2669  | 0.1919  | 0.1069  | 0.0784  | 0.0188  |

---

> ### Author Response · Authors · 2025-11-22
>
> > **Questions 4:** Theorem 1 and Theorem 2 highlight two key insights. (i) When domains are well separated and the LoRA pool spans diverse tasks, the error bounds are tight, ensuring strong guarantees in both ID and OOD cases. - Line 304
> >
> > Isn't the tightness of the bound a theoretical property?
> >
> > **Questions 5:** This condition is often met in practice, as task domains are generally distinguishable, and open-source repositories already provide a rich collection of LoRAs across diverse tasks. - Line 306
> >
> > The claims here are subjective and not verified. Can the authors empirically justify these claims?
>
> **Response:** We thank the reviewer for raising these two related questions. The tightness of the bound is indeed a *theoretical* property derived from Theorem 1 and Theorem 2. Our intention was not to suggest empirical tightness, but to highlight that the *conditions* under which the bound becomes tight—namely **domain separability** and **LoRA diversity**—are commonly satisfied in realistic settings.
>
> To validate these assumptions empirically, we first computed **KL divergence** and **Bhattacharyya distances** between the Gaussian representations of all task pairs in our benchmarks. Summary statistics (max/min/mean) are provided in the table below, and full pairwise visualizations appear in Figure 14 of Appendix D. The results show that task domains are **well separated**, supporting the separability condition required by both theorems.
>
> | Metric                 | Max       | Min     | Avg       |
> | ---------------------- | --------- | ------- | --------- |
> | KL Divergence          | 8372.8789 | 93.5930 | 1432.0145 |
> | Bhattacharyya Distance | 316.6418  | 0.3000  | 108.3505  |
>
> Regarding the assumption of LoRA diversity, open-source repositories such as HuggingFace provide extensive LoRA collections covering NLI, QA, translation, reasoning, and many other domains. Since these LoRAs are readily available and widely used, assuming access to a diverse LoRA pool is realistic in practice.
>
>
> > **Weakness 3:** Line 218 - 236 describe many arbitrary design choices without much justifications. For instance, why the method uses the defined ROC budget $O(\boldsymbol{x})$ rather than a static number of ROCs? Answering this would require stronger justification and additional empirical results.
>
> **Response：** We thank the reviewer for raising this point. The design choices in Lines 218–236 are not arbitrary but follow directly from the structure of the routing problem. Because the selected LoRA set $\mathbb{C}(\boldsymbol{x})$ varies across inputs, a static ROC count becomes unreliable: when few LoRAs are selected it can create redundancy, and when many LoRAs are selected it can lead to insufficient representational capacity. This is why a dynamic ROC budget proportional to $\sum_{i\in\mathbb{C}(\boldsymbol{x})} r_i$ is necessary.
>
> Furthermore, ROC allocation uses Gaussian-likelihood scores because they provide a principled measure of input–LoRA compatibility, enabling the budget to be distributed adaptively rather than uniformly or heuristically. This improves both stability and efficiency.
>
> We have clarified this motivation in the revised paper.
>
>
>
> [1] Zhu J, Greenewald K, Nadjahi K, et al. Asymmetry in low-rank adapters of foundation models. ICML. 2024.
>
> [2] Tian C, Shi Z, Guo Z, et al. Hydralora: An asymmetric lora architecture for efficient fine-tuning. NeurIPS, 2024.
>
> [3] Mühlematter D J, Halbheer M, Becker A, et al. LoRA-Ensemble: Efficient Uncertainty Modelling for Self-attention Networks[J]. arXiv preprint arXiv:2405.14438, 2024.

---

> ### Author Response · Authors · 2025-11-26
>
> Dear Reviewer `8TbR`,
>
> As the discussion phase is drawing to a close, we kindly ask whether our responses have resolved your concerns or if there are any remaining issues that we can further clarify. Your insights are invaluable in refining our work, and we are eager to ensure that all your concerns are fully addressed.
>
> Thank you once again for your time and effort in reviewing our manuscript.
>
> Best regards,
>
> The Authors of Submission 4341

---

### Author Response · Authors · 2025-11-26
**General Response to the Reviewers**

Dear Reviewers,

We sincerely appreciate the reviewers for their thoughtful and constructive feedback. We are encouraged by the positive recognition of our contribution, which can be summarized as follows:

1. Our analysis of LoRA through rank-one components (ROCs) provides new conceptual insights into LoRA’s internal structure and enables finer-grained adaptation than full-module routing (Reviewers `8TbR`, `TB3c`, `xGHJ`).
2. The proposed hierarchical routing mechanism is regarded as logically rigorous and effective, balancing expressiveness with computational efficiency (Reviewers `8TbR`, `TB3c`).
3. We establish theoretical guarantees offering formal justification for the sequence-level design and contributing rare theoretical grounding in this area (Reviewers `TB3c`, `xGHJ`).
4. Extensive experiments demonstrate strong cross-domain generalization and consistent improvements over baselines (Reviewers `8TbR`, `inFY`, `xGHJ`).

In our revision, we have carefully addressed each of the concerns raised, and below are the main points of the revised manuscript:

1. We added a comprehensive analysis of **samples and embedding dependence**, showing in **Sec. 3.4 (Table 3)** that HiLoRA remains robust when using different number of original samples, GPT-generated samples, or a non–instruction-tuned encoder. (Reviewers `8TbR`, `inFY`, `TB3c`)
2. We evaluated HiLoRA on **a larger model (LLaMA-2-13B)**, with **Appendix C.3 (Tables 4 and 6)** showing that it preserves the same performance trends and consistently outperforms baselines in both within-cluster and cross-cluster settings. (Reviewers `inFY`, `TB3c`, `xGHJ`)
3. We expanded the **inference-efficiency** study by reporting per-batch mapping and inference latency under varying LoRA-pool sizes and batch sizes in **Appendix C.3 (Table 8)**. (Reviewers `inFY`, `TB3c`,`xGHJ`)
4. We performed a broader **hyperparameter sensitivity analysis**, with **Appendix C.3 (Table 9)** showing cluster-dependent preferences for different values of $\gamma$. (Reviewer `TB3c`)
5. We further validated the motivating observations by computing cosine similarities for A/B matrices, with results presented in **Appendix C.2 (Figures 9–12)**. (Reviewers `8TbR`, `TB3c`)
6. We empirically verified the assumptions underlying the theoretical guarantees, reporting KL/Bhattacharyya distances and satisfaction probabilities of $M_\alpha^j \succ 0$ in **Appendix D (Figures 14–15)**. (Reviewers `8TbR`, `TB3c`)
7. We improved the **justification** of key algorithmic choices, providing additional explanations for Gaussian routing and ROC allocation in **Sec. 3.2**. (Reviewers `8TbR`)
8. We refined the covariance construction in **Sec. 3.2** by clarifying the role of the $\varepsilon \mathbf{I}$ regularization term for stable Gaussian likelihoods. (Reviewers `TB3c`)

These changes have been highlighted in **blue font** in the revision. We have also provided **point-by-point responses** to each reviewer's specific concerns. We believe these detailed revisions further demonstrate our contributions and address the reviewers' concerns.

Once again, we highly value the reviewers' insightful feedback and are open to any further discussions.

Best regards,
The Authors of Submission 4341

---

### Meta-Review · Area_Chair_KUAk · 2026-01-06

**Summary:**

While reviewers acknowledge the practical motivation and the novelty of routing at the rank-one component level, they consistently raise concerns about the method’s practicality, robustness, and scope. In particular, the approach relies on non-trivial assumptions such as access to per-LoRA sample data and high-quality embeddings, lacks validation at the scale of large LoRA repositories, and provides theoretical guarantees only for sequence-level routing, leaving token-level routing heuristic.

Although the rebuttal provides additional experiments and clarifications, it does not adequately resolve the core concerns about soundness and evaluation fairness. In particular, the explanation for the reported “2-sample robustness” reveals an implicit deviation from the mathematical formulation, undermining confidence in the theoretical consistency of the method. Therefore, this paper is not ready for acceptance in this form.

**Reviewer Concerns:**

Some concerns are addressed by the rebuttal, but Reviewer TB3c may not be convinced during the rebuttal period.

**Reviewer Scores:**

The final score could be 6644.

---

### Decision · Program_Chairs · 2026-01-26

Reject